# Improved Analysis of Sparse Linear Regression in Local Differential Privacy Model

**Liyang Zhu**[1*]**, Meng Ding**[2*]**, Vaneet Aggarwal**[3] **, Jinhui Xu**[2] **, Di Wang**[1]
[1]PRADA Lab, King Abdullah University of Science and Technology
[2]State University of New York at Buffalo
[3]Purdue University
{liyang.zhu, di.wang}@kaust.edu.sa
{mengding, jinhu}@buffalo.edu
vaneet@purdue.edu

## Abstract

In this paper, we revisit the problem of sparse linear regression in the local differential privacy (LDP) model. Existing research in the non-interactive and sequentially local models has focused on obtaining the lower bounds for the case where the underlying parameter is 1-sparse, and extending such bounds to the more general $k$-sparse case has proven to be challenging. Moreover, it is unclear whether efficient non-interactive LDP (NLDP) algorithms exist. To address these issues, we first consider the problem in the $\epsilon$ non-interactive LDP model and provide a lower bound of $\Omega(\frac{\sqrt{dk \log d}}{\sqrt{n}\epsilon})$ on the $\ell_2$-norm estimation error for sub-Gaussian data, where $n$ is the sample size and $d$ is the dimension of the space. We propose an innovative NLDP algorithm, the very first of its kind for the problem. As a remarkable outcome, this algorithm also yields a novel and highly efficient estimator as a valuable by-product. Our algorithm achieves an upper bound of $\tilde{O}(\frac{d\sqrt{k}}{\sqrt{n}\epsilon})$ for the estimation error when the data is sub-Gaussian, which can be further improved by a factor of $O(\sqrt{d})$ if the server has additional public but unlabeled data. For the sequentially interactive LDP model, we show a similar lower bound of $\Omega(\frac{\sqrt{dk}}{\sqrt{n}\epsilon})$. As for the upper bound, we rectify a previous method and show that it is possible to achieve a bound of $\tilde{O}(\frac{k\sqrt{d}}{\sqrt{n}\epsilon})$. Our findings reveal fundamental differences between the non-private case, central DP model, and local DP model in the sparse linear regression problem.

## 1 Introduction

Protecting data privacy is a major concern in many modern information or database systems. Such systems often contain personal and sensitive information, making it essential to preserve privacy when sharing aggregated data. Traditional data analysis techniques such as linear regression often face a number of challenges when dealing with sensitive data, especially in social research (Serlin & Marascuilo, 1988; Bůžková, 2013; Bühlmann & de Geer, 2011). Differential privacy (DP) (Dwork et al., 2006b) has emerged as a widely recognized approach for privacy-preserving, which provides verifiable protection against identification and is resistant to arbitrary auxiliary information that attackers may have access to.

Previous research on DP has given rise to two primary user models: the central model and the local model. The central model uses a trusted central entity to handle the data, including collecting data, determining which differentially private data analysis to perform, and distributing the results. The central model is commonly used for processing census data. Different from the central model, the local model empowers individuals to control their own data, using differentially private procedures to reveal it to a server. The server then "merges" the private data of each individual into a resultant data analysis. This paradigm is exemplified by Google's Chrome browser and Apple's iOS-10, which collect statistics from user devices (Tang et al., 2017; Erlingsson et al., 2014).

The local model, despite its widespread application in industry, has received less attention than the central model. This is because there are inherent constraints to what can be done in the local model,

---

[*]Equal contributions. Part of the work was done when Meng Ding was a research intern at KAUST.

resulting in many fundamental problems remaining unanswered. Linear regression, a fundamental model in both machine learning and statistics, has been extensively studied in recent years in the DP community in two different settings: the (stochastic) optimization and the (statistical) estimation settings. In the former, the aim is to find a private estimator $\theta \in \mathbb{R}^d$ that minimizes the empirical risk $L(\theta, D) = \frac{1}{n}\sum_{i=1}^{n}(\langle x_i, \theta \rangle - y_i)^2$ or population risk $L_{\mathcal{P}}(\theta) = \mathbb{E}_{(x,y)\sim\mathcal{P}}[(\langle x, \theta \rangle - y)^2]$ of the given dataset $D = \{(x_i, y_i)\}_{i=1}^{n}$, where $\mathcal{P}$ is the underlying distribution of $(x, y)$ with covariate $x$ and response $y$. In the latter, it considers a linear model with covariate $x$ and response $y$ that satisfy $y = \langle x, \theta^* \rangle + \zeta$, where $\zeta$ is a zero-mean random noise, and $\theta^*$ is an underlying parameter. The goal is to find a private estimator $\theta^{priv}$ that approximates $\theta^*$ as closely as possible, with the $\ell_2$-norm estimation error $\|\theta^{priv} - \theta^*\|_2$ being minimized.

DP linear regression has been extensively studied in the central model, including both the optimization and estimation settings (see the Related Work section A for further details). Recently, researchers have also investigated the problem in the high-dimensional space where the dimensionality is much larger than the sample size. For instance, Talwar et al. (2015) focused on the private LASSO problem in the optimization setting, where the underlying constraint set is an $\ell_1$-norm ball. In the estimation setting, which involves sparse linear regression where $\theta^*$ is a $k$-sparse vector with $k \ll d$, Cai et al. (2021); Kifer et al. (2012a) considered sub-Gaussian covariates, and this was later expanded by Hu et al. (2022) to include heavy-tailed covariates.

Despite the growing interest in DP linear regression, there is still a lack of understanding of the local model compared to the central one. While there have been numerous studies on the optimization setting in the low-dimensional case (Duchi et al., 2018), less attention has been paid to the statistical estimation setting in the high-dimensional space. Although Wang & Xu (2021) provided the first study on sparse linear regression in the LDP model, the problem is still far from well-understood in comparison to the non-private and central DP cases. By and large, there are three main challenges that need to be addressed. Firstly, for lower bounds, Wang & Xu (2021) only considered the 1-sparse case and their proof cannot be extended to the general $k$-sparse case. Thus, there is still no lower bound for the $k$-sparse case. Secondly, in the non-interactive setting, it is unclear whether there exists any efficient algorithm due to the non-interactivity constraint and the sparsity nature of the problem. Even for the 1-sparse case, Wang & Xu (2021) showed only a lower bound. Finally, for the upper bound in the sequentially interactive setting, while Wang & Xu (2021) provided an algorithm, it heavily relies on the assumption that the covariate follows the uniform distribution of $\{-1, +1\}^d$, and there are some technical flaws in their analysis (see Section 4.2 for details).

In this paper, we address the three challenges of sparse linear regression in the LDP model that were left by previous research. Specifically, we provide new hard instances of lower bounds, novel self-interested NLDP algorithms, and new proof techniques for lower and upper bounds. Our contributions are three-fold. See Table 1 in Appendix for comparisons with previous work.

1. In the first part of this paper, we focus on the non-interactive setting. For the $k$-sparse case, we show that even with 1-sub-Gaussian covariates and responses, the output of any $\epsilon$-NLDP algorithm must have an estimation error of at least $\Omega(\sqrt{\frac{dk \log d}{n\epsilon^2}})$, where $n$ is the sample size and $d$ is the dimension of the space. This lower bound is significantly different from the optimal rates achieved by non-private and central DP algorithms. Moreover, previous results only consider the case where $k = 1$ and it is technically difficult to extend to the general $k$-sparse case. Prior to our work, there were no comparable lower bounds. We give non-trivial proofs for our lower bounds by constructing hard instances that might be instructive for other related problems.

2. Then, we consider upper bounding the estimation error for our problem. Due to the constraints in the model, there is no previous study. We develop a novel and closed-form estimator for sparse linear regression and propose the first $(\epsilon, \delta)$-NLDP algorithm. We also give a non-trivial upper bound of $\tilde{O}(\frac{d\sqrt{k \log \frac{1}{\delta}}}{\sqrt{n}\epsilon})$ when the covariates and responses are sub-Gaussian and $n$ is large enough. Moreover, we show that if the server has enough public but unlabeled data, an error bound of $\tilde{O}(\sqrt{\frac{dk \log \frac{1}{\delta}}{n\epsilon^2}})$ can be achieved. Fianlly, we relax the assumption to the case where the responses only have bounded $2p$-moment with some $p > 1$.

3. In the second part of the paper, we investigate the problem in the sequentially interactive model. First, for sub-Gaussian data, we establish a lower bound of $\Omega(\sqrt{\frac{dk}{n\epsilon^2}})$, which is similar to the

non-interactive case, but requires a different hard instance construction and proof technique. The investigation on upper bound in the interactive setting is still quite deficient. We thus rectify and generalize the private iterative hard thresholding algorithm in Wang & Xu (2021) for sub-Gaussian covariates and responses and demonstrate that the algorithm can only achieve an upper bound of $\tilde{O}(\frac{k\sqrt{d}}{\sqrt{n}\epsilon})$ rather than $\tilde{O}(\frac{\sqrt{dk}}{\sqrt{n}\epsilon})$ in Wang & Xu (2021).

To adhere to space limitations, certain additional sections, including the related work section, along with all omitted proofs, have been included in the Appendix.

## 2 PRELIMINARIES

This section introduces the problem setting, local differential privacy, and some notations used throughout this paper. Additional preliminaries can be found in Section D of the Appendix.

**Notations.** Given a matrix $X \in \mathbb{R}^{n \times d}$, let $x_i^T$ be its $i$-th row and $x_{ij}$ (or $[X]_{ij}$) be its $(i, j)$-th entry (which is also the $j$-th element of the vector $x_i$). For any $p \in [1, \infty]$, $\|X\|_p$ is the $p$-norm, i.e., $\|X\|_p := \sup_{y \neq 0} \frac{\|Xy\|_p}{\|y\|_p}$, and $\|X\|_{\infty,\infty} = \max_{i,j} |x_{ij}|$ is the max norm of matrix $X$. For an event $A$, we let $I[A]$ denote the indicator, i.e., $I[A] = 1$ if $A$ occurs, and $I[A] = 0$ otherwise. The sign function of a real number $x$ is a piece-wise function which is defined as $\operatorname{sgn}(x) = -1$ if $x < 0$; $\operatorname{sgn}(x) = 1$ if $x > 0$; and $\operatorname{sgn}(x) = 0$ if $x = 0$. We also use $\lambda_{\min}(X)$ to denote the minimal eigenvalue of $X$. For a sub-Gaussian random variable $X$, its sub-Gaussian norm $\|X\|_{\psi_2}$ is defined as $\|X\|_{\psi_2} = \inf\{c > 0 : \mathbb{E}[\exp(\frac{X^2}{c^2})] \leq 2\}$.

### 2.1 PROBLEM SETTING

Throughout the paper, we consider the classical setting of sparse linear regression. Suppose that we have a data universe $\mathcal{D} = \mathcal{X} \times \mathcal{Y} \subseteq \mathbb{R}^d \times \mathbb{R}$ and $n$ users in the population, where each user $i$ has a feature vector $x_i \in \mathcal{X}$ and a response variable $y_i \in \mathcal{Y}$. We assume that $\{(x_i, y_i)\}_{i=1}^n$ are i.i.d. sampled from a sparse linear regression model, i.e., each $(x_i, y_i)$ is a realization of the sparse linear regression model $y = \langle \theta^*, x \rangle + \zeta$, where the distribution of $x$ has mean zero, $\zeta$ is some randomized noise that satisfies $\mathbb{E}[\zeta|x] = 0$, and $\theta^* \in \mathbb{R}^d$ is the underlying sparse estimator with $\|\theta^*\|_0 \leq k$. In the following, we provide some assumptions related to the model.

**Assumption 1.** *We assume that $\|\theta^*\|_1 \leq 1$. Moreover, for the covariance matrix of $x$, $\Sigma$, there exist $\kappa_\infty$ and $\kappa_x$ such that $\|\Sigma w\|_\infty \geq \kappa_\infty \|w\|_\infty, \forall w \neq 0$ and $\|\Sigma^{-\frac{1}{2}} x\|_{\psi_2} \leq \kappa_x$.* [1]

**Remark 1.** Due to the hardness of the problem in the NLDP model, rather than making a bounded $\ell_2$-norm assumption, we consider a stronger one where $\|\theta^*\|_1 \leq 1$, which has been previously studied in the literature such as Chen et al. (2023; 2022a); Fan et al. (2021). It is notable that all the lower bounds established in our paper remain valid even in the weaker case $\|\theta^*\|_2 \leq 1$, without any modifications to the proofs. Moreover, our upper bound in the interactive setting will still not be changed under the $\ell_2$ assumption. However, our upper bound for the NLDP model relies on such an assumption. The only result that relies on the $\ell_1$ assumption is our upper bound in the NLDP model.

Our focus in this paper is on estimating $\theta^*$ in the local differential privacy (LDP) model. We aim to design a locally differentially private algorithm that produces an output $\hat{\theta}^{priv}$ that is as close as possible to the true $\theta^*$, with the goal of minimizing the $\ell_2$-norm error $\|\hat{\theta}^{priv} - \theta^*\|_2$. We provide definitions related to LDP in Appendix B, with more detailed information available in reference Duchi et al. (2014a; 2018).

## 3 IMPROVED ANALYSIS FOR NON-INTERACTIVE SETTING

### 3.1 LOWER BOUND FOR GENERAL $k$-SPARSE CASE

In this section, we analyze the lower bound for the estimation error of non-interactive local differential privacy (NLDP) algorithms. According to Bun et al. (2019), any $(\epsilon, \delta)$-NLDP protocol can be transformed into an $\epsilon$-NLDP protocol without affecting its utility. [2] Therefore, we will focus solely on $\epsilon$-NLDP. To establish the lower bound, we consider a class of distributions for $(x, y)$, where $x$ follows the uniform distribution over $\{-1, +1\}^d$, and $\zeta$ is a bounded randomized noise. We denote

---

[1] To make our results comparable to the previous results and for simplicity, in this paper we assume $\kappa_\infty$ and $\kappa_x$ all are constants. Note that previous studies on private regression also hide factors related to $\Sigma$ in their main context(e.g. Wang (2018); Wang et al. (2022); Cai et al. (2021) hide the term of poly $(1/\lambda_{\min}(\Sigma))$.

[2] The lower bound results also hold for $(\epsilon, \delta)$-NLDP protocol.

$\mathcal{P}_{k,d,C}$ as

$$\mathcal{P}_{k,d,C} = \{P_{\theta,\zeta} \mid \exists \theta \in \mathbb{R}^d \text{ s.t. } \|\theta\|_1 \leq 1, \|\theta\|_0 \leq k, x \sim \text{Uniform}\{+1,-1\}^d, y = \langle \theta, x \rangle + \zeta, \\ \text{where } \zeta \text{ satisfies } \mathbb{E}[\zeta] = 0 \text{ and } |\zeta| \leq C\}. \tag{1}$$

Based on the notation introduced above, it is evident that for any given data $D = \{(x_i, y_i)\}_{i=1}^n \sim P_{\theta,\zeta}^{\otimes n}$, where $P_{\theta,\zeta} \in \mathcal{P}_{k,d,C}$, each $\|x_i\|_2 = \sqrt{d}$. Consequently, if we use the Gaussian mechanism on each $x_i$ to ensure $(\epsilon, \delta)$-non-interactive LDP, the scale of the noise should be $O(\frac{d}{\epsilon})$. This scaling implies that the Gaussian noise would introduce an error of $\text{Poly}(d)$. In the following, we will generalize the above observation and show for all NLDP algorithms, such polynomial dependency on $d$ is unavoidable.

**Theorem 1.** *Given $0 < \epsilon \leq 1$ and an error $\nu \leq \frac{1}{\sqrt{k}}$ , consider the distribution class $\mathcal{P}_{k,d,2}$, if for any $P_{\theta,\zeta} \in \mathcal{P}_{k,d,2}$, given $D = \{(x_1, y_1) \ldots (x_n, y_n)\}$ i.i.d. sampled from $P_{\theta,\zeta}$ there is an $\epsilon$ non-interactive LDP algorithm $\mathcal{A}$ whose output satisfies $\mathbb{E}_{\mathcal{A}, D \sim P_{\theta,\zeta}^{\otimes n}} [\|\mathcal{A}(D) - \theta\|_2] \leq \frac{\nu}{8}$. Then, we must have $n \geqslant \Omega(\frac{dk \log(\frac{d}{k})}{\nu^2 \epsilon^2})$.*

**Remark 2.** It is noteworthy that the class of distributions $\mathcal{P}_{k,d,C}$ reduces to the set of distributions studied in Wang & Xu (2021; 2019) when $k = 1$. The above theorem asserts that for any $\epsilon$-NLDP algorithm $\mathcal{A}$ with $0 < \epsilon \leq 1$, there exists an instance $P_{\theta,\zeta} \in \mathcal{P}_{k,d,2}$ such that $\mathbb{E}_{\mathcal{A}, D \sim P_{\theta,\zeta}^{\otimes n}}[\|\mathcal{A}(D) - \theta\|_2] \geq \Omega(\sqrt{\frac{dk \log \frac{d}{k}}{n\epsilon^2}})$. In contrast to the optimal rate of $O(\sqrt{\frac{k \log \frac{d}{k}}{n}})$ for the $\ell_2$-norm estimation error in the non-private case (Raskutti et al., 2011), and the nearly optimal rate of $O(\max\{\sqrt{\frac{k \log \frac{d}{k}}{n}}, \frac{k \log d}{n\epsilon}\})$ in the central $(\epsilon, \delta)$-DP model (Cai et al., 2021), for NLDP model we observe an additional factor of $O(\frac{\sqrt{d}}{\epsilon})$ and $O(\max\{\frac{\sqrt{d}}{\epsilon}, \frac{\sqrt{nd}}{\sqrt{k \log d}}\})$, respectively. These results indicate that sparse linear models in the non-interactive LDP setting are ill-suited for high-dimensional scenarios where $n \ll d$.

Theorem 1 recovers the lower bound of $\Omega(\sqrt{\frac{d \log d}{n\epsilon^2}})$ in Wang & Xu (2019; 2021) when $k = 1$. Thus, Theorem 1 is more general than previous work. Notably, our proof of the lower bound differs significantly from that in Wang & Xu (2019; 2021) where the private Fano's Lemma in Duchi et al. (2018) was mainly employed. The aim was to construct an $r$-separated family of distributions $\{P_v\}_{v \in \mathcal{V}}$ for some set $\mathcal{V}$ such that the term $\mathcal{C}^\infty\{P_v\}_{v \in \mathcal{V}}$ is minimized, where

$$\mathcal{C}^\infty\{P_v\}_{v \in \mathcal{V}} = \frac{1}{|\mathcal{V}|} \sup_{\gamma \in \mathbb{B}_\infty} \sum_{v \in \mathcal{V}} (\phi_v(\gamma))^2.$$

Here, each linear functional $\phi_v : \mathbb{B}_\infty \mapsto \mathbb{R}$ is defined by $\phi_v(\gamma) = \int \gamma(x)(dP_v(x) - d\bar{P}(x))$ with $\mathbb{B}_\infty = \{\gamma : \mathcal{X} \mapsto \mathbb{R} | \|\gamma\|_\infty \leq 1\}$ as the set of uniformly bounded functions, and $\bar{P}(x) = \frac{1}{|\mathcal{V}|} \sum_{v \in \mathcal{V}} P_v(x)$ is the average distribution. Wang & Xu (2019; 2021) considered the case where $\mathcal{V}$ is the set of all basis vectors, which is 1-sparse. They showed that for some $r$-separated family of distributions, $\mathcal{C}^\infty\{P_v\}_{v \in \mathcal{V}} \leq \frac{r^2}{d}$. However, their approach is challenging to extend to the $k$-sparse case as $|\mathcal{V}| = O(d^k)$, and it is difficult to bound the summation term. To overcome this difficulty, we adopt a private version of the Assouad's lemma in Acharya et al. (2022). In details, we first construct a random vector $Z \in \{-1, 0, +1\}^d$ with $\|Z\|_0 \leq k$ with high probability. For each realization of $Z$, $z$, we have an associated $\theta_z$ which is also $k$-sparse. Suppose $\tilde{D}$ is the message obtained via the $\epsilon$ non-interactive LDP algorithm $\mathcal{A}$ on $D \sim P_{\theta_z,\zeta}^{\otimes n}$. We consider the mutual information between $Z$ and $\tilde{D}$, i.e., $I(Z \wedge \tilde{D})$. On the one hand, we demonstrate that any sufficiently accurate (private) estimation protocol must provide sufficient information about each $Z_i$ from the messages $\tilde{D}$, which is reflected by the lower bound on mutual information $I(Z \wedge \tilde{D}) \geq \Omega(k \log \frac{d}{k})$. On the other hand, we show that if the output of algorithm $\mathcal{A}$ achieves an estimation error of $\nu$, the mutual information scales as the privacy budget $\epsilon$, which is reflected by the upper bound on mutual information $I(Z \wedge \tilde{D}) \leq O(\frac{n\epsilon^2 \nu^2}{d})$.

### 3.2 Efficient Non-interactive LDP Algorithms

In the preceding section, we established a lower bound of $\Omega(\sqrt{\frac{dk \log d}{n\epsilon^2}})$. This suggests that high-dimensional sparse linear regression, where $n \ll d$, becomes effortless in the NLDP model. However,

this raises two questions. First, in the low dimensional case where $n \gg d$, is the lower bound tight? Second, are there efficient algorithms for this problem? In this section, we focus on the upper bound. Before that, we introduce an assumption about the distribution of $(x, y)$ to elucidate our approach.

**Assumption 2.** *There exists a constant $\sigma = O(1)$ such that the covariates (feature vectors) $x_1, x_2, \cdots, x_n \in \mathbb{R}^d$ are i.i.d. (zero-mean) sub-Gaussian random vectors with variance $\sigma^2$, and the responses $y_1, y_2, \cdots, y_n$ are i.i.d. (zero-mean) sub-Gaussian random variables with variance $\sigma^2$.*

Before presenting our method, we will outline the challenges associated with the problem at hand and explain why existing (non-private) methods are not suitable for our purposes. In the private and classical setting, where the $\ell_2$-norm of each $(x_i, y_i)$ is bounded by some constant, the most direct approach is to perturb the sufficient statistics locally (Smith et al., 2017; Wang et al., 2021), i.e., $\hat{\Sigma}_{XX} = \frac{1}{n} \sum_{i=1}^{n} x_i x_i^T$ and $\hat{\Sigma}_{XY} = \frac{1}{n} \sum_{i=1}^{n} x_i y_i$, by adding Gaussian matrix and Gaussian vector to each $x_i x_i^T$ and $x_i y_i$, respectively. However, in our sparse setting, such a private estimator will provide a sub-optimal bound as it does not exploit the sparsity assumption of the model. In the non-private and high dimensional sparse setting, to achieve the optimal estimation error, one approach is based on the LASSO (Raskutti et al., 2011), i.e., to minimize $\frac{1}{2n}\|Y - X\theta\|_2^2 + \lambda_n\|\theta\|_1$ with some $\lambda_n$, where $X = (x_1^T, \cdots, x_n^T)^T \in \mathbb{R}^{n \times d}$ and $Y = (y_1, \cdots, y_n)^T$. The second type of approach is based on the Dantzig estimator (Candes & Tao, 2007), i.e., solving the linear program: $\min_\theta \|\theta\|_1$ s.t. $\frac{1}{n}\left\|X^\top(X\theta - Y)\right\|_\infty \leq \lambda_n$ with some $\lambda_n$. However, these two approaches are difficult to privatize. However, the significant amount of noise needed for privatization is problematic, as it destroys the assumptions of the theoretical results for LASSO and the Dantzig estimator. We see that existing estimators for sparse linear regression all rely on solving an optimization problem, which is difficult to privatize. Nonetheless, in the classical setting, the private estimator can be obtained by adding noise to the sufficient statistics without solving an optimization problem. Therefore, a closed-form estimator will serve our purpose and it can be used to design an efficient private estimator for the sparse linear model. This approach will minimize the amount of noise added to the model.

Before showing our private estimator, we first consider the non-private case. As we focus on the low dimension case, the empirical covariance matrix $\hat{\Sigma}_{XX}$ always exists. Thus, if there is no sparse assumption, the optimal estimator will be the ordinary least square (OLS) estimator $\hat{\Sigma}_{XX}^{-1}\hat{\Sigma}_{XY}$ given the dataset. However, as now $\theta^*$ is $k$-sparse, the OLS estimation will have a large estimation error since it is not sparse. Intuitively, our goal is to find a sparse estimator that is close to OLS, i.e., $\arg\min_\theta \|\theta - \hat{\Sigma}_{XX}^{-1}\hat{\Sigma}_{XY}\|_2^2$, s.t. $\|\theta\|_0 \leq k$, whose $\ell_1$ convex relaxation of the $\ell_0$ constraint is equivalent to $\arg\min_\theta \|\theta - \hat{\Sigma}_{XX}^{-1}\hat{\Sigma}_{XY}\|_2^2 + \lambda_n\|\theta\|_1$ with some $\lambda_n > 0$. Fortunately, the above minimizer is just the proximal operator on OLS: $\text{Prox}_{\lambda_n\|\cdot\|_1}(\hat{\Sigma}_{XX}^{-1}\hat{\Sigma}_{XY})$. Since the proximal operator is separable with respect to both vectors, $\theta$ and $\hat{\Sigma}_{XX}^{-1}\hat{\Sigma}_{XY}$,

$$(\text{Prox}_{\lambda_n\|\cdot\|_1}(\hat{\Sigma}_{XX}^{-1}\hat{\Sigma}_{XY}))_i = \arg\min_{\theta_i}(\theta_i - (\hat{\Sigma}_{XX}^{-1}\hat{\Sigma}_{XY})_i)^2 + \lambda_n|\theta_i|$$

$$= \text{sgn}((\hat{\Sigma}_{XX}^{-1}\hat{\Sigma}_{XY}))_i) \max\{|(\hat{\Sigma}_{XX}^{-1}\hat{\Sigma}_{XY})_i| - \lambda_n, 0\},$$

where the second equality is due to the first-order optimality condition. Thus, the previous $\ell_1$ regularized optimization problem has a closed-form optimal solution, which is denoted as $\hat{\theta}$:

$$\hat{\theta} = S_{\lambda_n}(\hat{\Sigma}_{XX}^{-1}\hat{\Sigma}_{XY}), \tag{2}$$

where for a given thresholding parameter $\lambda$, the element-wise *soft-thresholding* operator $S_\lambda : \mathbb{R}^d \mapsto \mathbb{R}^d$ for any $u \in \mathbb{R}^d$ is defined as the following: the $i$-th element of $S_\lambda(u)$ is defined as $[S_\lambda(u)]_i = \text{sgn}(u_i)\max(|u_i| - \lambda, 0)$.

Motivated by (2) and the preceding discussion, a direct approach to designing a private estimator is perturbing the terms of $\hat{\Sigma}_{XX}$ and $\hat{\Sigma}_{XY}$ in (2). However, the unbounded $\ell_2$-sensitivity of both terms under Assumption 2 suggests that we must preprocess the data before applying the Gaussian mechanism. Since each $x_i$ is sub-Gaussian, we can readily ensure that $\|x_i\|_2 \leq O(\sigma\sqrt{d \log n})$ for all $i \in [n]$ with high probability. Thus, we typically preprocess the data by $\ell_2$-norm clipping, i.e., $\bar{x}_i = \min\{\|x_i\|_2, r\}\frac{x_i}{\|x_i\|_2}$, where $r = O(\sigma\sqrt{d \log n})$ (Hu et al., 2022; Wang et al., 2022). However, if we preprocess each $x_i$ and $y_i$ in (2) using this strategy, it becomes difficult to bound the term $\|\hat{\theta} - \theta^*\|_\infty$, which is crucial for utility analysis.

To address the challenge, we propose a new approach. For the term of $\hat{\Sigma}_{XX}$, we use the ordinary $\ell_2$-norm clipping to each $x_i$ and get $\bar{x}_i$, and then add Gaussian matrix to $\bar{x}_i\bar{x}_i^T$. For the term $\hat{\Sigma}_{XY}$, we shrink each coordinate of $x_i$ and each $y_i$ via parameters $\tau_1$ and $\tau_2$ respectively, i.e., $\widetilde{x}_{ij} = \text{sgn}\,(x_{ij})\min\{|x_{ij}|,\tau_1\}$ for $j \in [d]$ and $\tilde{y}_i = \text{sgn}\,(y_i)\min\{|y_i|,\tau_2\}$. Then we add Gaussian noise to $\tilde{x}_i\tilde{y}_i$. Finally, the server aggregates these noisy terms and gets a noisy and clipped (shrunken) version of $\hat{\Sigma}_{XX}$ ($\hat{\Sigma}_{XY}$), i.e., $\dot{\Sigma}_{\bar{X}\bar{X}}$ and $\dot{\Sigma}_{\widetilde{X}\widetilde{Y}}$. Finally, we get $\hat{\theta}^{priv}(D) = S_{\lambda_n}(\dot{\Sigma}_{\bar{X}\bar{X}}^{-1}\dot{\Sigma}_{\widetilde{X}\widetilde{Y}})$. See Algorithm 1 for details. In the following we show with some $\tau, \tau_1$ and $\tau_2$, the previous $\hat{\theta}^{priv}(D)$ could achieve an upper bound of $\tilde{O}(\frac{d\sqrt{k}}{\sqrt{n}\epsilon})$.

**Theorem 2.** *For any $0 < \epsilon, \delta < 1$, Algorithm 1 satisfies $(\epsilon, \delta)$ non-interactive LDP.*

**Theorem 3.** *Under Assumptions 1 and 2, if we set $\tau_1 = \tau_2 = O(\sigma\sqrt{\log n}), r = O(\sigma\sqrt{d\log n})$, and $\lambda_n = O(\frac{d\log n\sqrt{\log \frac{1}{\delta}}}{\sqrt{n}\epsilon})$ in Algorithm 1. When $n$ is sufficiently large such that $n \geq \tilde{\Omega}(\max\{\frac{d^4}{\epsilon^2\kappa_\infty}, \frac{\|\Sigma\|_2^4 d^3}{\epsilon^2\lambda_{\min}^2(\Sigma)}\})$, with probability at least $1 - O(d^{-c}) - e^{-\Omega(d)}$ for some constant $c > 0$, [3] one has*

$$\left\|\hat{\theta}^{priv}(D) - \theta^*\right\|_2 \leq O\left(\frac{d\log n\sqrt{k\log d\log\frac{1}{\delta}}}{\sqrt{n}\epsilon}\right), \tag{3}$$

*where $\tilde{\Omega}$ ignores the logarithmic terms.*

**Remark 3.** Compared with Smith et al. (2017), we improve by a factor of $O\left(\frac{\sqrt{d}}{\sqrt{k}}\right)$ in our Theorem 3. It is worth noting that in the absence of the soft-thresholding operator, the upper bound can be shown to be $\tilde{O}(\frac{d^{\frac{3}{2}}}{\sqrt{n}\epsilon})$, which is consistent with previous work on linear regression (Wang et al., 2022; Smith et al., 2017). [4] Hence, we can observe that the soft-thresholding operator plays a critical role in our private estimator. The upper bound in equation 3 has an additional factor of $\tilde{O}(\sqrt{d})$ compared to the lower bound in Theorem 1. This is due to the fact that each entry of the Gaussian matrix we added to each $\bar{x}_i\bar{x}_i^T$ is $\tilde{O}(\frac{d}{\epsilon})$, which indicates that $\|\dot{\Sigma}_{\bar{X}\bar{X}} - \Sigma\|_{\infty,\infty} \leq \tilde{O}(\frac{d}{\sqrt{n}\epsilon})$. This $\tilde{O}(\sqrt{d})$ scaling seems necessary in the NLDP model because each $\|x_i\|_2 \leq O(\sqrt{d\log n})$ with high probability, and thus, we must add noise of scale $\tilde{O}(\frac{d}{\epsilon})$ to release the covariance matrix privately. Based on this, we conjecture that the lower bound in Theorem 1 is not tight, and the upper bound is nearly optimal. We leave it as an open problem. Additionally, equation 3 holds only when $n$ is sufficiently large such that $n \geq \tilde{\Omega}(\max\{\frac{d^4}{\epsilon^2\kappa_\infty}, \frac{\|\Sigma\|_2^4 d^3}{\epsilon^2\lambda_{\min}^2(\Sigma)}\})$ to ensure that the noisy empirical covariance matrix is invertible and $\|(\dot{\Sigma}_{\bar{X}\bar{X}})^{-1}\|_\infty \leq \frac{2}{\kappa_\infty}$.

**Improved rate with public unlabeled data.** As discussed in Remark 3, the main reason for the gap of $\tilde{O}(\sqrt{d})$ between the lower and upper bounds is due to $\|\dot{\Sigma}_{\bar{X}\bar{X}} - \Sigma\|_{\infty,\infty} \leq \tilde{O}(\frac{d}{\sqrt{n}\epsilon})$. However, when compared to the non-private case where the error is $\|\hat{\Sigma}_{XX} - \Sigma\|_{\infty,\infty} \leq \tilde{O}(\frac{1}{\sqrt{n}})$, we can see that the error due to the Gaussian matrix dominates. Since estimating the covariance matrix does not require the responses, we can use public but unlabeled data to achieve an improved estimation rate. It is worth noting that NLDP with public unlabeled data has been widely studied in recent years (Wang et al., 2022; Su et al., 2023; Daniely & Feldman, 2019). Here we assume that the server has access to $m$ unlabeled data points $D^{pub} = \{x_j\}_{j=n+1}^{n+m} \subset \mathcal{X}^m$, where each $x_j$ is sampled from the same sub-Gaussian distribution as $x_i$ in Assumption 2. Based on the above observations, rather than using private data, we can utilize these public data points to estimate the underlying covariance matrix. Subsequently, we propose our private estimator $\hat{\theta}^{unl}(D) = [\hat{\Sigma}_{XX}^{pub}]^{-1}\dot{\Sigma}_{\widetilde{X}\widetilde{Y}}$, where $\hat{\Sigma}_{XX}^{pub} = \frac{1}{m}\sum_{j=n+1}^{n+m} x_j x_j^T$ is the empirical covariance matrix of $\{x_j\}_{j=n+1}^{n+m}$. The details are provided in Algorithm 3. The following result shows that we can improve the estimation error by a factor of $O(\sqrt{d})$ compared with that in Theorem 3.

---

[3] Here we use $O(d^{-c})$ as the failure probability is for simplicity, we can get a similar result for any failure probability $\delta' > 0$. The same for other results in the following parts.

[4] It should be noted that Smith et al. (2017) assumes $\|x_i\|_2 \leq 1$, but we can get a bound of $\tilde{O}(\frac{d^{\frac{3}{2}}}{\sqrt{n}\epsilon})$ when we extend to $\|x_i\|_2 \leq \sqrt{d}$ via the same proof in Smith et al. (2017).

---

**Algorithm 1** Non-interactive LDP algorithm for Sparse Linear Regression

---

1: **Input:** Private data $\{(x_i, y_i)\}_{i=1}^n \in \left(\mathbb{R}^d \times \mathbb{R}\right)^n$. Predefined parameters $r, \tau_1, \tau_2, \lambda_n$.

2: **for** Each user $i \in [n]$ **do**

3:     Clip $\bar{x}_i = x_i \min\left\{1, \frac{r}{\|x_i\|_2}\right\}$. Add noise $\widehat{\bar{x}_i \bar{x}_i^T} = \bar{x}_i \bar{x}_i^T + n_{1,i}$, where $n_{1,i} \in \mathbb{R}^{d \times d}$ is a symmetric matrix and each entry of the upper triangular matrix is sampled from $\mathcal{N}(0, \frac{32 r^4 \log \frac{2.5}{\delta}}{\epsilon^2})$.
Release $\widehat{\bar{x}_i \bar{x}_i^T}$ to the server.

4:     **for** $j \in [d]$ **do**

5:         Coordinately shrink $\widetilde{x}_{ij} = \operatorname{sgn}(x_{ij}) \min\{|x_{ij}|, \tau_1\}$

6:     **end for**

7:     Clip $\tilde{y}_i := \operatorname{sgn}(y_i) \min\{|y_i|, \tau_2\}$. Add noise $\widehat{\widetilde{x}_i \widetilde{y}_i} = \tilde{x}_i \tilde{y}_i + n_{2,i}$, where the vector $n_{2,i} \in \mathbb{R}^d$ is sampled from $\mathcal{N}(0, \frac{32 d \tau_1^2 \tau_2^2 \log \frac{2.5}{\delta}}{\epsilon^2} I_d)$. Release $\widehat{\widetilde{x}_i \widetilde{y}_i}$ to the server.

8: **end for**

9: The server aggregates $\dot{\Sigma}_{\bar{X}\bar{X}} = \frac{1}{n} \sum_{i=1}^n \widehat{\bar{x}_i \bar{x}_i^T}$ and $\dot{\Sigma}_{\widetilde{X}\widetilde{Y}} = \frac{1}{n} \sum_{i=1}^n \widehat{\widetilde{x}_i \widetilde{y}_i}$

10: The server outputs $\hat{\theta}^{priv}(D) = S_{\lambda_n}([\dot{\Sigma}_{\bar{X}\bar{X}}]^{-1} \dot{\Sigma}_{\widetilde{X}\widetilde{Y}})$.

---

**Theorem 4.** *Under Assumptions 1 and 2, we suppose the server also has access to the additional public and unlabeled dataset $D^{pub} = \{x_j\}_{j=n+1}^{n+m} \in \mathcal{X}^m$ described above. When $m$ is sufficiently large that $m \geq \tilde{\Omega}(\max\{\frac{d^2}{\kappa_\infty}, \frac{d\|\Sigma\|_2^4 \kappa_x^4}{\lambda_{\min}^2(\Sigma)}\})$, set $\tau_1 = \tau_2 = O(\sigma\sqrt{\log n})$ and $\lambda_n = O(\frac{\log n \sqrt{dk \log d \log \frac{1}{\delta}}}{\epsilon \sqrt{n}})$ in Algorithm 3, with probability at least $1 - O(d^{-c}) - e^{-\Omega(d)}$ for some constant $c > 0$, then one has*

$$\left\|\hat{\theta}^{unl}(D) - \theta^*\right\|_2 \leq O\left(\frac{\log n \sqrt{dk \log d \log \frac{1}{\delta}}}{\epsilon \sqrt{n}}\right),$$

*where $\tilde{\Omega}$ ignores the logarithmic terms.*

In addition to improving the estimation error by a factor of $O(\sqrt{d})$, our proposed estimator $\hat{\theta}^{unl}(D)$ requires a smaller number of public unlabeled data points compared to the requirements in Theorem 3. We only need $m \geq \tilde{\Omega}(\max\{\frac{d^2}{\kappa_\infty}, \frac{d\|\Sigma\|_2^4 \kappa_x^4}{\lambda_{\min}^2(\Sigma)}\})$, instead of $n \geq \tilde{\Omega}(\max\{\frac{d^4}{\epsilon^2 \kappa_\infty}, \frac{\|\Sigma\|_2^4 d^3}{\epsilon^2 \lambda_{\min}^2(\Sigma)}\})$ in Theorem 3. This is because we do not need to estimate the covariance matrix privately.

**Estimation error for heavy-tailed responses.** In the preceding parts, our focus has been on the sub-Gaussian case, where both $x$ and $y$ are sub-Gaussian, meaning that the random noise $\zeta$ is sub-Gaussian as well. However, this assumption may be too stringent in real-world scenarios, where heavy-tailed noise is more commonly encountered. Our method is highly adaptable and can handle such heavy-tailed cases with ease. Here we consider the heavy-tailed case where the responses have only bounded $2p$-moment with some $p > 1$. This assumption has been widely studied in both the differential privacy and robust statistics communities (Hu et al., 2022; Kamath et al., 2020; Sun et al., 2020; Chen et al., 2022b).

**Assumption 3.** *There exist constants $\sigma$ and $M$ such that the covariates (feature vectors) $x_1, x_2, \cdots, x_n \in \mathbb{R}^d$ are i.i.d. (zero-mean) sub-Gaussian random vectors with variance $\sigma^2$ and $\forall i = 1, \ldots, n, \mathbb{E}[|y_i|]^{2p} \leq M < \infty$ for some (known) $p > 1$.*

**Theorem 5.** *Under Assumptions 1 and 3, we set $\tau_1 = O(\sigma\sqrt{\log n}), \tau_2 = (\frac{n}{\log d})^{\frac{1}{2p}}, r = O(\sigma\sqrt{d\log n})$, and $\lambda_n = O(d \log n \sqrt{\log \frac{1}{\delta}} (\frac{\log d}{n\epsilon^2})^{\frac{p-1}{2p}})$ in Algorithm 1, then as long as $n \geq \tilde{\Omega}(\max\{\frac{d^4}{\epsilon^2 \kappa_\infty}, \frac{\|\Sigma\|_2^4 d^3}{\epsilon^2 \lambda_{\min}^2(\Sigma)}\})$ for some constant $c > 0$, one has*

$$\left\|\hat{\theta}^{priv}(D) - \theta^*\right\|_2 \leq O\left(d \log n \sqrt{k \log \frac{1}{\delta}} \left(\frac{\log d}{n\epsilon^2}\right)^{\frac{p-1}{2p}}\right), \tag{4}$$

*where $\tilde{\Omega}$ ignores the logarithmic terms.*

The limit of the bound in equation 4 as $p \to \infty$ is the same as in equation 3. However, due to the heavy-tailed nature of the response variable $y$, our estimator requires more aggressive shrinking than

in the sub-Gaussian case. Hence, unlike the sub-Gaussian case, we have $\tau_1 \neq \tau_2$. It is worth noting that our current approach relaxes the assumption on the distribution of $y$ only. We anticipate that our general framework can also handle scenarios where the distributions of both $x$ and $y$ are heavy-tailed, which we plan to explore in future work.

## 4 IMPROVED ANALYSIS FOR INTERACTIVE LDP

In the previous section, we studied both the lower bound and upper bound of sparse linear regression in the non-interactive model and showed that even for $O(1)$-sub-Gaussian data, it is impossible to avoid the polynomial dependency on the dimension $d$ in the estimation error. However, since non-interactive protocols have more constraints compared to interactive ones, a natural question arises as to whether we can obtain better lower and upper bounds in the interactive model. To simplify the analysis, we mainly focus on sequentially interactive LDP protocols in this section, and note that all results can be extended to the fully interactive LDP model (Acharya et al., 2022).

### 4.1 LOWER BOUND FOR GENERAL $k$-SPARSE CASE

We begin by considering the lower bound, similar to the previous section. When $k = 1$, Wang & Xu (2021) provides a nearly optimal lower bound of $\Omega(\sqrt{\frac{d}{n\epsilon^2}})$ for the estimation error. Thus, we are more interested in whether we can obtain an improved rate for general $k$. Unfortunately, we will show that, for the same distribution class $\mathcal{P}_{k,d,2}$ as in Section 3.1, the term of $O(\sqrt{k})$ in the non-interactive case cannot be improved even if we allow interactions.

**Theorem 6.** *Given $0 < \epsilon \leq 1$ and an error $\nu \leq \frac{1}{4\sqrt{2k}}$, consider the distribution class $\mathcal{P}_{k,d,2}$, if for any $P_{\theta,\zeta} \in \mathcal{P}_{k,d,2}$, given $D = \{(x_1, y_1) \ldots (x_n, y_n)\}$ i.i.d. sampled from $P_{\theta,\zeta}$, there is an $\epsilon$-sequentially interactive LDP algorithm $\mathcal{A}$ whose output satisfies $\mathbb{E}_{\mathcal{A}, D \sim P_{\theta,\zeta}^{\otimes n}}[\|\mathcal{A}(D) - \theta\|_2] \leq \nu$. Then, we have $n \geqslant \Omega\left(\frac{dk}{\nu^2 \epsilon^2}\right)$.*

**Remark 4.** The above theorem states that for the class $\mathcal{P}_{k,d,2}$ and any $\epsilon$-LDP algorithm $\mathcal{A}$ with $0 < \epsilon \leq 1$, there exists an instance $P_{\theta,\zeta} \in \mathcal{P}_{k,d,2}$ such that $\mathbb{E}_{\mathcal{A}, D \sim P_{\theta,\zeta}^{\otimes n}}[\|\mathcal{A}(D) - \theta\|_2] \geq \Omega(\sqrt{\frac{dk}{n\epsilon^2}})$. Although the difference is only $O(\sqrt{\log d})$ compared to the lower bound in the non-interactive model, the proof and the hard instance construction are entirely different. Moreover, the lower bound proof of Theorem 6 is also distinct from that of the $k = 1$ case in Wang & Xu (2021). Briefly speaking, Wang & Xu (2021) mainly uses an LDP version of the Le Cam method, where it needs to upper bound the term $\mathcal{C}^\infty \{P_v\}_{v \in \mathcal{V}}$ (which is similar to the non-interactive LDP case). In contrast, we use a private Assouad's lemma in Acharya et al. (2020).

### 4.2 LDP ITERATIVE HARD THRESHOLDING REVISITED

Regarding the upper bound, Wang & Xu (2021) considers the case where the covariates $\{x_i\}_{i=1}^n$ satisfy Assumption 1, with $x_i \sim \text{Uniform}\{-1, +1\}^d$, and $|\zeta| \leq C$ for some constant $C$. Wang & Xu (2021) aims to solve the following optimization problem in the LDP model, where $k'$ is a parameter that will be specified later.

$$\min_\theta L(\theta; D) = \frac{1}{2n} \sum_{i=1}^n \left(\langle x_i, \theta \rangle - y_i\right)^2, \text{ s.t. } \|\theta\|_2 \leq 1, \|\theta\|_0 \leq k'. \tag{5}$$

The authors proposed a method called LDP Iterative Hard Thresholding, and claimed it achieves an upper bound of $\tilde{O}(\sqrt{\frac{dk}{n\epsilon^2}})$ for the general $k$-sparse case, nearly optimal based on Theorem 6. However, this rate is mistaken. The sensitivity analysis of the per-sample gradient is incorrect under the assumption of $\|\theta^*\|_2 \leq 1$, and such analysis leads to the incorrect utility bound. Specifically, in the proof of Theorem 9 in Wang & Xu (2021), it needs to upper-bound the each term $\langle x_i, \theta_{t-1} \rangle$, where $\|\theta_{t-1}\|_2 \leq 1$ and $\|\theta_{t-1}\|_0 \leq O(k)$. They claims that this term is upper bounded by 1, but in fact it is upper bounded by $O(\sqrt{k})$. Seeing the flaw of its own, we also highlight the technical constraint. Their sensitivity and utility analysis heavily relies on the uniform distribution assumption of $x$ and the assumption that the random noise is bounded, which is challenging to extend to general distributions (such as those in Assumption 2). In this section, we aim to rectify the previous analysis and show an upper bound of $\tilde{O}(\frac{k\sqrt{d}}{\sqrt{n}\epsilon})$ for the LDP Iterative Hard Thresholding method. Moreover, we generalize to the distributions satisfying Assumption 2

For data distributions satisfying Assumption 2, to ensure bounded sensitivity of the per-sample gradient of $L(\theta; D)$, i.e., $\|x_i^T(\langle\theta, x_i\rangle - y_i)\|_2$ for $i \in [n]$, we adopt a similar strategy as in Section 3.2. That is, each user $i$ conducts the same shrinkage operation: $\widetilde{x}_{ij} = \text{sgn}(x_{ij})\min\{|x_{ij}|, \tau_1\}$ for $j \in [d]$ and $\tilde{y}_i = \text{sgn}(y_i)\min\{|y_i|, \tau_2\}$. In this case, we can see the $i$-th sample gradient satisfies $\|\tilde{x}_i^T(\langle\theta, \tilde{x}_i\rangle - \tilde{y}_i)\|_2 \le \sqrt{d}\tau_1(\sqrt{k'}\tau_1 + \tau_2)$ if $\|\theta\|_2 \le 1$ and $\|\theta\|_0 \le k'$.

To privately solve the optimization problem equation 5, we apply a combination of private randomizer (Duchi et al., 2014a) (see equation 6 in Appendix C) and the iterative hard thresholding gradient descent method to develop an $\epsilon$-LDP algorithm. In total, our approach begins by assigning each user to one of the $T$ groups $\{S_t\}_{t=1}^T$, with the value of $T$ to be specified later. During the $t$-th iteration, users with $(x, y)$ in group $S_t$ randomize their current gradients $\tilde{x}^T(\langle\tilde{x}, \theta_{t-1}\rangle - \tilde{y})$ using equation 6. Once the server receives the gradient data from each user, it executes a gradient descent step followed by a truncation step $\theta_t' = \text{Trunc}(\tilde{\theta}_t, k')$, which retains the largest $k'$ entries of $\tilde{\theta}_t$ (in terms of magnitude) and sets the remaining entries to zero. Finally, our algorithm projects $\theta_t'$ onto the unit $\ell_2$ norm ball $\mathbb{B}_2$ to get $\theta_t$. See Algorithm 2 for details.

---

**Algorithm 2** LDP Iterative Hard Thresholding

1: **Input:** Private data $\{(x_i, y_i)\}_{i=1}^n \in (\mathbb{R}^d \times \mathbb{R})^n$. Iteration number $T$, privacy parameter $\epsilon$, step size $\eta$, truncation parameters $\tau, \tau_1, \tau_2$, threshold $k'$. Initial parameter $\theta_0 = 0$.

2: For the $i$-th user with $i \in [n]$, truncate his/her data as follows: shrink $x_i$ to $\tilde{x}_i$ with $\widetilde{x}_{ij} = \text{sgn}(x_{ij})\min\{|x_{ij}|, \tau_1\}$ for $j \in [d]$, and $\tilde{y}_i := \text{sgn}(y_i)\min\{|y_i|, \tau_2\}$. Partition the users into $T$ groups. For $t = 1, \cdots, T$, define the index set $S_t = \{(t-1)\lfloor\frac{n}{T}\rfloor + 1, \cdots, t\lfloor\frac{n}{T}\rfloor\}$; if $t = T$, then $S_t = S_t \bigcup \{t\lfloor\frac{n}{T}\rfloor + 1, \cdots, n\}$.

3: **for** $t = 1, 2, \cdots, T$ **do**

4:     The server sends $\theta_{t-1}$ to all the users in $S_t$. Each user $i \in S_t$ perturbs his/her own gradient: let $\nabla_i = \tilde{x}_i^T(\langle\theta_{t-1}, \tilde{x}_i\rangle - \tilde{y}_i)$, compute $z_i = \mathcal{R}_\epsilon^r(\nabla_i)$, where $\mathcal{R}_\epsilon^r$ is the randomizer defined in equation 6 with $r = \sqrt{d}\tau_1(\sqrt{k'}\tau_1 + \tau_2)$ and send back to the server.

5:     The server computes $\tilde{\nabla}_{t-1} = \frac{1}{|S_t|}\sum_{i\in S_t} z_i$ and performs the gradient descent update $\tilde{\theta}_t = \theta_{t-1} - \eta\tilde{\nabla}_{t-1}$.

6:     $\theta_t' = \text{Trunc}(\tilde{\theta}_{t-1}, k')$.

7:     $\theta_t = \arg_{\theta\in\mathbb{B}_2}\|\theta - \theta_t'\|_2$.

8: **end for**

9: **Output:** $\theta_T$

---

**Theorem 7.** *For any $\epsilon > 0$, Algorithm 2 is $\epsilon$ sequentially interactive LDP. Moreover, under Assumptions 1 and 2, and if the distribution of $x$ is isotropic, i.e., $\Sigma = I_d$. By taking $T = O(\log n)$, $k' = 8k$, $\eta = O(1)$, $\tau_1 = \tau_2 = O(\sigma\sqrt{\log n})$, the output $\theta_T$ of the algorithm satisfies $\|\theta_T - \theta^*\|_2 \le \tilde{O}(\frac{k\sqrt{d}}{\sqrt{n}\epsilon})$ with probability at least $1 - O(d^{-c})$ for some constant $c > 0$.*

In comparison to the upper bound presented in Theorem 3 for the non-interactive case, our algorithm exhibits a noteworthy improvement by a factor of approximately $\tilde{O}(\sqrt{d}/\sqrt{k})$. This improvement stems from our approach, which eliminates the need for private estimation of the covariance matrix, thus achieving an enhancement of approximately $\tilde{O}(\sqrt{d})$. However, it is worth noting that the sensitivity of the per-user gradient in our algorithm, denoted as $\tilde{O}(\sqrt{dk})$, differs from the sensitivity of $\tilde{O}(\sqrt{d})$ associated with $\tilde{x}_i\tilde{y}_i$ in Algorithm 1. Consequently, we introduce an additional factor of approximately $\tilde{O}(\sqrt{k})$. Importantly, when compared to Wang & Xu (2021), our Theorem 7's primary contribution is extending the $\{-1, +1\}^d$ uniform distribution assumption of covariates in Wang & Xu (2021) to general $O(1)$-sub-Gaussian assumption on covariates and heavy-tailed assumption on responses, rather than improving the upper bound. In fact, our upper bound aligns with the correct bound in Wang & Xu (2021).

**Remark 5.** We can see that Theorem 7 only holds for the case where the distribution of $x$ is isotropic. Actually, we can relax this assumption, and we can show the bound $\tilde{O}(\frac{\sqrt{d}k}{\sqrt{n}\epsilon})$ also holds for general sub-Gaussian distributions. Due to the space limit, please refer to Section F in the Appendix, where we have slightly modified Algorithm 2.

ACKNOWLEDGMENTS

Di Wang and Liyang Zhu are supported in part by the baseline funding BAS/1/1689-01-01, funding from the CRG grand URF/1/4663-01-01, FCC/1/1976-49-01 from CBRC, and funding from the AI Initiative REI/1/4811-10-01 of King Abdullah University of Science and Technology (KAUST). They are also supported by the funding of the SDAIA-KAUST Center of Excellence in Data Science and Artificial Intelligence (SDAIA-KAUST AI). Vaneet Aggarwal was visiting professor in KAUST when this study was performed. The research of Meng Ding and Jinhui Xu was supported in part by NSF through grant CCF-2200173 and by KAUST through grant CRG10-4663.2.

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

| Model | Method | Setting | Upper Bound | Lower Bound | Data Assumption |
|---|---|---|---|---|---|
| Central | Cai et al. (2021) | general | $\tilde{O}(\sqrt{\frac{d}{n}}+\frac{d}{n\epsilon})$ | $\Omega(\sqrt{\frac{d}{n}}+\frac{d}{n\epsilon})$ | sub-Gaussian |
| | Kifer et al. (2012b) | sparse | $O(\frac{k^{3/2}}{\sqrt{n\epsilon}})$ | $\Omega(\sqrt{\frac{k\log d}{n}}+\frac{k\log d}{n\epsilon})$ | sub-Gaussian |
| | Cai et al. (2021) | sparse | $\tilde{O}(\sqrt{\frac{k\log d}{n}}+\frac{k\log d}{n\epsilon})$ | | sub-Gaussian |
| | Hu et al. (2022) | sparse | $\tilde{O}(\frac{k\log d}{\sqrt{n\epsilon}})$ | - | heavy-tail |
| Non-interactive Local | Wang & Xu (2021) | 1-sparse | - | $\Omega(\sqrt{\frac{d\log d}{n\epsilon^2}})$ | sub-Gaussian |
| | **Our Work** | k-sparse | $\tilde{O}(\frac{d\sqrt{k\log d}}{\sqrt{n}\epsilon})$ | $\Omega(\sqrt{\frac{dk\log d}{n\epsilon^2}})$ | sub-Gaussian |
| | **Our Work** | k-sparse | $\tilde{O}(\frac{\sqrt{dk}}{\sqrt{n}\epsilon})$ | - | sub-Gaussian with public data |
| | **Our Work** | k-sparse | $\tilde{O}\left(\frac{\sqrt{dk}}{(n\epsilon^2)^{\frac{p-1}{2p}}}\right)$ | - | heavy-tailed response |
| Interactive Local | Smith et al. (2017) | general | $\tilde{O}(\frac{d^{\frac{3}{2}}}{\sqrt{n}\epsilon})$ | - | Sub-Gaussian distribution |
| | Wang & Xu (2021) | 1-sparse | - | $\Omega(\sqrt{\frac{d}{n\epsilon^2}})$ | sub-Gaussian |
| | Wang & Xu (2021) | k-sparse | $\tilde{O}(\frac{\sqrt{dk}}{\sqrt{n}\epsilon})*$ | - | Uniform distribution |
| | **Our Work** | k-sparse | $\tilde{O}(\frac{k\sqrt{d}}{\sqrt{n}\epsilon})$ | $\Omega(\sqrt{\frac{dk}{n\epsilon^2}})$ | sub-Gaussian |

Table 1: Comparison of our work with related studies on $(\epsilon,\delta)$-DP (sparse) linear regression in the statistical estimation setting. Here, $n$ represents the sample size, $k$ denotes the sparsity, and $d$ refers to the dimension. The asterisk (*) indicates that the proof of the upper bound in Wang & Xu (2021) contains technical flaws and is deemed incorrect. In our comparison, the term "sub-Gaussian" signifies that both the covariates and responses follow $O(1)$-sub-Gaussian distributions. On the other hand, "heavy-tail" indicates that both the covariates and responses have bounded fourth moments. Additionally, "heavy-tailed response" implies that the responses possess a $2p$-moment, where $p > 1$. "Sub-Gaussian with public data" characterizes the scenario where the data is sub-Gaussian, and the server possesses additional public but unlabeled data. Lastly, "uniform distribution" describes the situation where the covariates are drawn from $\{+1, -1\}^d$, and the responses are bounded by $O(1)$.

## A    RELATED WORK

There is a significant body of research on the differentially private (sparse) linear regression problem, which has been examined from multiple perspectives, such as Alabi et al. (2022); Chen et al. (2016); Barrientos et al. (2018); Qiu et al. (2022). In this study, we mainly focus on the works that are highly relevant to our research problem. Thus, we compare the research on sparse linear regression in the central model with that on linear regression in the local model. For a detailed comparison between these two directions of research, please refer to Table 1.

**Linear regression in the central DP model.** Most studies on the optimization setting consider more general problems, such as Stochastic Convex Optimization (SCO) and Empirical Risk Minimization (ERM) (Wang, 2018). In recent years, DP-SCO and DP-ERM have been extensively studied (Bassily et al., 2019; Feldman et al., 2020; Asi et al., 2021b; Su et al., 2022; Sarathy & Vadhan, 2022). However, it is worth noting that in order to apply these results to linear regression, we need to assume that both the covariates and responses are bounded, and the constraint set of $\theta$ is also bounded to ensure that the gradient of the loss is bounded. Some works, such as Wang et al. (2020); Kamath et al. (2022); Lowy & Razaviyayn (2023), have relaxed these assumptions to allow for sub-Gaussian or even heavy-tailed distributions. In the statistical estimation setting, Cai et al. (2021) provides a nearly optimal rate of $\tilde{O}(\sqrt{\frac{d}{n}}+\frac{d\sqrt{\log(1/\delta)}}{n\epsilon})$ for the $(\epsilon,\delta)$-DP model with $O(1)$-sub-Gaussian data. Later, Varshney et al. (2022) improves upon this rate by considering the variance of the random noise $\sigma^2$. Additionally, Liu et al. (2023) extends this work to the case where some response variables are adversarially corrupted.

**Sparse linear regression in the central DP model.** In the optimization setting, the LASSO problem with an $\ell_1$-norm ball constraint set is studied in Talwar et al. (2015). The authors demonstrate that this setting leads to an excess empirical risk of $O(\frac{\log d \log n}{(n\epsilon)^{2/3}})$, which is further extended to the population risk in Asi et al. (2021a). On the other hand, Kifer et al. (2012b) provides the first study for the estimation setting and develops an efficient algorithm that achieves an upper bound of $O(\frac{k^{3/2}}{\sqrt{n\epsilon}})$.

Recently, Cai et al. (2021) has shown a nearly optimal rate of $\tilde{O}(\sqrt{\frac{k\log d}{n}}+\frac{k\log d}{n\epsilon})$ for $(\epsilon,\delta)$-DP in

the case of $O(1)$-sub-Gaussian data. Additionally, Hu et al. (2022) has established an upper bound of $\tilde{O}(\frac{k\log d}{\sqrt{n\epsilon}})$ for the scenario where the covariate and response have only bounded fourth-order moments.

**Linear regression in the local DP model.** Regarding the non-interactive case, Smith et al. (2017) has demonstrated that if both covariates and responses are bounded by some constant, the $\epsilon$ private optimal rate for the excess empirical risk is $O(\sqrt{\frac{d}{n\epsilon^2}})$, indicating a bound of only $O\left(\frac{d^{\frac{3}{2}}}{\epsilon\sqrt{n}}\right)$ under the $O(1)$-sub-Gaussian assumption. On the other hand, in the (sequentially) interactive setting, the majority of research considers the DP-SCO or DP-ERM protocols (Duchi et al., 2014b; 2018).

**Sparse linear regression in local DP model.** Compared to the three aforementioned settings, there has been relatively less research conducted on sparse linear regression in LDP. For the optimization perspective, Zheng et al. (2017) has shown that when the covariate and response are bounded by some constant, it is possible to achieve an error of $O((\frac{\log d}{n\epsilon^2})^{\frac{1}{4}})$ if the constraint set is an $\ell_1$-norm ball. However, their method cannot be extended to the statistical setting since we always assume that the covariates are $O(1)$-sub-Gaussian, which indicates that their $\ell_2$-norm is bounded by $O(\sqrt{d})$. Regarding the estimation setting, as discussed in the introduction section, Wang & Xu (2021) provides the first study. Notably, the upper bound proof in Wang & Xu (2021) has a flaw, and the correct upper bound is $O\left(\frac{\sqrt{d}k}{\epsilon\sqrt{n}}\right)$.

# B  LOCAL DIFFERENTIAL PRIVACY

**Definition 1** (Differential Privacy (Dwork et al., 2006a)). *Given a data universe $\mathcal{X}$, we say that two datasets $D, D' \subseteq \mathcal{X}$ are neighbors if they differ by only one entry, which is denoted as $D \sim D'$. A randomized algorithm $\mathcal{A}$ is $(\epsilon, \delta)$-differentially private (DP) if for all neighboring datasets $D, D'$ and for all events $S$ in the output space of $\mathcal{A}$, we have $\mathbb{P}(\mathcal{A}(D) \in S) \leq e^\epsilon \mathbb{P}(\mathcal{A}(D') \in S) + \delta$.*

Since we will consider the sequentially interactive and non-interactive local models in this paper, we follow the definitions in Duchi et al. (2018). We assume that $\{Z_i\}_{i=1}^n$ are the private observations transformed from $\{X_i\}_{i=1}^n$ through some privacy mechanisms. We say that the mechanism is sequentially interactive when it has the following conditional independence structure: $\{X_i, Z_1, \cdots, Z_{i-1}\} \mapsto Z_i, Z_i \perp X_j \mid \{X_i, Z_1, \cdots, Z_{i-1}\}$ for all $j \neq i$ and $i \in [n]$, where $\perp$ means independent relation. The full conditional distribution can be specified in terms of conditionals $Q_i(Z_i \mid X_i = x_i, Z_{1:i} = z_{1:i})$. The full privacy mechanism can be specified by a collection $Q = \{Q_i\}_{i=1}^n$. When $Z_i$ only depends on $X_i$, the mechanism is called non-interactive and in this case we have a simpler form for the conditional distributions $Q_i(Z_i \mid X_i = x_i)$. We now define local differential privacy by restricting the conditional distribution $Q_i$.

**Definition 2** (Local Differential Privacy (Duchi et al., 2018)). *For given privacy parameters $0 < \epsilon, \delta < 1$, the random variable $Z_i$ is an $(\epsilon, \delta)$ sequentially locally differentially private view of $X_i$ if for all $z_1, z_2, \cdots, z_{i-1}$ and $x, x' \in \mathcal{X}$ we have the following for all events $S$.*

$$Q_i\left(Z_i \in S \mid X_i = x_i, Z_{1:i-1} = z_{1:i-1}\right) \leq e^\epsilon Q_i\left(Z_i \in S \mid X_i = x'_i, Z_{1:i-1} = z_{1:i-1}\right) + \delta.$$

*The random variable $Z_i$ is an $(\epsilon, \delta)$ non-interactively locally differentially private (NLDP) view of $X_i$ if $Q_i\left(Z_i \in S \mid X_i = x_i\right) \leq e^\epsilon Q_i\left(Z_i \in S \mid X_i = x'_i\right) + \delta$. We say that the privacy mechanism $Q = \{Q_i\}_{i=1}^n$ is $(\epsilon, \delta)$ sequentially (non-interactively) locally differentially private (LDP) if each $Z_i$ is a sequentially (non-interactively) locally differentially private view. If $\delta = 0$, then we call the mechanism $\epsilon$ sequentially (non-interactively) LDP.*

In this paper, we mainly use the Gaussian mechanism (Dwork et al., 2006b) to guarantee $(\epsilon, \delta)$-LDP.

**Definition 3.** *(Gaussian Mechanism). Given any function $q : \mathcal{X}^n \to \mathbb{R}^p$, the Gaussian Mechanism is defined as: $\mathcal{M}_G(D, q, \epsilon) = q(D) + Y$, where $Y$ is drawn from Gaussian Distribution $\mathcal{N}\left(0, \sigma^2 I_p\right)$ with $\sigma \geq \sqrt{2\ln(1.25/\delta)}\Delta_2(q)/\epsilon$. Here $\Delta_2(q)$ is the $\ell_2$-sensitivity of the function $q$, i.e. $\Delta_2(q) = \sup_{D \sim D'} \|q(D) - q(D')\|_2$. Gaussian Mechanism preserves $(\epsilon, \delta)$-differential privacy.*

## C   PRIVATE RANDOMIZER IN SECTION 4.2

**Private randomizer.** On input $x \in \mathbb{R}^p$, where $\|x\|_2 \leq r$, the randomizer $\mathcal{R}_\epsilon^r(x)$ does the following. It first sets $\tilde{x} = \frac{brx}{\|x\|_2}$ where $b \in \{-1, +1\}$ a Bernoulli random variable $\text{Ber}\left(\frac{1}{2} + \frac{\|x\|_2}{2r}\right)$. We then sample $s \sim \text{Ber}\left(e^\epsilon / e^\epsilon + 1\right)$ and outputs $O(r\sqrt{p})\mathcal{R}_\epsilon^r(x)$, where

$$\mathcal{R}_\epsilon^r(x) = \begin{cases} \text{Uni}\left(u \in \mathbb{S}^{p-1} : \langle u, \tilde{x} \rangle > 0\right) & \text{if } s = 1 \\ \text{Uni}\left(u \in \mathbb{S}^{p-1} : \langle u, \tilde{x} \rangle \leq 0\right) & \text{if } s = 0 \end{cases} \tag{6}$$

The following lemma, which is given by Smith et al. (2017); Wang & Xu (2021), shows that each coordinate of the randomizer is sub-Gaussian $\mathcal{R}_\epsilon^r(x)$ and is unbiased.

**Lemma 8.** *Given any vector $x \in \mathbb{R}^d$ with $\|x\|_2 \leq r$, each coordinate of the randomizer $\mathcal{R}_\epsilon^r(x)$ defined above is a sub-Gaussian random vector with variance $\sigma^2 = O(\frac{r^2}{\epsilon^2})$ and $\mathbb{E}[\mathcal{R}_\epsilon^r(x)] = x$. Moreover $\mathcal{R}_\epsilon^r(\cdot)$ is $\epsilon$-DP.*

## D   SUPPORTING LEMMAS

First, we introduce the definitions and lemmas related to sub-Gaussian random variables. The class of sub-Gaussian random variables is quite large. It includes bounded random variables and Gaussian random variables, and it enjoys strong concentration properties. We refer the readers to Vershynin (2011) for more details.

**Definition 4** (Sub-Gaussian random variable)**.** *A zero-mean random variable $X \in \mathbb{R}$ is said to be sub-Gaussian with variance $\sigma^2$ $\left(X \sim \text{subG}\left(\sigma^2\right)\right)$ if its moment generating function satisfies $\mathbb{E}[\exp(tX)] \leq \exp\left(\frac{\sigma^2 t^2}{2}\right)$ for all $t > 0$. For a sub-Gaussian random variable $X$, its sub-Gaussian norm $\|X\|_{\psi_2}$ is defined as $\|X\|_{\psi_2} = \inf\{c > 0 : \mathbb{E}[\exp(\frac{X^2}{c^2})] \leq 2\}$. Specifically, if $X \sim subG(\sigma^2)$ we have $\|X\|_{\psi_2} \leq O(\sigma)$.*

**Definition 5** (Sub-exponential random variable)**.** *A random variable $X$ with mean $\mathbb{E}[X]$ is $\zeta$-sub-exponential if for all $|t| \leq \frac{1}{\zeta}$, we have $\mathbb{E}[\exp(t(X - \mathbb{E}[X]))] \leq \exp(\frac{\zeta^2 t^2}{2})$. For a sub-exponential random variable $X$, its sub-exponential norm $\|X\|_{\psi_1}$ is defined as $\|X\|_{\psi_1} = \inf\{c > 0 : \mathbb{E}[\exp(\frac{|X|}{c})] \leq 2\}$.*

**Definition 6** (Sub-Gaussian random vector)**.** *. A zero mean random vector $X \in \mathbb{R}^d$ is said to be sub-Gaussian with variance $\sigma^2$ (for simplicity, we call it $\sigma^2$-sub-Gaussian), which is denoted as $\left(X \sim \text{subG}_d\left(\sigma^2\right)\right)$, if $\langle X, u \rangle$ is sub-Gaussian with variance $\sigma^2$ for any unit vector $u \in \mathbb{R}^d$.*

**Lemma 9.** *If $X$ is sub-Gaussian or sub-exponential, then we have $\|X - \mathbb{E}[X]\|_{\psi_2} \leq 2\|X\|_{\psi_2}$ or $\|X - \mathbb{E}[X]\|_{\psi_1} \leq 2\|X\|_{\psi_1}$.*

**Lemma 10.** *For two sub-Gaussian random variables $X_1$ and $X_2$, $X_1 \cdot X_2$ is a sub-exponential random variable with*

$$\|X_1 \cdot X_2\|_{\psi_1} \leq O(\max\{\|X_1\|_{\psi_2}^2, \|X_2\|_{\psi_2}^2\}).$$

**Lemma 11.** *If $X \sim \text{subG}\left(\sigma^2\right)$, then for any $t > 0$, it holds that $\mathbb{P}(|X| > t) \leq 2\exp\left(-\frac{t^2}{2\sigma^2}\right)$.*

**Lemma 12.** *For a sub-Gaussian vector $X \sim \text{sub} G_d\left(\sigma^2\right)$, with probability at least $1 - \delta'$ we have $\|X\|_2 \leq 4\sigma\sqrt{d\log\frac{1}{\delta'}}$.*

**Lemma 13.** *(Cai & Zhou, 2012) If $\{x_1, x_2, \cdots, x_n\}$ are $n$ realizations the a (zero mean) $\sigma^2$-sub-Gaussian random vector $X$ with covariance matrix $\Sigma_{XX} = \mathbb{E}[XX^T]$, and $\hat{\Sigma}_{XX} = \left(\hat{\sigma}_{xx^T, ij}\right)_{1 \leq i, j \leq d} = \frac{1}{n}\sum_{i=1}^n x_i x_i^T$ is the empirical covariance matrix, then there exist constants $C_1$ and $\gamma > 0$ such that for any $i, j \in [d]$, we have:*

$$\mathbb{P}\left(\left\|\hat{\Sigma}_{XX} - \Sigma_{XX}\right\|_{\infty, \infty} > t\right) \leq C_1 e^{-nt^2 \frac{8}{\gamma^2}},$$

*for all $|t| \leq \phi$ with some $\phi$, where $C_1$ and $\gamma$ are constants and depend only on $\sigma^2$. Specifically,*

$$\mathbb{P}\left(\left\|\hat{\Sigma}_{XX} - \Sigma_{XX}\right\|_{\infty, \infty} \geq \gamma\sqrt{\frac{\log d}{n}}\right) \leq C_1 d^{-8}.$$

**Lemma 14.** *Let $X_1, \cdots, X_n$ be $n$ independent (zero mean) random variables such that $X_i \sim \mathrm{subG}\left(\sigma^2\right)$. Then for any $a \in \mathbb{R}^n, t > 0$, we have:*

$$\mathbb{P}\left(\left|\sum_{i=1}^n a_i X_i\right| > t\right) \leq 2\exp\left(-\frac{t^2}{2\sigma^2\|a\|_2^2}\right).$$

**Lemma 15.** *(Vershynin, 2011) Let $X_1, X_2, \cdots, X_n$ be $n$ (zero mean) random variables such that each $X_i$ is sub-Gaussian with $\sigma^2$. Then the following holds*

$$\mathbb{P}\left(\max_{i \in n} X_i \geq t\right) \leq ne^{-\frac{t^2}{2\sigma^2}},$$

$$\mathbb{P}\left(\max_{i \in n} |X_i| \geq t\right) \leq 2ne^{-\frac{t^2}{2\sigma^2}}.$$

Below is a lemma related to the Gaussian random variable. We will employ it to bound the noise added by the Gaussian mechanism.

**Lemma 16.** *Let $\{x_1, \cdots, x_n\}$ be $n$ random variables sampled from Gaussian distribution $\mathcal{N}\left(0, \sigma^2\right)$. Then*

$$\mathbb{E}\left[\max_{1 \leq i \leq n} |x_i|\right] \leq \sigma\sqrt{2\log 2n},$$

$$\mathbb{P}\left(\left\{\max_{1 \leq i \leq n} |x_i| \geq t\right\}\right) \leq 2ne^{-\frac{t^2}{2\sigma^2}}.$$

*Particularly, if $n = 1$, we have $\mathbb{P}\left(\{|x_i| \geq t\}\right) \leq 2e^{-\frac{t^2}{2\sigma^2}}$.*

**Lemma 17** (Hoeffding's inequality). *Let $X_1, \cdots, X_n$ be independent random variables bounded by the interval $[a, b]$. Then, for any $t > 0$,*

$$\mathbb{P}\left(\left|\frac{1}{n}\sum_{i=1}^n X_i - \frac{1}{n}\sum_{i=1}^n \mathbb{E}[X_i]\right| > t\right) \leq 2\exp\left(-\frac{2nt^2}{(b-a)^2}\right).$$

**Lemma 18** (Bernstein's inequality for bounded random variables). *Let $X_1, X_2, \cdots, X_n$ be independent centered bounded random variables, i.e. $|X_i| \leq M$ and $\mathbb{E}[X_i] = 0$, with variance $\mathbb{E}\left[X_i^2\right] = \sigma^2$. Then, for any $t > 0$,*

$$\mathbb{P}\left(\left|\sum_{i=1}^n X_i\right| > \sqrt{2n\sigma^2 t} + \frac{2Mt}{3}\right) \leq 2e^{-t}.$$

**Lemma 19** (Bernstein's inequality for sub-exponential random variables). *Let $x_1, \cdots, x_n$ be $n$ i.i.d. realizations of $\zeta$-subexponential random variable $X$ with zero mean. Then,*

$$\mathbb{P}(|\sum_{i=1}^n x_i| \geq t) \leq 2\exp(-c\min\{\frac{t^2}{n\|x\|_{\psi_1}^2}, \frac{t}{\|x\|_{\psi_1}}\}).$$

**Lemma 20.** *(Non-communicative Matrix Bernstein inequality (Vershynin, 2011)) Consider a finite sequence $X_i$ of independent centered symmetric random $d \times d$ matrices. Assume we have for some numbers $K$ and $\sigma$ that*

$$\|X_i\|_2 \leq K, \left\|\sum_i \mathbb{E}\left[X_i^2\right]\right\|_2 \leq \sigma^2$$

*Then, for every $t \geq 0$ we have*

$$\mathbb{P}\left(\left\|\sum_i X_i\right\|_2 \geq t\right) \leq 2p\exp\left(-\frac{t^2/2}{\sigma^2 + Kt/3}\right).$$

**Lemma 21.** *(Jain et al., 2014) For any $\theta \in \mathbb{R}^k$ and an integer $s \leq k$, if $\theta_t = \mathrm{Trunc}(\theta, s)$ then for any $\theta^* \in \mathbb{R}^k$ with $\|\theta^*\|_0 \leq s$, we have $\|\theta_t - \theta\|_2 \leq \frac{k-s}{k-k} \mid \theta^* - \theta\|_2^2$.*

**Lemma 22.** *Let $\mathcal{K}$ be a convex body in $\mathbb{R}^p$, and $v \in \mathbb{R}^p$. Then for every $u \in \mathcal{K}$, we have*

$$\|\mathcal{P}_\mathcal{K}(v) - u\|_2 \leq \|v - u\|_2$$

*where $\mathcal{P}_\mathcal{K}$ is the operator of projection onto $\mathcal{K}$.*

# E  OMITTED PROOFS

## E.1  OMITTED PROOFS IN 3.1

**Proof of Theorem 1.** We consider the hard distribution class $\mathcal{P}_{k,d,2}$, where for each instance $P_{\theta,\zeta} \in \mathcal{P}_{k,d,2}$, its random noise $\zeta$ satisfies $\mathbb{E}[\zeta] = 0, |\zeta| \leq 2$, and $\|\theta\|_1 \leq 1$ and $\|\theta\|_0 \leq k$ hold. We denote $\gamma = \frac{\nu}{\sqrt{k}}$ and define a vector $\theta_z$ where $\theta_{z,i} = \gamma z_i$ for $i \in [d]$, $\{z_i\}_{i=1}^d$ are realizations of the random variable $Z \in \{+1, 0, -1\}^d$, where each coordinate $Z_i$ is independent to others and has the following distribution:

$$\mathbb{P}\{Z_i = +1\} = \frac{k}{4d}, \quad \mathbb{P}\{Z_i = -1] = \frac{k}{4d}, \quad \mathbb{P}\{Z_i = 0\} = 1 - \frac{k}{2d}.$$

We first show that $\theta_z$ satisfies the conditions that $\|\theta\|_1$ and $\|\theta\|_0 \leq k$. The resulting $\theta_z$ has an expected $(k/2)$-sparsity and $\mathbb{E}[Z_i] = 0, \sigma^2 = \mathbb{E}[Z_i^2] = k/2d$ for all $i \in [d]$. By utilizing a Chernoff bound, we can conclude that $\theta_z$ is $k$-sparse with probability at least $1 - k/4d$ if $k \geq 4 \log d$. This will be enough for our purposes and allows us to consider the random prior of hard instances above instead of enforcing $k$-sparsity with probability one (see the followings for details). When $\theta_z$ is $k$-sparse, we can also see $\|\theta_z\|_1 \leq \sqrt{k}\nu \leq 1$ as we assume $\nu \leq \frac{1}{\sqrt{k}}$.

Next, we construct the random noise $\zeta_z$ for each $\theta_z$. We first pick $z$ randomly from $\{-1, 0, +1\}^d$ as above. For each $z$ we let:

$$\zeta_z = \begin{cases} 1 - \langle x, \theta_z \rangle & \text{w.p.} \quad 1 + \frac{\langle x, \theta_z \rangle}{2} \\ -1 - \langle x, \theta_z \rangle & \text{w.p.} \quad 1 - \frac{\langle x, \theta_z \rangle}{2} \end{cases}$$

Note that since $|\langle x, \theta_z \rangle| \leq \sqrt{k} \cdot \nu \leq 1$.

The above distribution is well-defined and $|\zeta_z| \leq 2$, $\mathbb{E}[\zeta|x] = 0$ (thus $\mathbb{E}[\zeta] = 0$). Thus we can see our for $(\theta_z, \zeta_z)$, with probability at least $1 - \frac{k}{4d}$, we have $P_{\theta_z,\zeta_z} \in \mathcal{P}_{k,d,2}$. Morevoer, we can see that density function for $(x, y)$ is $P_{\theta_z,\zeta}((x,y)) = \frac{1+y\langle x,\theta_z \rangle}{2^{d+1}}$ for $(x, y) \in \{+1, -1\}^{d+1}$. Then, for the $i$-th user who has the data sample $(x_i, y_i)$ from the distribution $P_{\theta,\zeta}$ with mean vector $\theta_Z$, he/she sends his/her through a private algorithm $\mathcal{A}$ getting a message $S_i$.

Next, we will introduce the Lemma 23 to show that supposing the accuracy of the algorithm is $\nu$, then it will provide sufficient mutual information $I(Z_i \wedge S^n)$, where $S^n$ is the tuple of messages received from the private algorithm $\mathcal{A}$. The idea of the proof follows Acharya et al. (2022).

**Lemma 23.** *Given $0 < \epsilon < 1$, if algorithm $\mathcal{A}$ is an $\epsilon$-NLDP algorithm such that, for any $n$-size dataset $\mathcal{D} = \{(x_i, y_i)\}_{i=1}^n$ consisting of i.i.d. samples from $P_{\theta_Z,\zeta_Z}$ with the random variable $Z$ has the above probabilities, and its output $\theta^{priv}$ satisfies $\mathbb{E}[\|\theta^{priv} - \theta^*\|_2] \leq \frac{\nu}{8}$, then we have $\sum_{i=1}^d I(Z_i \wedge S^n) = \Omega(k \log \frac{d}{k})$. Therefore, $I(Z \wedge S^n) = \Omega(k \log \frac{d}{k})$ holds.*

*Proof of Lemma 23.* Consider the estimator $\hat{\theta} = \hat{\theta}(S^n)$, we define another estimator $\hat{Z}$ for $Z$ by choosing

$$\hat{Z} = \underset{z \in \{-1,0,+1\}^d}{\arg\min} \|\theta_z - \hat{\theta}\|_2.$$

Specifically, $\|\theta_{\hat{Z}} - \theta_Z\|_2 \leq 2 \|\hat{\theta} - \theta_Z\|_2$ with probability 1 , and

$$\mathbb{E}\left[\|\theta_{\hat{Z}} - \theta_Z\|_2^2\right] \leq \mathbb{E}\left[\|\theta_{\hat{Z}} - \theta_Z\|_2^2 \mathbb{1}_{\{\|\theta_Z\|_0 \leq k\}}\right] + \frac{k}{4d} \cdot \max_{z,z'} \|\theta_z - \theta_{z'}\|_2^2 \leq 4 \cdot \frac{\nu^2}{64} + \frac{k}{4d} \cdot \frac{\nu^2}{k} \cdot d = \frac{3\nu^2}{8},$$

In light of the fact that $\hat{\theta}$ is a good estimator with regards to the $\ell_2$ loss and has a deviation of no more than $\nu/4$, provided that $\theta_Z$ is $k$-sparse, and considering the bound on the probability that $Z$ is not $k$-sparse, as well as the fact that the maximum difference between any two mean vectors $\theta_z, \theta_{z'}$ derived from our construction is $\nu/\sqrt{k}$, it follows that $\|\theta_{\hat{Z}} - \theta_Z\|^2 = \frac{\nu^2}{k} \sum_{i=1}^d \mathbb{1}_{Z_i \neq \hat{Z}_i}$. As a result, it implies:

$$\sum_{i=1}^d \mathbb{P}[Z_i \neq \hat{Z}_i] \leqslant \frac{3k}{8}.$$

Also, we consider the Markov chain:

$$Z_i \to S^n \to \hat{Z}_i.$$

We could derive:

$$\sum_{i=1}^{d} I(Z_i \wedge \hat{Z}_i) \le \sum_{i=1}^{d} I\left(Z_i \wedge S^n\right).$$

Next we will bound $\sum_{i=1}^{d} I(Z_i \wedge \hat{Z}_i)$. By Fano's inequality, we have for all $i$:

$$I(Z_i \wedge \hat{Z}_i) = H(Z_i) - H\left(Z_i \mid \hat{Z}_i\right) \ge h\left(\frac{k}{2d}\right) - h(\mathbb{P}[Z_i \ne \hat{Z}_i])$$

where $h(x) = -x \log x - (1-x) \log(1-x)$ is the binary entropy. Thus, we could complete the proof by the subsequent inequalities, where the penultimate inequality arises from the concavity and monotonicity of $h$.

$$\begin{aligned}
\sum_{i=1}^{d} I(Z_i \wedge \hat{Z}_i) &\ge d\left(h\left(\frac{k}{2d}\right) - \frac{1}{d}\sum_{i=1}^{d} h(\mathbb{P}[Z_i \ne \hat{Z}_i])\right) \\
&\ge d\left(h\left(\frac{k}{2d}\right) - h\left(\frac{1}{d}\sum_{i=1}^{d} \mathbb{P}[Z_i \ne \hat{Z}_i]\right)\right) \\
&\ge d\left(h\left(\frac{k}{2d}\right) - h\left(\frac{3k}{8d}\right)\right) \ge \frac{3}{100} k \log \frac{ek}{d}
\end{aligned}$$

where the last one is established by noting that $\inf_{x \in [0,1]} \frac{h(x/2) - h(3x/8)}{x \log(e/x)} > 0.03$. $\qquad \square$

In the following, we will present Lemma 24 to connect the relationship between $Z$ and data size.

**Lemma 24.** *Given* $0 < \epsilon < 1$, *under the above setting, for any* $\epsilon$*-NLDP algorithm* $\mathcal{A}$, *we have* $I\left(Z \wedge S^n\right) = O\left(\frac{n\nu^2 \epsilon^2}{d}\right)$.

*Proof of Lemma 24.* Since $S_1, \cdots, S_n$ are mutually independent conditionally on $Z$, this implies that

$$I\left(Z \wedge S^n\right) \le \sum_{i=1}^{n} I\left(Z_i \wedge S_i\right)$$

Thus, it is sufficient to bound each term $I\left(Z \wedge S_i\right) = O\left(\frac{\nu^2 \epsilon^2}{d}\right)$. Let fix any $1 \le t \le n$, denote $\mathbf{u}$ be any uniformly distribution over $\{-1, +1\}^{d+1}$. For the LDP algorithm for the $t$-th user, $A_t : \{-1, +1\}^{d+1} \to \{-1, +1\}^{d+1}$, let $A_t^{Pz}$ be the distribution on $\mathcal{A} := \{-1, +1\}^{d+1}$ induced by the input $V = (x_t, y_t)$ drawn from $P_{\theta_Z, \varsigma_Z}$ :

$$A_t^{Pz} = \underset{V \sim P_{\theta_Z, \varsigma_Z}}{\mathbb{E}} [A_t(a \mid V)], \quad a \in \mathcal{A}.$$

We also denote $A_t^{\mathbf{u}}$ as the distribution on $\mathcal{A} := \{-1, +1\}^{d+1}$ where $\mathbf{u}$ is any uniformly distribution over $\{+1, -1\}^{d+1}$:

$$A_t^{\mathbf{u}} = \underset{\mathbf{u} \sim \text{uniform}\{+1, -1\}^{d+1}}{\mathbb{E}} [A_t(a \mid \mathbf{u})], \quad a \in \mathcal{A}.$$

Hence the mutual information for each user $t$ could be formulated as:

$$I\left(Z \wedge S_t\right) = \mathbb{E}_Z \left[\text{KL}\left(A_t^{Pz} \| A_t^{\mathbf{u}}\right)\right] \le \mathbb{E}_Z \left[\chi^2\left(A_t^{Pz} \| A_t^{\mathbf{u}}\right)\right].$$

To simplify the notions and make the proof clear, we omit the subscript $t$ for $W_t$, $x_t$, and $y_t$ here. By the definition of Chi-square divergence and suppose $V = (x, y)$, $V' = (x', y')$ generated i.i.d from

the distribution $\mathbf{u}$, then:

$$
\mathbb{E}_Z \left[ \chi^2 \left( A_t^{P_Z} \| A_t^{\mathbf{u}} \right) \right] = \mathbb{E}_Z \left[ \sum_{a \in \mathcal{A}} \frac{\left( \sum_V A(a \mid V) \left( P_Z(V) - \mathbf{u}(V) \right) \right)^2}{\sum_V A(a \mid V) \mathbf{u}(V)} \right]
$$

$$
= \sum_{a \in \mathcal{A}} \mathbb{E}_Z \left[ \frac{\mathbb{E}_{\mathbf{u}} \left[ A(a \mid V) \left( y \langle x, \theta_Z \rangle \right) \right]^2}{\mathbb{E}_{\mathbf{u}}[A(a \mid V)]} \right]
$$

$$
= \sum_{a \in \mathcal{A}} \mathbb{E}_Z \left[ \frac{\mathbb{E}_{V,V' \sim \mathbf{u}} \left[ A(a \mid V) A \left( a \mid V' \right) \left( \sum_{i=1}^d \gamma^2 Z_i^2 x^i (x')^i \right) \right]}{\mathbb{E}_{\mathbf{u}}[A(a \mid V)]} \right]
$$

where the $x^i$ denotes the $i$-th coordinate of $x$.

We first focus on the last two terms of the molecule. Since $\mathbb{E}_Z[Z_i] = 0$ and $\mathbb{E}_Z \left[ Z_i^2 \right] = \frac{k}{2d} = \sigma^2$ for all $i \in [d]$, we could derive that:

$$
\mathbb{E}_Z \left[ \left( \sum_{i=1}^d \gamma^2 Z_i^2 x^i (x')^i \right) \right] = \sum_{i=1}^d \sigma^2 \gamma^2 x^i (x')^i
$$

Now we combine the above results, the following will hold:

$$
\mathbb{E}_Z \left[ \chi^2 \left( A_t^{P_Z} \| A_t^{\mathbf{u}} \right) \right] = \sum_{a \in \mathcal{A}} \frac{\mathbb{E}_{V,V' \sim \mathbf{u}} \left[ A(a \mid V) A \left( a \mid V' \right) \left( \sum_{i=1}^d \sigma^2 \gamma^2 x^i (x')^i \right) \right]}{\mathbb{E}_{\mathbf{u}}[A(a \mid V)]}
$$

$$
= \sigma^2 \gamma^2 \sum_{a \in \mathcal{A}} \sum_{i=1}^d \frac{\mathbb{E}_{\mathbf{u}} \left[ A(a \mid V) x^i \right]^2}{\mathbb{E}_{\mathbf{u}} \left[ A (a \mid V) \right]}
$$

$$
= \sigma^2 \gamma^2 \sum_{a \in \mathcal{A}} \sum_{i=1}^d \left( \frac{\left( \frac{1}{2} \mathbb{E}_{\mathbf{u}|y=1}[A(a \mid V) x^i] + \frac{1}{2} \mathbb{E}_{\mathbf{u}|y=-1}[A(a \mid V) x^i] \right)^2}{\frac{1}{2} \mathbb{E}_{\mathbf{u}|y=1} A(a \mid V) + \frac{1}{2} \mathbb{E}_{\mathbf{u}|y=-1} A(a \mid V)} \right) \quad (7)
$$

$$
\leq 2 \sigma^2 \gamma^2 \sum_{a \in \mathcal{A}} \sum_{i=1}^d \left( \frac{\mathbb{E}_{\mathbf{u}|y=1}[A(a \mid V) x^i]^2}{\mathbb{E}_{\mathbf{u}|y=1} A(a \mid V)} + \frac{\mathbb{E}_{\mathbf{u}|y=-1}[A(a \mid V) x^i]^2}{\mathbb{E}_{\mathbf{u}|y=-1} A(a \mid V)} \right)
$$

$$
= 2 \sigma^2 \gamma^2 \sum_{a \in \mathcal{A}} \left( \frac{\left\| \mathbb{E}_{\mathbf{u}|y=1}[A(a \mid V) x^i] \right\|^2}{\mathbb{E}_{\mathbf{u}|y=1} A(a \mid V)} + \frac{\left\| \mathbb{E}_{\mathbf{u}|y=-1}[A(a \mid V) x^i] \right\|^2}{\mathbb{E}_{\mathbf{u}|y=-1} A(a \mid V)} \right)
$$

We introduce the Lemma to bound the above equation:

**Lemma 25.** *(Acharya et al., 2022) Let $\phi_i : \mathbb{R}^d \to \mathbb{R}$, for $i \leq 1$, be a family of functions. If the functions satisfy, for all $i, j$,*

$$
\underset{V=(x,y) \sim uniform\{+1,-1\}^{d+1}}{\mathbb{E}} [\phi_i(x) \phi_j(x)] = \mathbf{1}_{\{i=j\}},
$$

*then, for any $\epsilon$-LDP algorithm $A$ and $V = (x, y)$, we have*

$$
\sum_i \mathbb{E}_V \left[ \phi_i(X) A(a \mid V) \right]^2 \leq Var_V[A(a \mid V)]
$$

$$
\sum_{a \in \mathcal{A}} \frac{k}{2d} \gamma^2 \sum_{i \in [d]} \frac{\mathbb{E}_{V \sim \mathbf{u}} \left[ A(a \mid V) x^i \right]^2}{\mathbb{E}_V[A(a \mid V)]} \leq \frac{k}{2d} \gamma^2 \sum_{a \in \mathcal{A}} \frac{Var_V[A(a \mid V)]}{\mathbb{E}_V[A(a \mid V)]}
$$

$$
\leq \frac{k}{2d} \gamma^2 \sum_{a \in \mathcal{A}} \frac{(e^\epsilon - 1)^2 \mathbb{E}_V[A(a \mid V)]^2}{\mathbb{E}_V[A(a \mid V)]}
$$

$$
= \frac{\nu^2}{2d} (e^\epsilon - 1)^2 .
$$

Hence, we could derive the conclusion of Lemma 24:

$$I\left(Z \wedge S^n\right) \leqslant \sum_{i=1}^{n} I\left(Z_i \wedge S_i\right) \leqslant \frac{n\nu^2\epsilon^2}{d}.$$

By combining the above lemmas, we could have the final results:

$$k \log(\frac{d}{k}) \leqslant \frac{n\nu^2\epsilon^2}{d},$$

which implies:

$$n \geqslant \Omega(\frac{dk \log(\frac{d}{k})}{\nu^2\epsilon^2}).$$

$\square$

$\square$

### E.2 Omitted Proofs in Section 3.2

**Proof of Theorem 2.** For each user $i$, it is easily to see that releasing $\widehat{\bar{x}_i \bar{x}_i^T}$ satisfies $(\frac{\epsilon}{2}, \frac{\delta}{2})$-DP, releasing $\widehat{\bar{x}_i \widetilde{y}_i}$ satisfies $(\frac{\epsilon}{2}, \frac{\delta}{2})$-DP. Thus, the algorithm is $(\epsilon, \delta)$-LDP. Moreover, we case see it is non-interactive. $\square$

**Proof of Theorem 3.** Before giving theoretical analysis, we first prove that $\dot{\Sigma}_{\bar{X}\bar{X}}$ is invertible with high probability.

By Algorithm 1, we can see that the noisy sample covariance matrix aggregated by the server can be represented as $\dot{\Sigma}_{\bar{X}\bar{X}} = \hat{\Sigma}_{\bar{X}\bar{X}} + N_1 = \frac{1}{n}\sum_i^n \bar{x}_i \bar{x}_i^T + N_1$, where $N_1$ is a symmetric Gaussian matrix with each entry sampled from $\mathcal{N}\left(0, \sigma_{N_1}^2\right)$ and $\sigma_{N_1}^2 = O\left(\frac{r^4 \log \frac{1}{\delta}}{n\epsilon^2}\right)$. We present the following lemma to start our analysis.

**Lemma 26.** *(Weyl's Inequality(Stewart & Sun, 1990)) Let $X, Y \in \mathbb{R}^{d \times d}$ be two symmetric matrices, and $E = X - Y$. Then, for all $i = 1, \cdots, d$, we have*

$$|\lambda_i(X) - \lambda_i(Y)| \leq \|E\|_2,$$

*where we take some liberties with the notation and use $\lambda_i(M)$ to denote the $i$-th eigenvalue of the matrix $M$.*

To show $\dot{\Sigma}_{\bar{X}\bar{X}}$ is invertible it is sufficient to show that $\|\dot{\Sigma}_{\bar{X}\bar{X}} - \Sigma\|_2 \leq \frac{\lambda_{\min}(\Sigma)}{2}$. This is due to that by Lemma 26, we have

$$\lambda_{\min}(\Sigma) - \|\dot{\Sigma}_{\bar{X}\bar{X}} - \Sigma\|_2 \leq \lambda_{\min}(\dot{\Sigma}_{\bar{X}\bar{X}}).$$

Thus, if $\|\dot{\Sigma}_{\bar{X}\bar{X}} - \Sigma\|_2 \leq \frac{\lambda_{\min}(\Sigma)}{2}$, we have $\lambda_{\min}(\dot{\Sigma}_{\bar{X}\bar{X}}) \geq \frac{\lambda_{\min}(\Sigma)}{2} > 0$.

In the following, we split the term $\|\dot{\Sigma}_{\bar{X}\bar{X}} - \Sigma\|_2$,

$$\|\dot{\Sigma}_{\bar{X}\bar{X}} - \Sigma\|_2 \leq \|N_1\|_2 + \|\hat{\Sigma}_{\bar{X}\bar{X}} - \Sigma_{\bar{X}\bar{X}}\|_2 + \|\Sigma_{\bar{X}\bar{X}} - \Sigma\|_2$$

where we denote $\hat{\Sigma}_{\bar{X}\bar{X}} = \frac{1}{n}\sum_i^n \bar{x}_i \bar{x}_i^T$ and $\Sigma_{\bar{X}\bar{X}} = \mathbb{E}[\bar{x}_i \bar{x}_i^T]$.

**Lemma 27** (Corollary 2.3.6 in Tao (2012)). *Let $M \in \mathbb{R}^d$ be a symmetric matrix whose entries $m_{ij}$ are independent for $j > i$, have mean zero, and are uniformly bounded in magnitude by 1. Then, there exists absolute constants $C_2, c_1 > 0$ such that with probability at least $1 - \exp\left(-C_2 c_1 d\right)$, the following inequality holds $\|M\|_2 \leq C\sqrt{d}$.*

By Lemma 27, we can see that with probability $1 - \exp(-\Omega(d))$,

$$\|N_1\|_2 \leq O\left(\frac{r^2\sqrt{d \log \frac{1}{\delta}}}{\sqrt{n}\epsilon}\right).$$

**Lemma 28.** *If $n \geq \tilde{\Omega}\left(d\|\Sigma\|_2\right)$, with probability at least $1 - \xi$*

$$\left\|\hat{\Sigma}_{\bar{X}\bar{X}} - \Sigma_{\bar{X}\bar{X}}\right\|_2 \leq O\left(\frac{\sqrt{d\|\Sigma\|_2 \log n \log \frac{d}{\xi}}}{\sqrt{n}}\right).$$

*Proof.* Note that $\left\|\bar{x}_i \bar{x}_i^T - \Sigma_{\bar{X}\bar{X}}\right\|_2 \leq \left\|\bar{x}_i \bar{x}_i^T\right\|_2 + \|\Sigma_{\bar{X}\bar{X}}\|_2 \leq 2r^2$. And for any unit vector $v \in \mathbb{R}^d$ we have the following if we denote $\bar{X} = \bar{x}_i \bar{x}_i^T$

$$\mathbb{E}\left(v^T \bar{X}^T \bar{X} v\right) = \mathbb{E}\left[\|\bar{x}_i\|_2^2 \left(v^T \bar{x}_i\right)^2\right] \leq O\left(r^4\right).$$

Thus we have $\left\|\mathbb{E}\left[\bar{X}^T \bar{X}\right]\right\|_2 \leq O\left(r^2\right)$. Since $\left\|\mathbb{E}(\bar{X})^T \mathbb{E}(\bar{X})\right\|_2 \leq \|\mathbb{E}(\bar{X})\|_2^2 \leq r^2$, we have $\left\|\mathbb{E}[\bar{X} - \mathbb{E}\bar{X}]^T \mathbb{E}[\bar{X} - \mathbb{E}\bar{X}]\right\|_2 \leq O\left(r^2\right)$. Thus, by the Non-communicative Bernstein inequality (Lemma 20) we have for some constant $c > 0$ :

$$\mathbb{P}\left(\left\|\hat{\Sigma}_{\bar{X}\bar{X}} - \Sigma_{\bar{X}\bar{X}}\right\|_2 > t\right) \leq 2d \exp\left(-c \min\left(\frac{nt^2}{r^2}, \frac{nt}{r^2}\right)\right).$$

Thus we have with probability at least $1 - \xi$ and the definition of $r$ we have,

$$\left\|\hat{\Sigma}_{\bar{X}\bar{X}} - \Sigma_{\bar{X}\bar{X}}\right\|_2 \leq O\left(\frac{\sqrt{d\|\Sigma\|_2 \log n \log \frac{d}{\xi}}}{\sqrt{n}}\right)$$

$\square$

We can also see that $\|\Sigma_{\bar{X}\bar{X}} - \Sigma\|_2 \leq O\left(\frac{\|\Sigma\|_2^2}{n}\right)$. This is due to

$$\|\Sigma_{\bar{X}\bar{X}} - \Sigma\|_2 \leq \left\|\mathbb{E}\left[\left(\bar{x}_i \bar{x}_i^T - x_i x_i^T\right) \mathbb{I}_{\|x\|_2 \geq r}\right]\right\|_2.$$

For any unit vector $v \in \mathbb{R}^p$ we have

$$v^T \mathbb{E}\left[\left(x_i x_i^T - \bar{x}_i \bar{x}_i^T\right) \mathbb{I}_{\|x\|_2 \geq r}\right] v = \mathbb{E}\left[\left(\left(v^T x_i\right)^2 - \left(v^T \bar{x}_i\right)^2\right) \mathbb{I}_{\|x_i\|_2 \geq r}\right]$$

$$\leq \mathbb{E}\left[\left(v^T x_i\right)^2 \mathbb{I}_{\|x\|_2 \geq r}\right] \leq \sqrt{\mathbb{E}\left[\left(v^T x_i\right)^4\right] \mathbb{P}\left[\|x_i\|_2 \geq r\right]} \leq O\left(\frac{\|\Sigma\|_2^2}{n}\right)$$

where the last inequality is due to the assumption on sub-Gaussian where $\mathbb{P}(|x_i| \geq r) \leq 2 \exp(-\frac{r^2}{2\sigma^2}) = O(\frac{1}{n^2})$.

Thus, it is sufficient to show that $\lambda_{\min}(\Sigma) \geq O\left(\frac{\|\Sigma\|_2^2 r^2 \sqrt{d \log \frac{d}{\xi} \log \frac{1}{\delta}}}{\sqrt{n}\epsilon}\right)$, which is true under the

assumption of $n \geq \Omega\left(\frac{\|\Sigma\|_2^4 dr^4 \log \frac{d}{\xi} \log \frac{1}{\delta}}{\epsilon^2 \lambda_{\min}^2(\Sigma)}\right)$. Thus, with probability at least $1 - \exp(-\Omega(d)) - \xi$, it is invertible. In the following we will always assume that this event holds.

To prove the theorem, we first introduce the following lemma on the estimation error of $\hat{\theta}$ in equation 2.

**Lemma 29** (Theorem 2 in Yang et al. (2014)). *Suppose we solve the problem of the form $\min_\theta \|\theta - \widehat{\theta}\|_2^2 + \lambda_n \|\theta\|_1$ such that constraint term $\lambda_n$ is set as $\lambda_n \geq \left\|\theta^* - \widehat{\theta}\right\|_\infty$. Then, the optimal solution $\hat{\theta} = S_{\lambda_n}(\widehat{\theta})$ satisfies:*

$$\left\|\hat{\theta} - \theta^*\right\|_\infty \leq 2\lambda_n,$$

$$\left\|\hat{\theta} - \theta^*\right\|_2 \leq 4\sqrt{k}\lambda_n,$$

$$\left\|\hat{\theta} - \theta^*\right\|_1 \leq 8k\lambda_n.$$

Note that this is a non-probabilistic result, and it holds deterministically for any selection of $\lambda_n$ or any distributional setting of the covariates $x_i$. Our goal is to show that $\lambda_n \geq \left\| \theta^* - \left( \dot{\Sigma}_{\bar{X}\bar{X}} \right)^{-1} \left( \dot{\Sigma}_{\widetilde{X}\widetilde{Y}} \right) \right\|_\infty$ under the assumptions specified in Lemma 29.

$$
\begin{aligned}
\left\| \theta^* - \hat{\theta}^{priv}(D) \right\|_\infty &= \left\| \theta^* - \left( \dot{\Sigma}_{\bar{X}\bar{X}} \right)^{-1} \left( \dot{\Sigma}_{\widetilde{X}\widetilde{Y}} \right) \right\|_\infty \\
&\leq \left\| \left( \dot{\Sigma}_{\bar{X}\bar{X}} \right)^{-1} \right\|_\infty \left\| \left( \dot{\Sigma}_{\bar{X}\bar{X}} \right) \theta^* - \left( \widehat{\Sigma}_{\widetilde{X}\widetilde{Y}} + N_2 \right) \right\|_\infty
\end{aligned}
\tag{8}
$$

where the vector $N_2 \in \mathbb{R}^d$ is sampled from $\mathcal{N}(0, \frac{32 d \tau_1^2 \tau_2^2 \log \frac{1.25}{\delta}}{\sqrt{n}\epsilon^2} I_d)$. We first develop upper bound of $\dot{\Sigma}_{\bar{X}\bar{X}}$. For any nonzero vector $w \in \mathbb{R}^d$, Note that

$$
\begin{aligned}
\left\| \dot{\Sigma}_{\bar{X}\bar{X}} w \right\|_\infty &= \left\| \dot{\Sigma}_{\bar{X}\bar{X}} w - \Sigma w + \Sigma w \right\|_\infty \\
&\geq \left\| \Sigma w \right\|_\infty - \left\| \left( \dot{\Sigma}_{\bar{X}\bar{X}} - \Sigma \right) w \right\|_\infty \\
&\geq \left( \kappa_\infty - \left\| \dot{\Sigma}_{\bar{X}\bar{X}} - \Sigma \right\|_\infty \right) \left\| w \right\|_\infty.
\end{aligned}
$$

Our objective is to find a sufficiently large $n$ such that $\left\| \dot{\Sigma}_{\bar{X}\bar{X}} - \Sigma \right\|_\infty$ is less than $\frac{\kappa_\infty}{2}$.

From above we see that, we have $\|N_1\|_2 < O(\sqrt{d}\sigma_{N_1}) = O(\frac{r^2 \sqrt{d \log \frac{1}{\delta}}}{\sqrt{n}\epsilon^2})$ by Lemma 27, which indicates the following holds:

$$
\begin{aligned}
\| \dot{\Sigma}_{\bar{X}\bar{X}} - \Sigma \|_2 &\leq \| \widehat{\Sigma}_{\bar{X}\bar{X}} - \Sigma \|_2 + \| N_1 \|_2 \\
&\leq O \left( \frac{r^2 \sqrt{d \log d \log \frac{1.25}{\delta}}}{\sqrt{n}\epsilon^2} \right),
\end{aligned}
$$

where the second inequality comes from Tropp et al. (2015). The following inequality always hold $\| \dot{\Sigma}_{\bar{X}\bar{X}} - \Sigma \|_\infty \leq \sqrt{d} \| \dot{\Sigma}_{\bar{X}\bar{X}} - \Sigma \|_2 \leq O(\frac{dr^2 \sqrt{\log d \log \frac{1.25}{\delta}}}{\sqrt{n}\epsilon^2})$. Thus, when $n \geq \Omega \left( \frac{d^2 r^4 \log d \log \frac{1}{\delta}}{\epsilon^2 \kappa_\infty} \right)$, we have $\left\| \dot{\Sigma}_{\bar{X}\bar{X}} w \right\|_\infty \geq \frac{\kappa_\infty}{2} \| w \|_\infty$, which implies $\left\| \left( \dot{\Sigma}_{\bar{X}\bar{X}} \right)^{-1} \right\|_\infty \leq \frac{2}{\kappa_\infty}$. Given sufficiently large $n$, from Eq.equation 8, we have:

$$
\begin{aligned}
&\left\| \theta^* - \hat{\theta}^{priv}(D) \right\|_\infty \\
\leq& \frac{2}{\kappa_\infty} \left\| \left( \dot{\Sigma}_{\bar{X}\bar{X}} \right) \theta^* - \left( \widehat{\Sigma}_{\widetilde{X}\widetilde{Y}} + N_2 \right) \right\|_\infty \\
\leq& \frac{2}{\kappa_\infty} \left\{ \underbrace{\left\| \widehat{\Sigma}_{\widetilde{X}\widetilde{Y}} - \Sigma_{\widetilde{X}\widetilde{Y}} \right\|_\infty}_{T_1} + \underbrace{\left\| \Sigma_{\widetilde{X}\widetilde{Y}} - \Sigma_{YX} \right\|_\infty}_{T_2} + \underbrace{\left\| \left( \dot{\Sigma}_{\bar{X}\bar{X}} - \Sigma \right) \theta^* \right\|_\infty}_{T_3} + \underbrace{\| N_2 \|_\infty}_{N_2} \right\}
\end{aligned}
\tag{9}
$$

We will bound the above four terms one by one.

We first consider term $T_1$. Since $x$ and $y$ are both $O(1)$-sub-Gaussian, we denote their $\psi_2$-norm by $\kappa_X$ and $\kappa_Y$, respectively. For $1 \leq j \leq d$, we have $\text{Var}(\tilde{y}_i \tilde{x}_{ij}) \leq \mathbb{E}[(\tilde{y}_i \tilde{x}_{ij})^2] \leq \mathbb{E}[(y_i x_{ij})^2] \leq (\mathbb{E}[|y_i|^2])(\mathbb{E}[|x_{ij}|^2]) \leq (\mathbb{E}[|y_i|^{\frac{2k}{k-1}}])^{\frac{k-1}{k}}(\mathbb{E}[|x_{ij}|^{\frac{2k}{k-1}}])^{\frac{k-1}{k}} \leq 4\kappa_X^2 \kappa_Y^2 (\frac{k}{k-1})^2 =: v_1 < \infty$. We have $v_1 = O(1)$. Therefore, according to Lemma 18, we have:

$$
\mathbb{P} \left( \left| \widehat{\sigma}_{\widetilde{Y}\widetilde{x}_j} - \sigma_{\widetilde{Y}\widetilde{x}_j} \right| \geq \sqrt{\frac{2v_1 t}{n}} + \frac{c\tau_1 \tau_2 t}{n} \right) \leq \exp(-t),
$$

where $\widehat{\sigma}_{\widetilde{Y}\widetilde{x}_j} = \frac{1}{n}\sum_{i=1}^{n}\tilde{y}_i\widetilde{x}_{ij}$, $\sigma_{\widetilde{Y}\widetilde{x}_j} = \mathbb{E}[\tilde{y}_i\widetilde{x}_{ij}]$ and $c$ is a certain constant. Then by the union bound, the following can be derived:

$$\mathbb{P}\left(|T_1| > \sqrt{\frac{2v_1 t}{n}} + \frac{c\tau_1\tau_2 t}{n}\right) \leq d\exp(-t).$$

Next, we give an estimation of $T_2$. Note that for $1 \leq j \leq d$, by lemma 12 we have:

$$\mathbb{E}\left[\tilde{y}_i\widetilde{x}_{ij}\right] - \mathbb{E}\left[y_i x_{ij}\right] = \mathbb{E}\left[\tilde{y}_i\widetilde{x}_{ij}\right] - \mathbb{E}\left[\tilde{y}_i x_{ij}\right] + \mathbb{E}\left[\tilde{y}_i x_{ij}\right] - \mathbb{E}\left[y_i x_{ij}\right]$$
$$= \mathbb{E}\left[\tilde{y}_i\left(\widetilde{x}_{ij} - x_{ij}\right)\right] + \mathbb{E}\left[\left(\tilde{y}_i - y_i\right)x_{ij}\right]$$
$$\leq \sqrt{\mathbb{E}\left[y_i^2\left(\widetilde{x}_{ij} - x_{ij}\right)^2\right]\mathbb{P}\left(|x_{ij}| \geq \tau_1\right)} + \sqrt{\mathbb{E}\left[\left(\tilde{y}_i - y_i\right)^2 x_{ij}^2\right]\mathbb{P}\left(|y_i| \geq \tau_2\right)}$$
$$\leq \sqrt{v_1}\left(2e^{-\frac{\tau_1^2}{2\sigma^2}} + 2e^{-\frac{\tau_2^2}{2\sigma^2}}\right),$$

which shows that $T_2 \leq \sqrt{v_1}\left(2e^{-\frac{\tau_1^2}{2\sigma^2}} + 2e^{-\frac{\tau_2^2}{2\sigma^2}}\right)$.

To upper bound term $T_3$, we need to evaluate $\|\dot{\Sigma}_{\bar{X}\bar{X}} - \Sigma\|_{\infty,\infty}$. It can be seen that $\dot{\Sigma}_{\bar{X}\bar{X}} = \hat{\Sigma}_{\bar{X}\bar{X}} + N_1 = \sum_i^n \bar{x}_i\bar{x}_i^T + N_1$. Therefore by Lemma 13 and Lemma 16 with probability at least $1 - Cd^{-8}$, for all $1 \leq i, j \leq d$, and for some constants $\gamma$ and $C$ that depends on $\sigma_{N_1}$,

$$\left|\dot{\sigma}_{\bar{x}\bar{x}^T,ij} - \sigma_{xx^T,ij}\right| \leq \gamma\sqrt{\frac{\log d}{n}} + \frac{128r^2\sqrt{2\log\frac{1.25}{\delta}\log d}}{\sqrt{n}\epsilon} \leq O\left(\gamma r^2\sqrt{\frac{\log d\log\frac{1}{\delta}}{n\epsilon^2}}\right). \tag{10}$$

We can see that $T_3$ is bounded by $O(d\log n\sqrt{(\frac{\log d\log\frac{1}{\delta}}{n\epsilon^2})})$. Here we used the fact that $\|(\dot{\Sigma}_{\bar{X}\bar{X}} - \Sigma)\theta^*\|_\infty \leq \|\dot{\Sigma}_{\bar{X}\bar{X}} - \Sigma\|_{\infty,\infty}\|\theta^*\|_1 \leq O(r^2\sqrt{(\frac{\log d\log\frac{1}{\delta}}{n\epsilon^2})})\|\theta^*\|_1$ given the selection of $r$, where the last inequality is from Eq.10.

The last term of Eq.equation 9 can be bounded by Gaussian tail bound by lemma 16. With probability $1 - O(d^{-8})$, we have:

$$\|N_2\|_\infty \leq O\left(\frac{\tau_1\tau_2\sqrt{d\log\frac{1}{\delta}\log d}}{\epsilon\sqrt{n}}\right). \tag{11}$$

Finally combining all pieces, we can find that $T_3$ is the dominating term. Since $\lambda_n \geq \left\|\theta^* - \left(\dot{\Sigma}_{\bar{X}\bar{X}}\right)^{-1}\left(\dot{\Sigma}_{\widetilde{X}\widetilde{Y}}\right)\right\|_\infty$, Lemma 29 implies that with probability at least $1 - O(d^{-8}) - e^{-\Omega(d)}$,

$$\left\|\theta^* - \left[\dot{\Sigma}_{\bar{X}\bar{X}}\right]^{-1}\left(\hat{\Sigma}_{\widetilde{X}\widetilde{Y}} + N_2\right)\right\|_2 \leq O\left(\frac{d\log n\sqrt{k\log d\log\frac{1}{\delta}}}{\sqrt{n}\epsilon}\right),$$

which completes our proof of Theorem. $\qquad\square$

**Proof of Theorem 4.** We basically follow the same ideas in the proof of Theorem 3.

First using similar argument, we can show that $(\hat{\Sigma}_{XX}^{pub})^{-1}$ exists with high probability.

The following lemma is the concentration result on sub-Gaussian matrix.

**Lemma 30.** *(Theorem 4.7.1 in Vershynin (2018) ) Let $x$ be a random vector in $\mathbb{R}^d$ that is sub-Gaussian with covariance matrix $\Sigma$ and $\left\|\Sigma^{-\frac{1}{2}}x\right\|_{\psi_2} \leq \kappa_x$. Then, with probability at least $1 - \exp(-d)$, the empirical covariance matrix $\hat{\Sigma}_{XX} = \frac{1}{n}\sum_{i=1}^{n}x_i x_i^T$ satisfies*

$$\left\|\hat{\Sigma}_{XX} - \Sigma\right\|_2 \leq C\kappa_x^2\sqrt{\frac{d}{n}}\|\Sigma\|_2$$

---

**Algorithm 3** Non-interactive LDP algorithm for Sparse Linear Regression with public but unlabeled data

---

1: **Input:** Private data $\{(x_i, y_i)\}_{i=1}^n \in \left(\mathbb{R}^d \times \mathbb{R}\right)^n$. Predefined parameters $\tau_1, \tau_2, \lambda_n$.
2: **for Each user** $i \in [n]$ **do**
3:     **for** $j \in [d]$ **do**
4:         Coordinately shrink $\widetilde{x}_{ij} = \text{sgn}\,(x_{ij}) \min\{|x_{ij}|, \tau_1\}$
5:     **end for**
6:     Clip $\tilde{y}_i := \text{sgn}\,(y_i) \min\{|y_i|, \tau_2\}$. Add noise $\widehat{\tilde{x}_i \tilde{y}_i} = \tilde{x}_i \tilde{y}_i + n_{2,i}$, where the vector $n_{2,i} \in \mathbb{R}^d$
    is sampled from $\mathcal{N}(0, \frac{2d\tau_1^2 \tau_2^2 \log \frac{1.25}{\delta}}{\epsilon^2} I_d)$. Release $\widehat{\tilde{x}_i \tilde{y}_i}$ to the server.
7: **end for**
8: The server aggregates $\dot{\Sigma}_{\widetilde{X}\widetilde{Y}} = \frac{1}{n} \sum_{i=1}^n \widehat{\tilde{x}_i \tilde{y}_i}$ and compute $\hat{\Sigma}_{XX}^{pub} = \frac{1}{m} \sum_{j=n+1}^{n+m} x_j x_j^T$ using dataset
9: The server outputs $\hat{\theta}^{priv}(D) = S_{\lambda_n}([\hat{\Sigma}_{XX}^{pub}]^{-1} \dot{\Sigma}_{\widetilde{X}\widetilde{Y}})$.

---

By Lemma 26 and Lemma 30, with probability at least $1 - \exp(-\Omega(d))$,

$$\lambda_{\min}\left(\hat{\Sigma}_{XX}^{pub}\right) \geq \lambda_{\min}(\Sigma) - O\left(\kappa_x^2 \|\Sigma\|_2 \sqrt{\frac{d}{m}}.\right)$$

We know that under the assumption of $m \geq \Omega\left(\frac{\kappa_x^4 \|\Sigma\|_2^2 d}{\lambda_{\min}^2(\Sigma)}\right)$, it is sufficient to show that $\lambda_{\min}(\Sigma) \geq O\left(\frac{\kappa_x^2 \|\Sigma\|_2 \sqrt{d}}{\sqrt{m}}\right)$. Thus, with probability at least $1 - \exp(-\Omega(d))$, it is invertible. In the following we will always assume that this event holds.

With the benefit of Lemma 29, we need to show that $\lambda_n \geq \|\theta^* - (\hat{\Sigma}_{XX}^{pub})^{-1}(\dot{\Sigma}_{\widetilde{X}\widetilde{Y}})\|_\infty$.

We know from the proof of Theorem 4 in Cai & Zhou (2012) that $\|\dot{\Sigma}_{\bar{X}\bar{X}} - \Sigma\|_\infty \leq \sqrt{d}\|\dot{\Sigma}_{\bar{X}\bar{X}} - \Sigma\|_2 \leq (\frac{4d\sqrt{2\log d}}{\sqrt{n}})$. Therefore when $n \geq \Omega(\frac{d^2 \log d}{\kappa_\infty})$, we have $\|\hat{\Sigma}_{XX}^{pub} w\|_\infty \geq \frac{\kappa_\infty}{2}\|w\|_\infty$, which implies $\|(\hat{\Sigma}_{XX}^{pub})^{-1}\|_\infty \leq \frac{2}{\kappa_\infty}$. Given this sufficiently large $n$, from Eq.equation 8, we have that

$$
\begin{aligned}
&\left\|\theta^* - \hat{\theta}^{priv}(D)\right\|_\infty \\
\leq &\left\|\left(\hat{\Sigma}_{XX}^{pub}\right)^{-1}\right\|_\infty \left\|\left(\dot{\Sigma}_{\bar{X}\bar{X}}\right)\theta^* - \left(\hat{\Sigma}_{\widetilde{X}\widetilde{Y}} + N_2\right)\right\|_\infty \\
\leq &\frac{2}{\kappa_\infty} \left\|\left(\hat{\Sigma}_{XX}^{pub}\right)\theta^* - \left(\hat{\Sigma}_{\widetilde{X}\widetilde{Y}} + N_2\right)\right\|_\infty \\
\leq &\frac{2}{\kappa_\infty} \left\{\underbrace{\left\|\hat{\Sigma}_{\widetilde{X}\widetilde{Y}} - \Sigma_{\widetilde{X}\widetilde{Y}}\right\|_\infty}_{T_1} + \underbrace{\left\|\Sigma_{\widetilde{X}\widetilde{Y}} - \Sigma_{YX}\right\|_\infty}_{T_2} + \underbrace{\left\|\left(\hat{\Sigma}_{XX}^{pub} - \Sigma\right)\theta^*\right\|_\infty}_{T_3} + \underbrace{\|N_2\|_\infty}_{N_2}\right\}
\end{aligned}
\tag{12}
$$

It is easy to see that only $T_3$ term is different from Equation 9. By Lemma 16 we can get the following, with probability at least $1 - O(d^{-8})$, for all $1 \leq i, j \leq d$, for some constant $\gamma$

$$\left|\hat{\sigma}_{xx^T,ij}^{pub} - \sigma_{xx^T,ij}\right| \leq O\left(\gamma \sqrt{\frac{\log d}{n}}\right)$$

By similar argument in the proof of Theorem 3, we have that $T_3$ is bounded by $O(\sqrt{(\frac{\log d}{n})})\|\theta^*\|_1$ given the selection of $r$.

Therefore taking $\tau_1 = \Theta(\sigma\sqrt{\log n}), \tau_2 = \Theta(\sigma\sqrt{\log n}), r = \Theta(\sigma\sqrt{d\log n}), \lambda_n = O(\frac{d\log n \sqrt{\log \frac{1}{\delta}}}{\sqrt{n}\epsilon})$, we can see that the dominating terms are $T_2$ and $N_2$ thus the result follows.

□

**Proof of Theorem 5.** We follow basically the same techniques as in the proof of Theorem 3. With the benefit of Lemma 29, we need to show that $\lambda_n \geq \left\| \theta^* - \left( \dot{\Sigma}_{\bar{X}\bar{X}} \right)^{-1} \left( \dot{\Sigma}_{\widetilde{X}\widetilde{Y}} \right) \right\|_\infty$. From the proof of Theorem 3, when $n \geq \Omega \left( \frac{d^2 r^4 \log d \log \frac{1}{\delta}}{\epsilon^2 \kappa_\infty} \right)$, we have $\left\| \dot{\Sigma}_{\bar{X}\bar{X}} w \right\|_\infty \geq \frac{\kappa_\infty}{2} \|w\|_\infty$, which implies $\left\| \left( \dot{\Sigma}_{\bar{X}\bar{X}} \right)^{-1} \right\|_\infty \leq \frac{2}{\kappa_\infty}$. Given this sufficiently large $n$, from Eq.equation 8, the following can be obtained:

$$
\begin{aligned}
&\left\| \theta^* - \hat{\theta}^{priv}(D) \right\|_\infty \\
\leq & \left\| \left( \dot{\Sigma}_{\bar{X}\bar{X}} \right)^{-1} \right\|_\infty \left\| \left( \dot{\Sigma}_{\bar{X}\bar{X}} \right) \theta^* - \left( \widehat{\Sigma}_{\widetilde{X}\widetilde{Y}} + N_2 \right) \right\|_\infty \\
\leq & \frac{2}{\kappa_\infty} \left\| \left( \dot{\Sigma}_{\bar{X}\bar{X}} \right) \theta^* - \left( \widehat{\Sigma}_{\widetilde{X}\widetilde{Y}} + N_2 \right) \right\|_\infty \\
\leq & \frac{2}{\kappa_\infty} \left\{ \underbrace{\left\| \widehat{\Sigma}_{\widetilde{X}\widetilde{Y}} - \Sigma_{\widetilde{X}\widetilde{Y}} \right\|_\infty}_{T_1} + \underbrace{\left\| \Sigma_{\widetilde{X}\widetilde{Y}} - \Sigma_{YX} \right\|_\infty}_{T_2} + \underbrace{\left\| \left( \dot{\Sigma}_{\bar{X}\bar{X}} - \Sigma \right) \theta^* \right\|_\infty}_{T_3} + \underbrace{\|N_2\|_\infty}_{N_2} \right\}
\end{aligned}
\tag{13}
$$

Since the new assumption is made on $y_i$, $T_3$ is not affected by this difference. Therefore, we only need to examine $T_1$ and $T_2$.

Using similar arguments, we can see that $T_1$ is bounded by $O(\frac{\tau_1 \tau_2}{n})$ and $N_2$ is still bouneded by $O(\frac{\tau_1 \tau_2 \sqrt{d \log \frac{1}{\delta} \log d}}{\epsilon \sqrt{n}})$ with high probability.

We bound the terms $\mathbb{E}\left[ y_i^2 \tilde{x}_{ij}^2 \right], \mathbb{E}\left[ \tilde{y}_i^2 x_{ij}^2 \right]$ by

$$
\max\{\mathbb{E}\left[ y_i^2 \tilde{x}_{ij}^2 \right], \mathbb{E}\left[ \tilde{y}_i^2 x_{ij}^2 \right]\} \leq \mathbb{E}\left[ y_i^2 x_{ij}^2 \right] \leq \left( \mathbb{E}\left[ y_i^{2p} \right] \right)^{\frac{1}{p}} \left( \mathbb{E}\left[ x_{ij} \right]^{\frac{2p}{p-1}} \right)^{\frac{p-1}{p}} \leq 2M^{\frac{1}{p}} \kappa_X^2 p/(p-1) < \infty
$$

which is a constant that we denote by $v$. Note that for $1 \leq j \leq d$, by lemma 12 and Markov's inequality, the following holds:

$$
\begin{aligned}
& \mathbb{E}\left[ \tilde{y}_i \tilde{x}_{ij} \right] - \mathbb{E}\left[ y_i x_{ij} \right] \\
=& \mathbb{E}\left[ \tilde{y}_i \tilde{x}_{ij} \right] - \mathbb{E}\left[ \tilde{y}_i x_{ij} \right] + \mathbb{E}\left[ \tilde{y}_i x_{ij} \right] - \mathbb{E}\left[ y_i x_{ij} \right] \\
=& \mathbb{E}[\tilde{y}_i \left( \tilde{x}_{ij} - x_{ij} \right)] + \mathbb{E}[(\tilde{y}_i - y_i) x_{ij}] \\
\leq& \sqrt{\mathbb{E}\left[ y_i^2 \left( \tilde{x}_{ij} - x_{ij} \right)^2 \right] \mathbb{P}\left( |x_{ij}| \geq \tau_1 \right)} + \sqrt{\mathbb{E}\left[ \left( \tilde{y}_i - y_i \right)^2 x_{ij}^2 \right] \mathbb{P}\left( |y_i| \geq \tau_2 \right)} \\
\leq& \sqrt{v} \left( 2e^{-\frac{\tau_1^2}{2\sigma^2}} + \sqrt{\frac{\mathbb{E}[|y_i|]^{2p}}{\tau_2^{2p}}} \right) \\
\leq& \sqrt{v} \left( 2e^{-\frac{\tau_1^2}{2\sigma^2}} + \frac{\sqrt{M}}{\tau_2^p} \right),
\end{aligned}
$$

which shows that $T_2 \leq \sqrt{v} \left( 2e^{-\frac{\tau_1^2}{2\sigma^2}} + \frac{2\sqrt{M}}{\tau_2^{2p}} \right)$.

Taking $\tau_2 = \left( \frac{n\epsilon^2}{\log d} \right)^{\frac{1}{2p}}$ completes the proof.

$\square$

### E.3 OMITTED PROOFS IN SECTION 4.1

Before presenting the full proof of Theorem 6, we first introduce several necessary definitions and assumptions.

**Definition 7.** *For distributions $P_1, P_2$ over sample space $\mathcal{X}$, denote their Kullback-Leibler divergence (in nats) by* $\text{KL}\left(P_1\|P_2\right)$, *and their Hellinger distance by*

$$d_{\text{H}}\left(P_1, P_2\right) := \sqrt{\frac{1}{2}\int\left(\sqrt{\frac{dP_1}{d\lambda}} - \sqrt{\frac{dP_2}{d\lambda}}\right)^2 d\lambda}$$

**Definition 8.** *Let $Z = (Z_1, \ldots, Z_d)$ be a random variable over $\mathcal{Z} = \{-1, +1\}^d$ such that $\mathbb{P}\left[Z_i = 1\right] = \tau$ for all $i \in [d]$ and the $Z_i$ s are all independent; we denote this distribution by $\text{Rad}(\tau)^{\otimes d}$. For $z \in \mathcal{Z}$, we denote $z^{\oplus i} \in \mathcal{Z}$ as the vector obtained by flipping the sign of the $i$-th coordinate of $z$.*

**Assumption 4** (Densities Exist). *For every $z \in \mathcal{Z}$ and $i \in [d]$ it holds that $P_{z^{\oplus i}} \ll P_z$ (we refer to $P_{\theta_z}$ simply as $Pz$), and there exist measurable functions $\phi_{z,i} : \mathbb{R}^d \to \mathbb{R}$ such that*

$$\frac{dP_{z^{\oplus i}}}{dP_z} = 1 + \phi_{z,i}.$$

**Assumption 5** (Orthogonality). *There exists some $\alpha^2 \geq 0$ such that, for all $z \in \mathcal{Z}$ and distinct $i, j \in [d]$, $\mathbb{E}_{P_z}\left[\phi_{z,i} \cdot \phi_{z,j}\right] = 0$ and $\mathbb{E}_{P_z}\left[\phi_{z,i}^2\right] \leq \alpha^2$.*

**Assumption 6** (Additive loss). *For every $z, z' \in \mathcal{Z} = \{-1, +1\}^d$,*

$$\ell_2\left(\theta_z, \theta_{z'}\right) = 4\nu\left(\frac{d_{\text{Ham}}\left(z, z'\right)}{\tau d}\right)^{1/2}$$

*where $d_{\text{Ham}}\left(z, z'\right) := \sum_{i=1}^d \mathbb{1}\left\{z_i \neq z'_i\right\}$ denotes the Hamming distance, where $\tau = k/2d$, $k$ and $\nu$ denotes sparsity and error rate respectively.*

**Proof of Theorem 6.** Similar to the non-interactive setting, we consider the hard distribution class $\mathcal{P}_{k,d,2}$, where for each instance $P_{\theta,\zeta} \in \mathcal{P}_{k,d,2}$, its random noise $\zeta$ satisfies $\mathbb{E}[\zeta \mid x] = 0, |\zeta| \leq 2$, and $\|\theta\|_1 \leq 1$ and $\|\theta\|_0 \leq k$ hold. By setting $\gamma = \frac{4\sqrt{2}\nu}{\sqrt{k}}$ and defining $\theta_{z,i} = \frac{\gamma(z_i+1)}{2}$ for $i \in [d]$, $\{z_i\}_{i=1}^d$ are realizations of the random variable $Z \in \{-1, +1\}^d$, where each coordinate $Z_i$ is independent to others and has the following distribution:

$$\mathbb{P}\left\{Z_i = +1\right\} = k/2d, \quad \mathbb{P}\left\{Z_i = -1\right] = 1 - k/2d$$

We will first show that $\theta_z$ satisfies the conditions that $\|\theta\|_1$ and $\|\theta\|_0 \leq k$ and $\theta_z$ is $k$-sparse with probability of $1 - \tau$, where $\tau = k/2d$ by the following fact.

**Fact 1.** *(Acharya et al., 2020) For $Z \sim \text{Rad}(\tau)$ and $\tau d \geq 4 \log d$, then we have $\mathbb{P}\left(\|Z\|_+ \leq 2\tau d\right) \geq 1 - \tau/4$, where $\|Z\|_+ = \{i \in \{d\}|z_i = 1\}$.*

When $\theta_z$ is $k$-sparse, we can also see $\|\theta_z\|_1 \leq 4\sqrt{2k}\nu \leq 1$ as we assume $\nu \leq \frac{1}{4\sqrt{2k}}$.

Next, we construct the following generative process, we first pick $z$ randomly from $\{-1, +1\}^d$ as above. For each $Z$ we let:

$$\zeta_z = \begin{cases} 1 - \langle x, \theta_z \rangle & \text{w.p.} \quad 1 + \frac{\langle x, \theta_z \rangle}{2} \\ -1 - \langle x, \theta_z \rangle & \text{w.p.} \quad 1 - \frac{\langle x, \theta_z \rangle}{2} \end{cases}$$

Note that since $|\langle x, \theta_z \rangle| \leq \sqrt{k} \cdot \gamma = 4\sqrt{2}\nu \leq 1$. The above distribution is well-defined and $|\zeta_z| \leq 2$. We can see that density function for $(x, y)$ is $P_z = P_{\theta_z, \zeta_z}((x, y)) = \frac{1+y\langle x, \theta_z \rangle}{2^{d+1}}$ for $(x, y) \in \{-1, +1\}^{d+1}$.

In the subsequent, we will confirm that $P_z$ under the above constructions could satisfy the Assumptions 4, 5.

$$\frac{dP_{z^{\oplus i}}}{dP_z} = \frac{1 + y\langle x, \theta_{z^{\oplus i}}\rangle}{1 + y\langle x, \theta_z\rangle} = 1 + \frac{y\langle x, \theta_{z^{\oplus i}} - \theta_z\rangle}{1 + y\langle x, \theta_z\rangle}$$

$$\theta_{z^{\oplus i}} - \theta_z = \left(0, \cdots 0, \frac{\gamma(-z_i+1)}{2} - \frac{\gamma(z_i+1)}{2}, 0, \cdots 0\right)$$

$$= (0, \cdots 0, -\gamma z_i, 0, \cdots 0)$$

Thus, we could simplify the previous formulations as:

$$\frac{dP_{z\oplus i}}{dP_z} = 1 - \frac{y\gamma x_i z_i}{1 + y\langle x, \theta_z\rangle}$$

Let $\alpha_{z,i} = \frac{r}{1+y\langle x,\theta_z\rangle}$. Since $|y\langle x,\theta_z\rangle| \leqslant \sqrt{k}\|x\|_\infty\gamma \leqslant 1/2$, we have $\alpha_{z_i} \leqslant \frac{4\sqrt{2}\nu}{2\sqrt{k}}$, where the right hand side is denoted as $\alpha$.

Now there exists a measurable function $\phi_{z,i} = yx_i z_i$, with $\mathbb{E}[\phi_{z,i} \cdot \phi_{z,j}] = 0$ if $i \neq j$ and $\mathbb{E}[yx_i^2 z_i^2] = 1$ if $i \neq j$, indicating that assumptions 4 and 5 hold.

**Lemma 31** ((Acharya et al., 2020)). *For a $\epsilon$-sequentially interactive LDP algorithm $\mathcal{A}$ and any family of distributions $\{P_z = P_{\theta_z,\zeta_z}((x,y))\}$ satisfying Assumptions 4 and 5, let $Z$ be a random variable on $\mathcal{Z} = \{-1, +1\}^d$ with distribution $\mathrm{Rad}(\tau)^{\otimes d}$. Let $S^n$ be the tuple of messages from the algorithm $\mathcal{A}$ when the input $V_1 = (x_1, y_1), \ldots, V_n = (x_n, y_n)$ is i.i.d. with common distribution $P_{\theta_z,\zeta_z}$, then we have:*

$$\left(\frac{1}{d}\sum_{i=1}^d \mathrm{d_{TV}}\left(P_{+i}^{S^n}, P_{-i}^{S^n}\right)\right)^2 \leq \frac{7}{d}n\alpha^2\left((e^\varrho - 1)^2 \wedge e^\varrho\right) \tag{14}$$

*where $P_{+i}^{S^n} := \mathbb{E}\left[P_{\theta_z,\zeta_z}^{S^n} \mid Z_i = +1\right], P_{-i}^{S^n} := \mathbb{E}\left[P_{\theta_z,\zeta_z}^{S^n} \mid Z_i = -1\right]$. In Eq.equation 14, the left-hand side is defined as the average discrepancy, which represents the average amount of information that the transcript conveys about each coordinate of $Z$.*

With the assumptions holding and introducing the Lemma31, we have the following conclusions for our LDP algorithm and constructed distribution: $\left(\frac{1}{d}\sum_{i=1}^d \mathrm{d_{TV}}\left(P_{+i}^{S^n}, P_{-i}^{S^n}\right)\right)^2 \leq \frac{7}{d}n\alpha^2\epsilon^2$, where $\alpha = \frac{2\sqrt{2}\nu}{\sqrt{k}}$. Next, we will verify the remaining assumptions that should be satisfied for the lower bound of $\left(\frac{1}{d}\sum_{i=1}^d \mathrm{d_{TV}}\left(P_{+i}^{S^n}, P_{-i}^{S^n}\right)\right)^2$. Since

$$\|\theta_z - \theta_{z'}\| = \sqrt{\frac{32\nu^2}{k}\sum_{i=1}^d \mathbb{1}\{Z_i \neq \hat{Z}_i\}} = 4\nu\left(\frac{d_{\mathrm{Ham}(z,\hat{z})}}{\tau d}\right)^{1/2},$$

thus the assumption 6 holds for any $z, z' \in \{-1, +1\}^d$ with $\tau = k/2d$.

**Lemma 32** ((Acharya et al., 2020)). *Assume that $P_{\theta_z,\zeta_z}((x,y))$ satisfy Assumption 6, and $\tau = k/2d \in [0, 1/2]$. Let $Z$ be a random variable on $\mathcal{Z} = \{-1, +1\}^d$ with distribution $\mathrm{Rad}(\tau)^{\otimes d}$. if algorithm $\mathcal{A}$ is an $\epsilon$- sequentially interactive LDP algorithm such that, for any $n$-size dataset $\mathcal{D} = \{(x_i, y_i)\}_{i=1}^n$ consisting of i.i.d. samples from $P_{\theta_Z,\zeta_Z}$ with the probability $\mathbb{P}_Z[P_{\theta_z,\zeta_z} \in \mathcal{P}_{k,d,2}] \geq 1 - \tau/4$, and its output $\theta^{priv}$ satisfies $\mathbb{E}[\|\theta^{priv} - \theta^*\|_2] \leq \nu$, then the tuple of messages $S^n$ from the algorithm $\mathcal{A}$ satisfies*

$$\frac{1}{d}\sum_{i=1}^d \mathrm{d_{TV}}\left(P_{+i}^{S^n}, P_{-i}^{S^n}\right) \geq \frac{1}{4},$$

With the fact of $\theta_z$ is $k$-sparse w.p. $1 - \tau/4$ and assumption 6 holding, it is easy to obtain that lemma 32 is applicable for our sequentially interactive LDP algorithm.

Combining the above results, we have:

$$\frac{1}{4} \leq \frac{1}{d}\sum_{i=1}^d \mathrm{d_{TV}}\left(P_{+i}^{S^n}, P_{-i}^{S^n}\right) \leq \frac{7n\nu^2\epsilon^2}{dk}$$

which implies $n \geq O(\frac{dk}{\nu^2\epsilon^2})$.

$\square$

### E.4 OMITTED PROOFS IN SECTION 4

**Proof of Theorem 7.** We first show the guarantee of $\epsilon$-LDP. First, we will show that $\left\|\tilde{x}_i^T\left((\langle\tilde{x}_i,\theta_{t-1}\rangle)-\tilde{y}_i\right)\right\|_2 \leq \sqrt{d}\tau_1(\sqrt{k'}\tau_1+\tau_2)$, this is due to that

$$\left\|\tilde{x}_i^T\left(\langle\tilde{x}_i,\theta_{t-1}\rangle-\tilde{y}_i\right)\right\|_2$$
$$\leq \left\|\tilde{x}_i^T\right\|_2 |\langle\tilde{x}_i,\theta_{t-1}\rangle-\tilde{y}_i|)$$
$$\leq \left\|\tilde{x}_i^T\right\|_2 (|\sqrt{k}\|\tilde{x}_i\|_\infty\|\theta_{t-1}\|_2-\tilde{y}_i|) \leq \quad \sqrt{d}\tau_1(\sqrt{k'}\tau_1+\tau_2),$$

where the last inequality is due to that $\theta_{t-1}$ is $k'$-sparse, $\|\theta_{t-1}\|_2 \leq 1$ and each $\|\tilde{x}_i\|_\infty \leq \tau_1$.

Based on this and Lemma 8, we can easily see Algorithm 2 is $\epsilon$-LDP. Due to the partition of the dataset, we can see it is sequentially interactive.

Next, we consider the utility. Without loss of generality, we assume each $|S_t| = m = \frac{n}{T}$. From the randomizer $\mathcal{R}_\epsilon^r(\cdot)$ and Lemma 8 , we can see that $\tilde{\nabla}_t = \frac{1}{m}\sum_{i\in S_t}\tilde{x}_i^T\left(\langle\tilde{x}_i,\theta_{t-1}\rangle-\tilde{y}_i\right)+\phi_t$, where each coordinate of $\phi_t$ is a sub-Gaussian vector with variance $= O\left(\frac{d\tau_1^2(k'\tau_1^2+\tau_2^2)}{m\epsilon^2}\right)$.

Let $\mathcal{S}^* = \text{supp}(\theta^*)$ denote the support of $\theta^*$, and $k = |\mathcal{S}^*|$. Similarly, we define $\mathcal{S}^t = \text{supp}(\theta_t)$, and $\mathcal{F}^{t-1} = \mathcal{S}^{t-1}\cup\mathcal{S}^t\cup\mathcal{S}^*$. Thus, we have $\left|\mathcal{F}^{t-1}\right| \leq 2k'+k$. Let $\tilde{\theta}_{t-\frac{1}{2}}$ denote as the following:

$$\tilde{\theta}_{t-\frac{1}{2}} = \theta_{t-1} - \eta\tilde{\nabla}_{t-1,\mathcal{F}^{t-1}},$$

where $v_{\mathcal{F}^{t-1}}$ means keeping $v_i$ for $i\in\mathcal{F}^{t-1}$ and converting all other terms to 0 . By the definition of $\mathcal{F}^{t-1}$, we have $\theta_t' = \text{Trunc}\left(\tilde{\theta}_{t-\frac{1}{2}},k'\right)$.

For each iteration $t$, we also denote $\tilde{\nabla}L_{t-1}(\theta_{t-1}) = \frac{1}{m}\sum_{i\in S_t}\tilde{x}_i\left(\langle\tilde{x}_i,\theta_{t-1}\rangle-\tilde{y}_i\right)$, $\nabla L_{t-1}(\theta_{t-1}) = \frac{1}{m}\sum_{i\in S_t}x_i\left(\langle x_i,\theta_{t-1}\rangle-y_i\right)$, and $\nabla L_\mathcal{P}(\theta_{t-1}) = \mathbb{E}[x(\langle x,\theta_{t-1}\rangle-y)]$.

Denote by $\Delta_t$ the difference of $\theta_t - \theta^*$. We have the following:

$$\left\|\tilde{\theta}_{t-\frac{1}{2}}-\theta^*\right\|_2 = \|\Delta_{t-1}-\eta\tilde{\nabla}_t\|_2 = \|\Delta_{t-1}-\eta\tilde{\nabla}_{t,\mathcal{F}^{t-1}}\|_2$$
$$\leq \underbrace{\|\Delta_{t-1}-\eta[\nabla L_{t-1}(\theta_{t-1})]_{\mathcal{F}^{t-1}}\|_2}_{A} + \eta\underbrace{\|\tilde{\nabla}_{t,\mathcal{F}^{t-1}}-[\nabla L_{t-1}(\theta_{t-1})]_{\mathcal{F}^{t-1}}\|_2}_{B}.$$

We first bound the term $B$. Specifically, we have

$$\|\tilde{\nabla}_{t,\mathcal{F}^{t-1}}-[\nabla L_{t-1}(\theta_{t-1})]_{\mathcal{F}^{t-1}}\|_2 = \|[\tilde{\nabla}_t-\nabla L_{t-1}(\theta_{t-1})]_{\mathcal{F}^{t-1}}\|_2$$
$$\leq \sqrt{|\mathcal{F}^{t-1}|}\|\tilde{\nabla}_t-\nabla L_{t-1}(\theta_{t-1})\|_\infty$$
$$\leq \sqrt{|\mathcal{F}^{t-1}|}(\underbrace{\|\tilde{\nabla}L_{t-1}(\theta_{t-1})-\nabla L_{t-1}(\theta_{t-1})\|_\infty}_{B_1}+\underbrace{\|\phi_t\|_\infty}_{B_2})$$

For term $B_2$, by Lemma 15 we have with probability at least $1-\delta'$

$$B_2 \leq O\left(\frac{(\tau_1^2\sqrt{dk'}+\tau_1\tau_2\sqrt{d})\sqrt{\log\frac{d}{\delta'}}}{\sqrt{m}\epsilon}\right). \tag{15}$$

For $B_1$, we have

$$B_1 \leq \underbrace{\sup_{\|\theta\|_2\leq 1}\|\tilde{\nabla}L_{t-1}(\theta)-\nabla L_\mathcal{P}(\theta)\|_\infty}_{B_{1,1}} + \underbrace{\sup_{\|\theta\|_2\leq 1}\|\nabla L_{t-1}(\theta)-\nabla L_\mathcal{P}(\theta)\|_\infty}_{B_{1,2}}.$$

Next we bound the term $B_{1,1}$, we have

$$\sup_{\|\theta\|_1\leq 1}\|\tilde{\nabla}L_{t-1}(\theta)-\nabla L_\mathcal{P}(\theta)\|_\infty \leq \sup_{\|\theta\|_1\leq 1}\|[\frac{1}{m}\sum_{i=1}^n\tilde{x}_i\tilde{x}_i^T-\mathbb{E}[xx^T]]\theta\|_\infty + \sup_{\|\theta\|_1\leq 1}\|\frac{1}{m}\sum_{i=1}^m\tilde{x}_i\tilde{y}_i-\mathbb{E}[xy]\|_\infty$$

$$\leq \|[\frac{1}{m}\sum_{i=1}^{n}\tilde{x}_i\tilde{x}_i^T - \mathbb{E}[xx^T]]\|_{\infty,\infty} + \|\frac{1}{m}\sum_{i=1}^{m}\tilde{x}_i\tilde{y}_i - \mathbb{E}[xy]\|_\infty.$$

We consider the first term $\|[\frac{1}{m}\sum_{i=1}^{n}\tilde{x}_i\tilde{x}_i^T - \mathbb{E}[xx^T]]\|_{\infty,\infty}$, for simplicity for each $j,k \in [d]$ denote $\hat{\sigma}_{jk} = (\frac{1}{n}\sum_{i=1}^{n}\tilde{x}_i\tilde{x}_i^T)_{jk} = \frac{1}{n}\sum_{i=1}^{n}\tilde{x}_{i,j}\tilde{x}_{i,k}$, $\tilde{\sigma}_{jk} = (\mathbb{E}[\tilde{x}\tilde{x}^T])_{jk} = \mathbb{E}[\tilde{x}_j\tilde{x}_k]$ and $\sigma_{jk} = (\mathbb{E}[xx^T])_{jk} = \mathbb{E}[x_jx_k]$. We have

$$|\hat{\sigma}_{jk} - \sigma_{jk}| \leq |\hat{\sigma}_{jk} - \tilde{\sigma}_{jk}| + |\tilde{\sigma}_{jk} - \sigma_{jk}|.$$

We know that $|\tilde{x}_j\tilde{x}_k| \leq \tau_1^2$ and $\mathrm{Var}(\tilde{x}_j\tilde{x}_k) \leq \mathrm{Var}(x_jx_k) \leq \mathbb{E}(x_jx_k)^2 \leq O(\sigma^4)$. By Bernstein's inequality we have

$$\mathbb{P}(\max_{j,k}|\hat{\sigma}_{jk} - \tilde{\sigma}_{jk}| \leq C\sqrt{\frac{\sigma^4 t}{m}} + \frac{\tau_1^2 t}{m}) \geq 1 - d^2\exp(-t) \qquad (16)$$

Moreover, we have

$$|\tilde{\sigma}_{jk} - \sigma_{jk}| = |\mathbb{E}[|\tilde{x}_j(\tilde{x}_k - x_k)\mathbb{I}(|x_k| \geq \tau_1)] + |\mathbb{E}[|x_k(\tilde{x}_j - x_j)\mathbb{I}(|x_j| \geq \tau_1)]$$
$$\leq \sqrt{\mathbb{E}(\tilde{x}_j(\tilde{x}_k - x_k))^2\mathbb{P}(|x_k| \geq \tau_1)} + \sqrt{\mathbb{E}((\tilde{x}_j - x_j)x_k)^2\mathbb{P}(|x_j| \geq \tau_1)}$$
$$\leq O(\frac{\sigma^2}{n}),$$

where the last inequality is due to the assumption on sub-Gaussian where $\mathbb{P}(|x_j| \geq \tau_1) \leq 2\exp(-\frac{\tau_1^2}{2\sigma^2}) = O(\frac{1}{n})$, $\mathbb{E}(\tilde{x}_j(\tilde{x}_k - x_k))^2 \leq 4\mathbb{E}(x_jx_k))^2 \leq O(\sigma^4)$ and $\mathbb{E}((\tilde{x}_j - x_j)x_k)^2 \leq 4\mathbb{E}(x_jx_k))^2 \leq O(\sigma^4)$. In total we have with probability at least $1 - \delta'$

$$\|[\frac{1}{m}\sum_{i=1}^{n}\tilde{x}_i\tilde{x}_i^T - \mathbb{E}[xx^T]]\|_{\infty,\infty} \leq O(\frac{\sigma^2\log n\log\frac{d}{\delta'}}{\sqrt{m}}).$$

We can use the same technique to term $\|\frac{1}{m}\sum_{i=1}^{m}\tilde{x}_i\tilde{y}_i - \mathbb{E}[xy]\|_\infty$, for simplicity for each $j \in [d]$ denote $\hat{\sigma}_j = \frac{1}{n}\sum_{i=1}^{n}\tilde{y}_i\tilde{x}_j$, $\tilde{\sigma}_j = \mathbb{E}[\tilde{y}\tilde{x}_j]$ and $\sigma_j = \mathbb{E}[yx_j]$. We have

$$|\hat{\sigma}_j - \sigma_j| \leq |\hat{\sigma}_j - \tilde{\sigma}_j| + |\tilde{\sigma}_j - \sigma_j|.$$

Since $|\tilde{x}_j\tilde{y}| \leq \tau_1\tau_2$ and we have the following by the Holder's inequality

$$\mathrm{Var}(\tilde{x}_j\tilde{y}) \leq \mathrm{Var}(x_jy) \leq \mathbb{E}[x_j^2 y^2] \leq (\mathbb{E}[y^4])^{\frac{1}{2}}(\mathbb{E}[|x_j|^4])^{\frac{1}{2}} \leq O(\sigma^4)$$

Thus, by Bernstein's inequality we have for all $j \in [d]$

$$\mathbb{P}(|\hat{\sigma}_j - \tilde{\sigma}_j| \leq O(\sqrt{\frac{\sigma^4 t}{m}} + \frac{\tau_1\tau_2 t}{m})) \geq 1 - d\exp(-t).$$

Moreover

$$|\tilde{\sigma}_j - \sigma_j| \leq |\mathbb{E}[\tilde{y}(\tilde{x}_j - x_j)\mathbb{I}(|x_j|) \geq \tau_1]| + |\mathbb{E}[x_j(\tilde{y} - y)\mathbb{I}(|y| \geq \tau_2)]|$$
$$\leq \sqrt{\mathbb{E}((\tilde{y}(\tilde{x}_j - x_j))^2\mathbb{P}(|x_j| \geq \tau_1)} + \sqrt{\mathbb{E}(x_j(\tilde{y} - y))^2\mathbb{P}(|y| \geq \tau_2)}$$
$$\leq O(\frac{\sigma^2}{n} + \frac{\sigma^2}{n}) \leq O(\frac{\sigma^2}{n})$$

we can easily see that with probability at most $1 - \delta'$,

$$\|\frac{1}{m}\sum_{i=1}^{m}\tilde{x}_i\tilde{y}_i - \mathbb{E}[xy]\|_\infty \leq O(\frac{\sigma^2\log n\log\frac{d}{\delta'}}{\sqrt{m}}). \qquad (17)$$

Thus with probability at least $1 - \delta'$

$$B_{1,1} \leq O(\frac{\sigma^2\log n\log\frac{d}{\delta'}}{\sqrt{m}}). \qquad (18)$$

Next, we consider $B_{1,2}$, similar to $B_{1,1}$ we have

$$\sup_{\|\theta\|_2 \leq 1} \|\nabla L_{t-1}(\theta) - \nabla L_{\mathcal{P}}(\theta)\|_\infty \leq \|[\frac{1}{m}\sum_{i=1}^n x_i x_i^T - \mathbb{E}[xx^T]]\|_{\infty,\infty} + \|\frac{1}{m}\sum_{i=1}^m x_i y_i - \mathbb{E}[xy]\|_\infty.$$

For term $\|[\frac{1}{m}\sum_{i=1}^n x_i x_i^T - \mathbb{E}[xx^T]]\|_{\infty,\infty}$, by Lemma 13 we have with probability at least $1 - O(d^{-8})$ we have

$$\|[\frac{1}{m}\sum_{i=1}^n x_i x_i^T - \mathbb{E}[xx^T]]\|_{\infty,\infty} \leq O(\sqrt{\frac{\log d}{m}}).$$

For term $\|\frac{1}{m}\sum_{i=1}^m x_i y_i - \mathbb{E}[xy]\|_\infty$, we consider each coordinate, $\frac{1}{m}\sum_{i=1}^m x_{i,j} y_i - \mathbb{E}[x_j y]$. Noted that $x_j$ is $\sigma^2$-sub-Gaussian and $y$ is $\sigma^2$-sub-Gaussian, thus, by Lemma 10 we have $x_j y$ is sub-exponential with $\|x_j y\|_{\psi_1} \leq O(\sigma^2)$. Thus, by Bernstein's inequality, we have with probability at least $1 - \zeta'$

$$|\frac{1}{m}\sum_{i=1}^m x_{i,j} y_i - \mathbb{E}[x_j y]| \leq O(\frac{\sigma^2 \sqrt{\log 1/\delta'}}{\sqrt{m}}).$$

Thus, with probability at least $1 - \zeta'$

$$\|\frac{1}{m}\sum_{i=1}^m x_i y_i - \mathbb{E}[xy]\|_\infty \leq O(\frac{\sigma^2 \sqrt{\log d/\delta'}}{\sqrt{m}}).$$

Thus, with probability at least $1 - O(d^{-8})$ we have

$$B_{1,2} \leq O(\frac{\sqrt{\log d}}{\sqrt{m}}).$$

and

$$B_1 \leq O(\frac{\sqrt{\log d}}{\sqrt{m}}).$$

Thus, we have

$$B \leq O\left(\sqrt{2k' + k}\frac{(\tau_1^2 \sqrt{dk'} + \tau_1 \tau_2 \sqrt{d})\sqrt{\log \frac{d}{\delta'}}}{\sqrt{m}\epsilon}\right). \tag{19}$$

In the following, we consider term $A$. Noted that we have $y_i = \langle x_i, \theta^* \rangle + \zeta_i$, thus, we have

$$\underbrace{\|\Delta_{t-1} - \eta[\nabla L_{t-1}(\theta_{t-1})]_{\mathcal{F}^{t-1}}\|_2}_{A} \leq \|\Delta_{t-1} - \eta[\frac{1}{m}\sum_{i=1}^m (x_i(\langle x_i, \theta_{t-1} - \theta^* \rangle) + x_i\zeta_i)]_{\mathcal{F}^{t-1}}\|_2$$

$$\leq \|\Delta_{t-1} - \eta[\frac{1}{m}\sum_{i=1}^m (x_i(\langle x_i, \theta_{t-1} - \theta^* \rangle))]_{\mathcal{F}^{t-1}}\|_2 + |\sqrt{\mathcal{F}^{t-1}}|\|\frac{1}{m}\sum_{i=1}^m x_i\zeta_i\|_\infty.$$

We first consider the term $\|\frac{1}{m}\sum_{i=1}^m x_i\zeta_i\|_\infty$. Specifically, we consider each coordinate $j \in [d]$, $|\frac{1}{m}\sum_{i=1}^m x_{i,j}\zeta_i|$. Since $\mathbb{E}[\zeta_i] = 0$ and is independent on $x$ we have $\mathbb{E}[\zeta_i x_j] = 0$. Moreover, we have

$$\|\zeta_i\|_{\psi_2} \leq \|\langle x_i, \theta^* \rangle\|_{\psi_2} + \|y_i\|_{\psi_2} \leq O(\sigma) = O(1).$$

Thus, $\|\zeta x\|_{\psi_1} \leq O(\sigma^2)$ by Lemma 10. By Bernstein's inequality we have

$$|\frac{1}{m}\sum_{i=1}^m x_{i,j}\zeta_i| \leq O(\frac{\sqrt{\log 1/\delta'}}{\sqrt{m}}). \tag{20}$$

Thus, with probability $1 - O(d^{-c})$ we have

$$\|\frac{1}{m}\sum_{i=1}^m x_i\zeta_i\|_\infty \leq O(\frac{\sqrt{\log d}}{\sqrt{m}}).$$

Finally, we consider the term $\|\Delta_{t-1} - \eta[\frac{1}{m}\sum_{i=1}^m (x_i(\langle x_i, \theta_{t-1} - \theta^*\rangle))]_{\mathcal{F}^{t-1}}\|_2$:

$$\|\Delta_{t-1} - \eta[\frac{1}{m}\sum_{i=1}^m (x_i(\langle x_i, \theta_{t-1} - \theta^*\rangle))]_{\mathcal{F}^{t-1}}\|_2 = \|[(I - D^{t-1})\Delta_{t-1}]_{\mathcal{F}^{t-1}}\|_2,$$

where $D^{t-1} = \frac{1}{m}\sum_{i \in S_t} x_i x_i^T \in \mathbb{R}^{d \times d}$. Since $\text{Supp}\left(D^{t-1}\Delta_{t-1}\right) \subset \mathcal{F}^{t-1}$ (by assumption), we have $\left\|\Delta_{t-1} - \eta D^{t-1}_{\mathcal{F}^{t-1},\cdot}\Delta_{t-1}\right\|_2 \le \left\|\left(I - \eta D_{\mathcal{F}^{t-1},\mathcal{F}^{t-1}}\right)\right\|_2 \|\Delta_{t-1}\|_2$. Next we will bound the term $\|\left(I - \eta D_{\mathcal{F}^{t-1},\mathcal{F}^{t-1}}\right)\|_2$, where $I$ is the $\left|\mathcal{F}^{t-1}\right|$-dimensional identity matrix.

Before giving analysis, we show that each of the partitioned dataset safisfies the Restriced Isometry Property (RIP) defined as follows.

**Definition 9.** *We say that a data matrix $X \in \mathbb{R}^{n \times d}$ satisfies the Restricted Isometry Property (RIP) with parameter $2k' + k$, if for any $v \in \mathbb{R}^p$ with $\|v\|_0 \le 2k' + k$, there exists a constant $\Delta$ which satisfies $(1 - \Delta)\|v\|^2 \le \frac{1}{n}\|Xv\|_2^2 \le (1 + \Delta)\|v\|_2^2$.*

The following lemma states that with high probability, where $c$ is some constant each $X_{S_t}$ on our algorithm satisfies Definition 9 and thus we can make use of this property to bound the term $D^{t-1}_{\mathcal{F}^{t-1},\mathcal{F}^{t-1}}$.

**Lemma 33.** *(Theorem 10.5.11 in Vershynin (2018)). Consider an $n \times d$ matrix $A$ whose rows $(A_i)$ are independent, isotropic, and sub-gaussian random vectors, and let $K := \max_i \|A_i\|_{\psi_2}$. Assume that*

$$n \ge CK^4 s \log(ed/s).$$

*Then, with probability at least $1 - 2\exp\left(-cn/K^4\right)$, the random matrix $A$ satisfies RIP with parameters $s$ and $\Delta = 0.1$.*

Thus, since $\{x_i\}$ are isotropic and $\|x_i\|_{\psi_2} \le O(\sigma)$, we have with probability at least $1 - 2T\exp\left(-cm/\sigma^4\right)$, $\{X_{S_t}\}_{t=1}^T$ all satisfy RIP when $m \ge \tilde{\Omega}(\sigma^4(2k' + k))$. By the RIP property and $\left|\mathcal{F}^{t-1}\right| \le 2k' + k$, we obtain the following using Lemma 33 for any $\left|\mathcal{F}^{t-1}\right|$-dimensional vector $v$

$$0.9\|v\|_2^2 \le v^T D^{t-1}_{\mathcal{F}^{t-1},\mathcal{F}^{t-1}} v \le 1.1\|v\|_2^2.$$

Thus, $\left\|\left(I - \eta D^{t-1}_{\mathcal{F}^{t-1},\mathcal{F}^{t-1}}\right)\right\|_2 \le \max\{1 - \eta \cdot 0.9, \eta \cdot 1.1 - 1\}$. This means that we can take $\eta = O(1)$ such that

$$\left\|\left(I - \eta D^{t-1}_{\mathcal{F}^{t-1},\mathcal{F}^{t-1}}\right)\right\|_2 \le \frac{2}{7}.$$

In total we have with probability at least $1 - O(d^{-c})$

$$\left\|\tilde{\theta}_{t-\frac{1}{2}} - \theta^*\right\|_2 \le \frac{2}{7}\|\Delta_{t-1}\|_2 + O\left(\sqrt{2k' + k}\frac{(\tau_1^2\sqrt{dk'} + \tau_1\tau_2\sqrt{d})\sqrt{\log\frac{d}{\delta'}}}{\sqrt{m}\epsilon}\right). \qquad (21)$$

Our next task is to bound $\|\theta_t' - \theta^*\|_2$ by $\left\|\tilde{\theta}_{t-\frac{1}{2}} - \theta^*\right\|_2$ by Lemma 21 . Thus, we have $\left\|\theta_t' - \tilde{\theta}_{t-\frac{1}{2}}\right\|_2^2 \le \frac{|\mathcal{F}^{t-1}| - k'}{|\mathcal{F}^{t-1}| - k}\left\|\tilde{\theta}_{t-\frac{1}{2}} - \theta^*\right\|_2^2 \le \frac{k' + k}{2k'}\left\|\tilde{\theta}_{t-\frac{1}{2}} - \theta^*\right\|_2^2$.

Taking $k' = 8k$, we get

$$\left\|\theta_t' - \tilde{\theta}_{t-\frac{1}{2}}\right\|_2 \le \frac{3}{4}\left\|\tilde{\theta}_{t-\frac{1}{2}} - \theta^*\right\|_2$$

and

$$\|\theta_t' - \theta^*\|_2 \le \frac{7}{4}\left\|\tilde{\theta}_{t-\frac{1}{2}} - \theta^*\right\|_2 \le \frac{1}{2}\|\Delta_{t-1}\|_2 + O\left(\sqrt{k}\frac{(\tau_1^2\sqrt{dk} + \tau_1\tau_2\sqrt{d})\sqrt{\log\frac{d}{\delta'}}}{\sqrt{m}\epsilon}\right).$$

Finally, we need to show that $\|\Delta_t\|_2 = \|\theta_t - \theta^*\|_2 \le \|\theta'_t - \theta^*\|_2$, which is due to the Lemma 22. Putting all together, we have the following with probability at least $1 - O(d^{-c})$,

$$\|\Delta_t\|_2 \le \frac{1}{2}\|\Delta_{t-1}\|_2 + O\left(\sqrt{k}\frac{\log n\sqrt{Tdk\log d}}{\sqrt{n}\epsilon}\right).$$

Thus, with probability at least $1 - O(Td^{-c})$ we have

$$\|\Delta_T\|_2 \le (\frac{1}{2})^T\|\theta^*\|_2 + O\left(\frac{k\log n\sqrt{Td\log d}}{\sqrt{n}\epsilon}\right).$$

Take $T = O(\log n)$. We have the result. $\qquad\square$

## F UPPER BOUND OF LDP-IHT FOR GENERAL SUB-GAUSSIAN DISTRIBUTIONS

---

**Algorithm 4** LDP Iterative Hard Thresholding

---

1: **Input:** Private data $\{(x_i, y_i)\}_{i=1}^n \in (\mathbb{R}^d \times \mathbb{R})^n$. Iteration number $T$, privacy parameter $\epsilon$, step size $\eta$, truncation parameters $\tau, \tau_1, \tau_2$, threshold $k'$. Initial parameter $\theta_0 = 0$.
2: For the $i$-th user with $i \in [n]$, truncate his/her data as follows: shrink $x_i$ to $\tilde{x}_i$ with $\widetilde{x}_{ij} = \mathrm{sgn}(x_{ij})\min\{|x_{ij}|, \tau_1\}$ for $j \in [d]$, and $\tilde{y}_i := \mathrm{sgn}(y_i)\min\{|y_i|, \tau_2\}$. Partition the users into $T$ groups. For $t = 1, \cdots, T$, define the index set $S_t = \{(t-1)\lfloor\frac{n}{T}\rfloor + 1, \cdots, t\lfloor\frac{n}{T}\rfloor\}$; if $t = T$, then $S_t = S_t \bigcup \{t\lfloor\frac{n}{T}\rfloor + 1, \cdots, n\}$.
3: **for** $t = 1, 2, \cdots, T$ **do**
4:     The server sends $\theta_{t-1}$ to all the users in $S_t$. Each user $i \in S_t$ perturbs his/her own gradient: let $\nabla_i = \tilde{x}_i^T(\langle\theta_{t-1}, \tilde{x}_i\rangle - \tilde{y}_i)$, compute $z_i = \mathcal{R}_\epsilon^r(\nabla_i)$, where $\mathcal{R}_\epsilon^r$ is the randomizer defined in equation 6 with $r = \sqrt{d}\tau_1(2\sqrt{k'}\tau_1 + \tau_2)$ and send back to the server.
5:     The server computes $\tilde{\nabla}_{t-1} = \frac{1}{|S_t|}\sum_{i \in S_t} z_i$ and performs the gradient descent update $\tilde{\theta}_t = \theta_{t-1} - \eta_0\tilde{\nabla}_{t-1}$.
6:     $\theta'_t = \mathrm{Trunc}(\tilde{\theta}_{t-1}, k')$.
7:     $\theta_t = \arg_{\theta \in \mathbb{B}_2(2)}\|\theta - \theta'_t\|_2$.
8: **end for**
9: **Output:** $\theta_T$

---

Theorem 7 establishes the upper bound specifically for isotropic sub-Gaussian distributions. However, we can also demonstrate that the aforementioned upper bound also holds for general sub-Gaussian distributions, albeit with different parameters. Notably, for general sub-Gaussian distributions, we need to slightly modify the LDP-IHT algorithm (Algorithm 2). Specifically, rather than projecting onto the unit $\ell_2$-norm ball, here we need to project onto the centered $\ell_2$-norm ball with radius 2 (actually, we can project onto any centered ball with a radius larger than 1). See Algorithm 4 for details. Such a modification is necessary for our proof, as we can show that with high probability, $\|\theta'_t\|_2 \le 2$ for all $t \in [T]$, which implies there is no projection with high probability. Since we use a different radius, the $\ell_2$-norm sensitivity of $\nabla_i$ also has been changed to ensure $\epsilon$-LDP. In the following, we present the theoretical result assuming that the initial parameter $\theta_0$ is sufficently close to $\theta^*$.

**Theorem 34.** *For any $\epsilon > 0$, Algorithm 4 is $\epsilon$-LDP. Moreover, under Assumptions 1 and 2, if the initial parameter $\theta_0$ satisfies $\|\theta_0 - \theta^*\|_2 \le \frac{1}{2}\frac{\mu}{\gamma}$ and $n$ is sufficiently large such that $n \ge \tilde{\Omega}(\frac{k'^2 d}{\epsilon^2})$, setting $\eta_0 = \frac{2}{3\gamma}$, $k' = 72\frac{\gamma^2}{\mu^2}k$, with probability at least $1 - \delta'$ we have*

$$\|\theta_T - \theta^*\|_2 \le O(\frac{\sqrt{d}k\log^2 n\sqrt{\log\frac{d}{\delta}}}{\sqrt{n}\epsilon}),$$

*where $\gamma = \lambda_{\max}(\mathbb{E}[xx^T])$, $\mu = \lambda_{\min}(\mathbb{E}[xx^T])$, big-O and big-$\Omega$ notations omit the terms of $\sigma, \gamma$ and $\mu$.*

**Proof of Theorem 34.** The proof of privacy is almost the same as the proof of Theorem 7. The only difference is that here we have $\|\nabla_i\|_2 \leq \sqrt{d}\tau_1(2\sqrt{k'}\tau_1 + \tau_2)$. In the following, we will show the utility. We first recall two definitions and one lemma.

**Definition 10.** *A function $f$ is $L$-Lipschitz w.r.t the norm $\|\cdot\|$ if for all $w, w' \in \mathcal{W}$, $|f(w) - f(w')| \leq L\|w - w'\|$.*

**Definition 11.** *A function $f$ is $\alpha$-smooth on $\mathcal{W}$ if for all $w, w' \in \mathcal{W}$, $f(w') \leq f(w) + \langle \nabla f(w), w' - w \rangle + \frac{\alpha}{2}\|w' - w\|_2^2$.*

**Lemma 35** (Lemma 1 in Jain et al. (2014) ). *For any index set $I$, any $v \in \mathbb{R}^{|I|}$, let $\tilde{v} = Trunc(v, k)$. Then for any $v^* \in \mathbb{R}^{|I|}$ such that $\|v^*\|_0 \leq k^*$ we have*

$$\|\tilde{v} - v\|_2^2 \leq \frac{|I| - k}{|I| - k^*}\|v^* - v\|_2^2. \tag{22}$$

For simplicity we denote $L(\theta) = \mathbb{E}[(\langle x, \theta \rangle - y)^2]$, $\tilde{\nabla}L_{t-1} = \frac{1}{m}\sum_{x \in \tilde{D}_t} \tilde{x}(\langle \tilde{x}, \theta_{t-1} \rangle - \tilde{y})$, $\nabla L_{t-1} = \nabla L(\theta_{t-1}) = \mathbb{E}[x(\langle x, \theta_{t-1} \rangle - y)]$, $S^{t-1} = \text{supp}(\theta_{t-1})$, $S^t = \text{supp}(\theta_t)$, $S^* = \text{supp}(\theta^*)$ and $I^t = S^t \bigcup S^{t-1} \bigcup S^*$. We can see that $|S^{t-1}| \leq k'$, $|S^t| \leq k'$ and $|I^t| \leq 2k' + k$. We let $\gamma = \lambda_{\max}(\mathbb{E}[xx^T])$, $\mu = \lambda_{\min}(\mathbb{E}[xx^T])$ and $\eta_0 = \frac{\eta}{\gamma}$ for some $\eta$. We can easily see that $L(\cdot)$ is $\mu$-strongly convex and $\gamma$-smooth.

Then from the smooth property we have

$$
\begin{aligned}
L(\theta'_t) &- L(\theta_{t-1}) \\
&\leq \langle \theta'_t - \theta_{t-1}, \nabla L_{t-1} \rangle + \frac{\gamma}{2}\|\theta'_t - \theta_{t-1}\|_2^2 \\
&= \langle \theta'_{t,I^t} - \theta_{t-1,I^t}, \nabla L_{t-1,I^t} \rangle + \frac{\gamma}{2}\|\theta'_{t,I^t} - \theta_{t-1,I^t}\|_2^2 \\
&\leq \frac{\gamma}{2}\|\theta'_{t,I^t} - \theta_{t-1,I^t} + \frac{\eta}{\gamma}\nabla L_{t-1,I^t}\|_2^2 - \frac{\eta^2}{2\gamma}\|\nabla L_{t-1,I^t}\|_2^2 + (1-\eta)\langle \theta'_t - \theta_{t-1}, \nabla L_{t-1} \rangle \quad (23)
\end{aligned}
$$

First, let us focus on the third term of (23). By Lemma 8 and the definition, we know that $\theta'_t$ can be written as $\theta'_t = \hat{\theta}_{t,S^t} + \phi_{t,S^t}$, where $\hat{\theta}_t = (\theta_{t-1} - \eta_0\tilde{\nabla}L_{t-1})_{S^t}$ and $\phi_t$ is a sub-Gaussian vector with variance $= O\left(\frac{d\tau_1^2(k'\tau_1^2 + \tau_2^2)}{m\epsilon^2}\right)$. Thus,

$$
\begin{aligned}
\langle \theta'_t - \theta_{t-1}, \nabla L_{t-1} \rangle &= \langle \hat{\theta}_{t,S^t} - \theta_{t-1,S^t}, \nabla L_{t-1,S^t} \rangle \\
&+ \langle \phi_{t,S^t}, \nabla L_{t-1,S^t} \rangle - \langle \theta_{t-1,S^{t-1}\setminus S^t}, \nabla L_{t-1,S^{t-1}\setminus S^t} \rangle. \quad (24)
\end{aligned}
$$

For the first term in (24) we have

$$
\begin{aligned}
\langle \hat{\theta}_{t,S^t} - \theta_{t-1,S^t}, \nabla L_{t-1,S^t} \rangle &= \langle -\eta_0\tilde{\nabla}L_{t-1,S^t}, \nabla L_{t-1,S^t} \rangle = -\frac{\eta}{\gamma}\langle \tilde{\nabla}L_{t-1,S^t}, \nabla L_{t-1,S^t} \rangle \\
&= -\frac{\eta}{\gamma}\|\nabla L_{t-1,S^t}\|_2^2 - \frac{\eta}{\gamma}\langle \tilde{\nabla}L_{t-1,S^t} - \nabla L_{t-1,S^t}, \nabla L_{t-1,S^t} \rangle \\
&\leq -\frac{\eta}{\gamma}\|\nabla L_{t-1,S^t}\|_2^2 + \frac{\eta}{2\gamma}\|\nabla L_{t-1,S^t}\|_2^2 + \frac{\eta}{2\gamma}\|\tilde{\nabla}L_{t-1,S^t} - \nabla L_{t-1,S^t}\|_2^2 \\
&= -\frac{\eta}{2\gamma}\|\nabla L_{t-1,S^t}\|_2^2 + \frac{\eta}{2\gamma}\|\tilde{\nabla}L_{t-1,S^t} - \nabla L_{t-1,S^t}\|_2^2. \quad (25)
\end{aligned}
$$

Take (25) into (24) we have for $c_1 > 0$

$$
\begin{aligned}
\langle \theta'_t - \theta_{t-1}, \nabla L_{t-1} \rangle &\leq -\frac{\eta}{2\gamma}\|\nabla L_{t-1,S^t}\|_2^2 + \frac{\eta}{2\gamma}\|\tilde{\nabla}L_{t-1,S^t} - \nabla L_{t-1,S^t}\|_2^2 \\
&+ c_1\|\phi_{t,S^t}\|_2^2 + \frac{1}{4c_1}\|\nabla L_{t-1,S^t}\|_2^2 - \langle \theta_{t-1,S^{t-1}\setminus S^t}, \nabla L_{t-1,S^{t-1}\setminus S^t} \rangle. \quad (26)
\end{aligned}
$$

For the last term of (26) we have

$$-\langle \theta_{t-1,S^{t-1}\setminus S^t}, \nabla L_{t-1,S^{t-1}\setminus S^t} \rangle$$

$$\leq \frac{\gamma}{2\eta}(\|\theta_{t-1,S^{t-1}\backslash S^t} - \frac{\eta}{\gamma}\nabla L_{t-1,S^{t-1}\backslash S^t}\|_2^2 - (\frac{\eta}{\gamma})^2\|\nabla L_{t-1,S^{t-1}\backslash S^t}\|_2^2)$$

$$= \frac{\gamma}{2\eta}\|\theta_{t-1,S^{t-1}\backslash S^t} - \frac{\eta}{\gamma}\nabla L_{t-1,S^{t-1}\backslash S^t}\|_2^2 - \frac{\eta}{2\gamma}\|\nabla L_{t-1,S^{t-1}\backslash S^t}\|_2^2$$

$$\leq \frac{\eta}{2\gamma}(1+\frac{1}{c_1})\|\nabla L_{t-1,S^t\backslash S^{t-1}}\|_2^2 + \frac{2\eta}{\gamma}(1+c_1)\|\nabla L_{t-1,S^t\backslash S^{t-1}} - \tilde\nabla L_{t-1,S^t\backslash S^{t-1}} - \phi_{t,S^t\backslash S^{t-1}}\|_2^2$$

$$- \frac{\eta}{2\gamma}\|\nabla L_{t-1,S^{t-1}\backslash S^t}\|_2^2, \tag{27}$$

where the last inequality comes from

$$\|\theta_{t-1,S^{t-1}\backslash S^t} - \frac{\eta}{\gamma}\nabla L_{t-1,S^{t-1}\backslash S^t}\|_2 - \frac{\eta}{\gamma}\|\nabla L_{t-1,S^{t-1}\backslash S^t} - \tilde\nabla L_{t-1,S^{t-1}\backslash S^t} - \phi_{t,S^{t-1}\backslash S^t}\|_2$$

$$\leq \|\theta_{t-1,S^{t-1}\backslash S^t} - \frac{\eta}{\gamma}(\tilde\nabla L_{t-1,S^{t-1}\backslash S^t} + \phi_{t,S^{t-1}\backslash S^t})\|_2$$

$$\leq \|\theta_{t-1,S^t\backslash S^{t-1}} - \frac{\eta}{\gamma}(\tilde\nabla L_{t-1,S^t\backslash S^{t-1}} + \phi_{t,S^t\backslash S^{t-1}})\|_2 = \frac{\eta}{\gamma}\|\tilde\nabla L_{t-1,S^t\backslash S^{t-1}} + \phi_{t,S^t\backslash S^{t-1}}\|_2$$

$$\leq \frac{\eta}{\gamma}\|\nabla L_{t-1,S^t\backslash S^{t-1}}\|_2 + \frac{\eta}{\gamma}\|\nabla L_{t-1,S^t\backslash S^{t-1}} - \tilde\nabla L_{t-1,S^t\backslash S^{t-1}} - \phi_{t,S^t\backslash S^{t-1}}\|_2,$$

where the second inequality is due to the fact that $|S^t\backslash S^{t-1}| = |S^{t-1}\backslash S^t|$. the definitions of hard thresholding, $\theta_t' = (\theta_{t-1} - \frac{\eta}{\gamma}(\tilde\nabla L_{t-1} + \phi_t))_{S^t}$, $S^t$ and $S^{t-1}$; the first equality is due to $\text{Supp}(\theta_{t-1}) = S^{t-1}$ Thus we have

$$\frac{\gamma}{2\eta}\|\theta_{t-1,S^{t-1}\backslash S^t} - \frac{\eta}{\gamma}\nabla L_{t-1,S^{t-1}\backslash S^t}\|_2^2$$

$$\leq \frac{\eta}{2\gamma}(1+\frac{1}{c_1})\|\nabla L_{t-1,S^t\backslash S^{t-1}}\|_2^2 + \frac{2\eta}{\gamma}(1+c_1)\|\nabla L_{t-1,S^t\backslash S^{t-1}} - \tilde\nabla L_{t-1,S^t\backslash S^{t-1}} - \phi_{t,S^t\backslash S^{t-1}}\|_2^2$$

We can easily see that

$$\frac{\eta}{2\gamma}\|\nabla L_{t-1,S^t\backslash S^{t-1}}\|_2^2 - \frac{\eta}{2\gamma}\|\nabla L_{t-1,S^{t-1}\backslash S^t}\|_2^2 - \frac{\eta}{2\gamma}\|\nabla L_{t-1,S^t}\|_2^2$$

$$= -\frac{\eta}{2\gamma}\|\nabla L_{t-1,S^{t-1}\backslash S^t}\|_2^2 - \frac{\eta}{2\gamma}\|\nabla L_{t-1,S^t\cap S^{t-1}}\|_2^2$$

$$= -\frac{\eta}{2\gamma}\|\nabla L_{t-1,S^t\cup S^{t-1}}\|_2^2.$$

In total

$$\langle\theta_t' - \theta_{t-1}, \nabla L_{t-1}\rangle$$

$$\leq -\frac{\eta}{2\gamma}\|\nabla L_{t-1,S^t\cup S^{t-1}}\|_2^2 + (\frac{1}{4c_1} + \frac{\eta}{2\gamma c_1})\|\nabla L_{t-1,S^t}\|_2^2 + \frac{\eta}{2\gamma}\|\tilde\nabla L_{t-1,S^t} - \nabla L_{t-1,S^t}\|_2^2$$

$$+ c_1\|\phi_{t,S^t}\|_2^2 + \frac{2\eta}{\gamma}(1+c_1)\|\nabla L_{t-1,S^t\backslash S^{t-1}} - \tilde\nabla L_{t-1,S^t\backslash S^{t-1}} - \phi_{t,S^t\backslash S^{t-1}}\|_2^2 \tag{28}$$

Take (28) into (23) we have

$$L(\theta_t') - L(\theta_{t-1}) \leq \frac{\gamma}{2}\|\theta_{t,I^t}' - \theta_{t-1,I^t} + \frac{\eta}{\gamma}\nabla L_{t-1,I^t}\|_2^2 - \frac{\eta^2}{2\gamma}\|\nabla L_{t-1,I^t}\|_2^2 + (1-\eta)\langle\theta_t' - \theta_{t-1}, \nabla L_{t-1}\rangle$$

$$\leq \frac{\gamma}{2}\|\theta_{t,I^t}' - \theta_{t-1,I^t} + \frac{\eta}{\gamma}\nabla L_{t-1,I^t}\|_2^2 - \frac{\eta^2}{2\gamma}\|\nabla L_{t-1,I^t}\|_2^2 - \frac{(1-\eta)\eta}{2\gamma}\|\nabla L_{t-1,S^t\cup S^{t-1}}\|_2^2$$

$$+ (1-\eta)(\frac{1}{4c_1} + \frac{\eta}{2\gamma c_1})\|\nabla L_{t-1,S^t}\|_2^2 + (1-\eta)[\frac{\eta}{2\gamma}\|\tilde\nabla L_{t-1,S^t} - \nabla L_{t-1,S^t}\|_2^2 + c_1\|\phi_{t,S^t}\|_2^2$$

$$+ \frac{2\eta}{\gamma}(1+c_1)\|\nabla L_{t-1,S^t\backslash S^{t-1}} - \tilde\nabla L_{t-1,S^t\backslash S^{t-1}} - \phi_{t,S^t\backslash S^{t-1}}\|_2^2]$$

$$\leq \frac{\gamma}{2}\|\theta_{t,I^t}' - \theta_{t-1,I^t} + \frac{\eta}{\gamma}\nabla L_{t-1,I^t}\|_2^2 - \frac{\eta^2}{2\gamma}\|\nabla L_{t-1,I^t\backslash(S^{t-1}\cup S^*)}\|_2^2$$

$$-\frac{\eta^2}{2\gamma}\|\nabla L_{t-1,(S^{t-1}\bigcup S^*)}\|_2^2 - \frac{(1-\eta)\eta}{2\gamma}\|\nabla L_{t-1,S^t\bigcup S^{t-1}}\|_2^2$$

$$+(1-\eta)(\frac{1}{4c_1}+\frac{\eta}{2\gamma c_1})\|\nabla L_{t-1,S^t}\|_2^2 + (1-\eta)[\frac{\eta}{2\gamma}\|\tilde{\nabla}L_{t-1,S^t}-\nabla L_{t-1,S^t}\|_2^2 + c_1\|\phi_{t,S^t}\|_2^2$$

$$+\frac{2\eta}{\gamma}(1+c_1)\|\nabla L_{t-1,S^t\setminus S^{t-1}}-\tilde{\nabla}L_{t-1,S^t\setminus S^{t-1}}-\phi_{t,S^t\setminus S^{t-1}}\|_2^2]$$

$$\leq \frac{\gamma}{2}\|\theta'_{t,I^t}-\theta_{t-1,I^t}+\frac{\eta}{\gamma}\nabla L_{t-1,I^t}\|_2^2 - \frac{\eta^2}{2\gamma}\|\nabla L_{t-1,I^t\setminus(S^{t-1}\bigcup S^*)}\|_2^2 - \frac{\eta^2}{2\gamma}\|\nabla L_{t-1,(S^{t-1}\bigcup S^*)}\|_2^2$$

$$-\frac{(1-\eta)\eta}{2\gamma}\|\nabla L_{t-1,S^t\setminus(S^*\bigcup S^{t-1})}\|_2^2 + (1-\eta)(\frac{1}{4c_1}+\frac{\eta}{2\gamma c_1})\|\nabla L_{t-1,S^t}\|_2^2$$

$$+\underbrace{(1-\eta)(\frac{\eta}{2\gamma}\|\tilde{\nabla}L_{t-1,S^t}-\nabla L_{t-1,S^t}\|_2^2 + c_1\|\phi_{t,S^t}\|_2^2 + \frac{2\eta}{\gamma}(1+c_1)\|\nabla L_{t-1,S^t\setminus S^{t-1}}-\tilde{\nabla}L_{t-1,S^t\setminus S^{t-1}}-\phi_{t,S^t\setminus S^{t-1}}\|_2^2)}_{N_0^t},$$

$$\tag{29}$$

where the last inequality is due to $S^t\setminus(S^*\bigcup S^{t-1})\subseteq S^t\bigcup S^{t-1}$. Next we will analyze the term $\frac{\gamma}{2}\|\theta'_{t,I^t}-\theta_{t-1,I^t}+\frac{\eta}{\gamma}\nabla L_{t-1,I^t}\|_2^2 - \frac{\eta^2}{2\gamma}\|\nabla L_{t-1,I^t\setminus(S^{t-1}\bigcup S^*)}\|_2^2$ in (29).

Let $R$ be a subset of $S^{t-1}\setminus S^t$ such that $|R|=|I^t\setminus(S^*\bigcup S^{t-1})|=|S^t\setminus(S^{t-1}\bigcup S^*)|$. By the definition of hard thresholding, we can easily see

$$\|\theta_{t-1,R}-\frac{\eta}{\gamma}(\tilde{\nabla}L_{t-1,R}+\phi_{t,R})\|_2^2 \leq \|(\theta_{t-1}-\frac{\eta}{\gamma}(\tilde{\nabla}L_{t-1}+\phi_t))_{I^t\setminus(S^*\bigcup S^{t-1})}\|_2^2$$
$$=\frac{\eta^2}{\gamma^2}\|(\tilde{\nabla}L_{t-1}+\phi_t)_{I^t\setminus(S^*\bigcup S^{t-1})}\|_2^2. \tag{30}$$

Thus we have

$$(\frac{\eta}{\gamma})\|\nabla L_{t-1,I^t\setminus(S^*\bigcup S^{t-1})}\|_2$$

$$\geq \underbrace{\|\theta_{t-1,R}-\frac{\eta}{\gamma}\nabla L_{t-1,R}\|_2}_{a} - \frac{\eta}{\gamma}(\underbrace{\|\tilde{\nabla}L_{t-1,R}-\nabla L_{t-1,R}+\phi_{t,R}\|_2}_{b} \tag{31}$$

$$+\underbrace{\|\nabla L_{t-1,I^t\setminus(S^*\bigcup S^{t-1})}-\tilde{\nabla}L_{t-1,I^t\setminus(S^*\bigcup S^{t-1})}-\phi_{t,I^t\setminus(S^*\bigcup S^{t-1})}\|_2}_{c})$$

Then we have for any $c_2>0$

$$\frac{\gamma}{2}\|\theta'_{t,I^t}-\theta_{t-1,I^t}+\frac{\eta}{\gamma}\nabla L_{t-1,I^t}\|_2^2 - \frac{\eta^2}{2\gamma}\|\nabla L_{t-1,I^t\setminus(S^{t-1}\bigcup S^*)}\|_2^2$$

$$\leq \frac{\gamma}{2}\|\theta'_{t,I^t}-\theta_{t-1,I^t}+\frac{\eta}{\gamma}\nabla L_{t-1,I^t}\|_2^2 - \frac{\gamma}{2}(\frac{\eta^2}{\gamma^2}(b+c)^2+a^2-\frac{2\eta}{\gamma}(b+c)a)$$

$$\leq \frac{\gamma}{2}\|\theta'_{t,I^t}-\theta_{t-1,I^t}+\frac{\eta}{\gamma}\nabla L_{t-1,I^t}\|_2^2 - \frac{\gamma}{2}(1-\frac{1}{c_2})a^2+(2c_2-\frac{1}{2})\frac{\eta^2}{\gamma}(b+c)^2$$

$$= \frac{\gamma}{2}\|\theta'_{t,I^t\setminus R}-\theta_{t-1,I^t\setminus R}+\frac{\eta}{\gamma}\nabla L_{t-1,I^t\setminus R}\|_2^2 + \frac{\gamma}{2c_2}\|\theta_{t-1,R}-\frac{\eta}{\gamma}\nabla L_{t-1,R}\|_2^2$$

$$+\underbrace{(4c_2-1)\frac{\eta^2}{\gamma}(\|\tilde{\nabla}L_{t-1,R}-\nabla L_{t-1,R}+\phi_{t,I^t}\|_2^2 + \|\nabla L_{t-1,I^t\setminus(S^*\bigcup S^{t-1})}-\tilde{\nabla}L_{t-1,I^t\setminus(S^*\bigcup S^{t-1})}-\phi_{t,I^t\setminus(S^*\bigcup S^{t-1})}\|_2^2)}_{N_1^t}$$

$$\tag{32}$$

$$\leq \frac{\gamma}{2}\|\theta'_{t,I^t\setminus R}-\theta_{t-1,I^t\setminus R}+\frac{\eta}{\gamma}\nabla L_{t-1,I^t\setminus R}\|_2^2 + \frac{\gamma}{c_2}\|\theta_{t-1,R}-\frac{\eta}{\gamma}(\tilde{\nabla}L_{t-1,R}+\phi_{t,R})\|_2^2$$

$$+\underbrace{\frac{\eta^2}{c_2\gamma}\|\nabla L_{t-1,I^t\setminus R}-(\tilde{\nabla}L_{t-1,R}+\phi_{t,R})\|_2^2 + N_1^t}_{N_2^t} \tag{33}$$

$$= \frac{\gamma}{2}\|\theta'_{t,I^t\backslash R} - \theta_{t-1,I^t\backslash R} + \frac{\eta}{\gamma}\nabla L_{t-1,I^t\backslash R}\|_2^2 + N_2^t, \tag{34}$$

where (32) is due to that $\theta'_{t-1,R} = 0$, thus $\|\theta'_{t-1,R} - (\theta_{t-1,R} - \frac{\eta}{\gamma}\nabla L_{t-1,R})\|_2 = \|\theta_{t-1,R} - \frac{\eta}{\gamma}\nabla L_{t-1,R}\|_2$. In the following, we will consider the first term in (34).

In Lemma 35, take $v = \theta_{t-1,I^t\backslash R} - \frac{\eta}{\gamma}(\tilde{\nabla}L_{t-1,I^t\backslash R} + \phi_{t-1,I^t\backslash R})$, $\tilde{v} = \mathrm{Trunc}(v,k') = \theta'_{t-1,I^t\backslash R}$, $I = I^t\backslash R$, $v^* = \theta^*_{I^t\backslash R} = \theta^*$, we have

$$\|\theta'_{t,I^t\backslash R} - \theta_{t-1,I^t\backslash R} - \frac{\eta}{\gamma}(\tilde{\nabla}L_{t-1,I^t\backslash R} + \phi_{t-1,I^t\backslash R})\|_2^2 \le \frac{|I^t\backslash R| - k'}{|I^t\backslash R| - k}\|\theta^* - \theta_{t-1,I^t\backslash R} - \frac{\eta}{\gamma}(\tilde{\nabla}L_{t-1,I^t\backslash R} + \phi_{t-1,I^t\backslash R})\|_2^2.$$

Then we have

$$(1 - \frac{1}{c_3})\|\theta'_{t,I^t\backslash R} - \theta_{t-1,I^t\backslash R} + \frac{\eta}{\gamma}\nabla L_{t-1,I^t\backslash R}\|_2^2 - (c_3 - 1)\frac{\eta^2}{\gamma^2}\|\nabla L_{t-1,I^t\backslash R} - \tilde{\nabla}L_{t-1,I^t\backslash R} - \phi_{t-1,I^t\backslash R}\|_2^2$$

$$\le \|\theta'_{t,I^t\backslash R} - \theta_{t-1,I^t\backslash R} + \frac{\eta}{\gamma}\tilde{\nabla}L_{t-1,I^t\backslash R}\|_2^2$$

$$\le \frac{|I^t\backslash R| - k'}{|I^t\backslash R| - k}\|\theta^* - \theta_{t-1,I^t\backslash R} + \frac{\eta}{\gamma}\tilde{\nabla}L_{t-1,I^t\backslash R}\|_2^2$$

$$\le \frac{|I^t\backslash R| - k'}{|I^t\backslash R| - k}\left((1 + \frac{1}{c_3})\|\theta^* - \theta_{t-1,I^t\backslash R} + \frac{\eta}{\gamma}\nabla L_{t-1,I^t\backslash R}\|_2^2 + (1 + c_3)\frac{\eta^2}{\gamma^2}\|\nabla L_{t-1,I^t\backslash R} - \tilde{\nabla}L_{t-1,I^t\backslash R} - \phi_{t-1,I^t\backslash R}\|_2^2\right)$$

Since $|I^t\backslash R| \le 2k' + k$ and $k' \ge k$, we have $\frac{|I^t\backslash R| - k'}{|I^t\backslash R| - k} \le \frac{k'+k}{2k'} \le \frac{2k'}{k+k'}$. Thus

$$\|\theta'_{t,I^t\backslash R} - \theta_{t-1,I^t\backslash R} + \frac{\eta}{\gamma}\nabla L_{t-1,I^t\backslash R}\|_2^2 \le \frac{2k}{k+k'}\frac{c_3+1}{c_3-1}\|\theta^* - \theta_{t-1,I^t\backslash R} + \frac{\eta}{\gamma}\nabla L_{t-1,I^t\backslash R}\|_2^2$$

$$+ ((1+c_3)\frac{2k}{k+k'} + c_3 - 1)\frac{\eta^2}{\gamma^2}\|\nabla L_{t-1,I^t\backslash R} - \tilde{\nabla}L_{t-1,I^t\backslash R} - \phi_{t-1,I^t\backslash R}\|_2^2$$

Take $c_3 = 5$ and $k' = O(k)$, we have

$$\|\theta'_{t,I^t\backslash R} - \theta_{t-1,I^t\backslash R} + \frac{\eta}{\gamma}\nabla L_{t-1,I^t\backslash R}\|_2^2 \le \frac{3}{2}\frac{2k}{k+k'}\|\theta^* - \theta_{t-1,I^t\backslash R} + \frac{\eta}{\gamma}\nabla L_{t-1,I^t\backslash R}\|_2^2$$

$$+ \underbrace{O(\frac{\eta^2}{\gamma^2}\|\nabla L_{t-1,I^t\backslash R} - \tilde{\nabla}L_{t-1,I^t\backslash R} - \phi_{t-1,I^t\backslash R}\|_2^2)}_{N_3^t}. \tag{35}$$

Take (35) into (34) we have

$$\frac{\gamma}{2}\|\theta'_{t,I^t} - \theta_{t-1,I^t} + \frac{\eta}{\gamma}\nabla L_{t-1,I^t}\|_2^2 - \frac{\eta^2}{2\gamma}\|\nabla L_{t-1,I^t\backslash(S^{t-1}\bigcup S^*)}\|_2^2$$

$$\le \frac{3\gamma}{2}\frac{k}{k+k'}\|\theta^* - \theta_{t-1,I^t\backslash R} + \frac{\eta}{\gamma}\nabla L_{t-1,I^t\backslash R}\|_2^2 + N_2^t + \gamma N_3^t. \tag{36}$$

Take (36) into (29) we have

$$L(\theta'_t) - L(\theta_{t-1})$$

$$\le \frac{\gamma}{2}\|\theta'_{t,I^t} - \theta_{t-1,I^t} + \frac{\eta}{\gamma}\nabla L_{t-1,I^t}\|_2^2 - \frac{\eta^2}{2\gamma}\|\nabla L_{t-1,I^t\backslash(S^{t-1}\bigcup S^*)}\|_2^2 - \frac{\eta^2}{2\gamma}\|\nabla L_{t-1,(S^{t-1}\bigcup S^*)}\|_2^2$$

$$- \frac{(1-\eta)\eta}{2\gamma}\|\nabla L_{t-1,S^t\backslash(S^*\bigcup S^{t-1})}\|_2^2 + (1-\eta)(\frac{1}{4c_1} + \frac{\eta}{2\gamma c_1})\|\nabla L_{t-1,S^t}\|_2^2 + N_0^t$$

$$\le \frac{3\gamma}{2}\frac{k}{k'+k}\|\theta^* - \theta_{t-1,I^t\backslash R} + \frac{\eta}{\gamma}\nabla L_{t-1,I^t\backslash R}\|_2^2 - \frac{\eta^2}{2\gamma}\|\nabla L_{t-1,(S^{t-1}\bigcup S^*)}\|_2^2$$

$$- \frac{(1-\eta)\eta}{2\gamma}\|\nabla L_{t-1,S^t\backslash(S^*\bigcup S^{t-1})}\|_2^2 + (1-\eta)(\frac{1}{4c_1} + \frac{\eta}{2\gamma c_1})\|\nabla L_{t-1,S^t}\|_2^2 + N_0^t + N_2^t + \gamma N_3^t. \tag{37}$$

Note that when $\eta \geq \frac{1}{2}$, there exists a sufficiently large $c_1$ is such that $\frac{1}{4c_1} + \frac{\eta}{2\gamma c_1} \leq \frac{\eta}{4\gamma}$, we have

$$(1-\eta)(\frac{1}{4c_1} + \frac{\eta}{2\gamma c_1})\|\nabla L_{t-1,S^t}\|_2^2 \leq \frac{\eta(1-\eta)}{4\gamma}\|\nabla L_{t-1,S^t}\|_2^2$$

$$\leq \frac{\eta^2}{4\gamma}\|\nabla L_{t-1,(S^{t-1}\bigcup S^*)}\|_2^2 + \frac{(1-\eta)\eta}{4\gamma}\|\nabla L_{t-1,S^t\setminus(S^*\bigcup S^{t-1})}\|_2^2$$

Thus

$$L(\theta'_t) - L(\theta_{t-1})$$

$$\leq \frac{3\gamma}{2}\frac{k}{k'+k}\|\theta^* - \theta_{t-1,I^t\setminus R} + \frac{\eta}{\gamma}\nabla L_{t-1,I^t\setminus R}\|_2^2 - \frac{\eta^2}{2\gamma}\|\nabla L_{t-1,(S^{t-1}\bigcup S^*)}\|_2^2$$

$$- \frac{(1-\eta)\eta}{2\gamma}\|\nabla L_{t-1,S^t\setminus(S^*\bigcup S^{t-1})}\|_2^2 + \frac{(1-\eta)}{4c}\|\nabla L_{t-1,S^t}\|_2^2 + N_0^t + N_2^t + \gamma N_3^t$$

$$\leq \frac{3\gamma}{2}\frac{k}{k'+k}\|\theta^* - \theta_{t-1,I^t\setminus R} + \frac{\eta}{\gamma}\nabla L_{t-1,I^t\setminus R}\|_2^2 - \frac{\eta^2}{4\gamma}\|\nabla L_{t-1,(S^{t-1}\bigcup S^*)}\|_2^2$$

$$- \frac{(1-\eta)\eta}{4\gamma}\|\nabla L_{t-1,S^t\setminus(S^*\bigcup S^{t-1})}\|_2^2 + N_0^t + N_2^t + \gamma N_3^t$$

It is notable that by strong convexity

$$\frac{3\gamma}{2}\frac{k}{k'+k}\|\theta^* - \theta_{t-1,I^t\setminus R} + \frac{\eta}{\gamma}\nabla L_{t-1,I^t\setminus R}\|_2^2$$

$$\leq \frac{3\gamma}{2}\frac{k}{k'+k}\|\theta^* - \theta_{t-1,I^t} + \frac{\eta}{\gamma}\nabla L_{t-1,I^t}\|_2^2$$

$$= \frac{3\gamma}{2}\frac{k}{k'+k}(\|\theta^* - \theta_{t-1,I^t\setminus R}\|_2^2 + \frac{\eta^2}{\gamma^2}\|\nabla L_{t-1,I^t}\|_2^2 + \frac{2\eta}{\gamma}\langle \theta^* - \theta_{t-1,I^t}, \nabla L_{t-1,I^t}\rangle)$$

$$= \frac{3\gamma}{2}\frac{k}{k'+k}(\|\theta^* - \theta_{t-1,I^t\setminus R}\|_2^2 + \frac{\eta^2}{\gamma^2}\|\nabla L_{t-1,I^t}\|_2^2 + \frac{2\eta}{\gamma}\langle \theta^* - \theta_{t-1}, \nabla L_{t-1}\rangle)$$

$$\leq \frac{3k}{k'+k}(\frac{\gamma}{2}\|\theta^* - \theta_{t-1}\|_2^2 + \frac{\eta^2}{2\gamma}\|\nabla L_{t-1,I^t}\|_2^2 + \eta(L(\theta^*) - L(\theta_{t-1})) - \frac{\eta\mu}{2}\|\theta^* - \theta_{t-1}\|_2^2)$$

Take $\eta = \frac{2}{3}$, $k' = 72\frac{\gamma^2}{\mu^2}k$ so that $\frac{3k}{k'+k} \leq \frac{\mu^2}{24\gamma(\gamma-\eta\mu)} \leq \frac{1}{8}$, we have

$$L(\theta'_t) - L(\theta_{t-1})$$

$$\leq \frac{3k}{k+k'}(\eta(L(\theta^*) - L(\theta_{t-1})) + \frac{\gamma-\eta\mu}{2}\|\theta^* - \theta_{t-1}\|_2^2 + \frac{\eta^2}{2\gamma}\|\nabla L_{t-1,I^t}\|_2^2)$$

$$- \frac{\eta^2}{4\gamma}\|\nabla L_{t-1,(S^{t-1}\bigcup S^*)}\|_2^2 - \frac{(1-\eta)\eta}{4\gamma}\|\nabla L_{t-1,S^t\setminus(S^*\bigcup S^{t-1})}\|_2^2 + N_0^t + N_2^t + \gamma N_3^t$$

$$\leq \frac{2k}{k'+k}(L(\theta^*) - L(\theta_{t-1})) + \frac{\mu^2}{48\gamma}\|\theta^* - \theta_{t-1}\|_2^2 + \frac{1}{36\gamma}\|\nabla L_{t-1,I^t}\|_2^2$$

$$- \frac{1}{9\gamma}\|\nabla L_{t-1,(S^{t-1}\bigcup S^*)}\|_2^2 - \frac{1}{18\gamma}\|\nabla L_{t-1,S^t\setminus(S^*\bigcup S^{t-1})}\|_2^2 + N_0^t + N_2^t + \gamma N_3^t$$

$$\leq \frac{2k}{k+k'}(L(\theta^*) - L(\theta_{t-1})) - \frac{3}{36\gamma}(\|\nabla L_{t-1,(S^{t-1}\bigcup S^*)}\|_2^2 - \frac{\mu^2}{4}\|\theta^* - \theta_{t-1}\|_2^2) + N_0^t + N_2^t + \gamma N_3^t \tag{38}$$

$$\leq (\frac{2k}{k+k'} + \frac{\mu}{24\gamma})(L(\theta^*) - L(\theta_{t-1})) + N_0^t + N_2^t + \gamma N_3^t. \tag{39}$$

Where (38) is due to the following lemma:

**Lemma 36.** *[Lemma 6 in Jain et al. (2014)]*

$$\|\nabla L_{t-1,(S^{t-1}\bigcup S^*)}\|_2^2 - \frac{\mu^2}{4}\|\theta^* - \theta_{t-1}\|_2^2 \geq \frac{\mu}{2}(L(\theta_{t-1}) - L(\theta^*)). \tag{40}$$

Thus

$$L(\theta'_t) - L(\theta^*) \le (1 - \frac{5}{72}\frac{\mu}{\gamma})(L(\theta_{t-1}) - L(\theta^*)) + N_0^t + N_2^t + \gamma N_3^t.$$

Next, we will bound the term $N_0^t + N_2^t + \gamma N_3^t$. For $N_0^t$ we have

$$N_0^t = (1-\eta)(\frac{\eta}{2\gamma}\|\tilde{\nabla}L_{t-1,S^t} - \nabla L_{t-1,S^t}\|_2^2$$

$$+ c_1\|\phi_{t,S^t}\|_2^2 + \frac{2\eta}{\gamma}(1+c_1)\|\nabla L_{t-1,S^t\setminus S^{t-1}} - \tilde{\nabla}L_{t-1,S^t\setminus S^{t-1}} - \phi_{t,S^t\setminus S^{t-1}}\|_2^2)$$

$$= O(\frac{1}{\gamma}k'\|\tilde{\nabla}L_{t-1} - \nabla L_{t-1}\|_\infty^2 + \gamma k'\|\phi_t\|_\infty^2).$$

By equation 18, we know that with probability at least $1 - \delta'$

$$\|\tilde{\nabla}L_{t-1} - \nabla L_{t-1}\|_\infty \le O(\frac{\sigma^2 \log n \log \frac{d}{\delta'}}{\sqrt{m}}). \tag{41}$$

Moreover, by Lemma 15 we have with probability at least $1 - \delta'$

$$\|\phi_t\|_\infty \le O\left(\frac{(\tau_1^2\sqrt{dk'} + \tau_1\tau_2\sqrt{d})\sqrt{\log\frac{d}{\delta'}}}{\sqrt{m}\epsilon}\right). \tag{42}$$

Thus, with probability at least $1 - \delta'$ we have

$$N_0^t = O(\frac{\sigma^4 dk'^2 \log d/\delta' \log^2 n}{m\epsilon^2}).$$

Similarly, we have

$$N_2^t, N_3^t = O(\frac{\sigma^4 dk'^2 \log d/\delta' \log^2 n}{m\epsilon^2}).$$

Thus we have with probability at least $1 - \delta'$

$$L(\theta'_t) - L(\theta^*) \le (1 - \frac{5}{72}\frac{\mu}{\gamma})(L(\theta_{t-1}) - L(\theta^*)) + O(\frac{\sigma^4 dk'^2 \log d/\delta' \log^2 n}{m\epsilon^2}). \tag{43}$$

In the following we will assume the above event holds. We note that by our model for any $\theta$

$$\gamma\|\theta - \theta^*\|_2^2 \ge L(\theta) - L(\theta^*) \ge \mu\|\theta - \theta^*\|_2^2.$$

In the following we will show that $\theta_t = \theta'_t$ for all $t$. We will use induction, assume $\theta_i = \theta'_i$ holds for all $i \in [t-1]$, we will show that it will also true for $t$. Use (43) for $i \in [t-1]$ we have

$$\mu\|\theta'_t - \theta^*\|_2^2 \le L(\theta'_t) - L(\theta^*) \le (1 - \frac{5}{72}\frac{\mu}{\gamma})(L(\theta_{t-1}) - L(\theta^*)) + O(\frac{\sigma^4 dk'^2 \log d/\delta' \log^2 n}{m\epsilon^2})$$

$$\le (1 - \frac{5}{72}\frac{\mu}{\gamma})^t(L(\theta_0) - L(\theta^*)) + O(\frac{\sigma^4 dk'^2 \log d/\delta' \log^2 n}{m\epsilon^2})$$

$$\le \gamma(1 - \frac{5}{72}\frac{\mu}{\gamma})^{t-1}\|\theta_0 - \theta^*\|_2^2 + O(\frac{\gamma}{\mu}\frac{\sigma^4 dk'^2 \log d \log^2 n}{m\epsilon^2})$$

When $\|\theta_0 - \theta^*\|_2^2 \le \frac{1}{2}\frac{\mu}{\gamma}$, and $n$ is large enough such that

$$n \ge \tilde{\Omega}(\frac{\gamma}{\mu^2}\frac{k'^2 d\sigma^4 T}{\epsilon^2})$$

Then $\|\theta'_t\|_2 \le \|\theta^*\|_2 + \sqrt{\frac{1}{2} + \frac{1}{2}} \le 2$. Thus $\theta_t = \theta'_t$. So we have with probability at least $1 - \delta'$

$$\mu\|\theta'_t - \theta^*\|_2^2 \le L(\theta^T) - L(\theta^*) \le (1 - \frac{5}{72}\frac{\mu}{\gamma})^T(L(\theta_0) - L(\theta^*)) + O(\frac{\gamma}{\mu}\frac{\sigma^4 dk'^2 T \log\frac{dT}{\delta'} \log^2 n}{n\epsilon^2})$$

Thus, take $T = \tilde{O}(\frac{\gamma}{\mu}\log n)$ and $k' = O((\frac{\gamma}{\mu})^2 k)$ we have the result.

$$\square$$

