# OpenReview forum: "Improved Analysis of Sparse Linear Regression in Local Differential Privacy Model"
_ICLR.cc/2024/Conference — ICLR 2024 poster_

### Official Review · Reviewer_ZWvr · 2023-10-31

**Soundness:** 4 excellent
**Presentation:** 3 good
**Contribution:** 3 good
**Rating:** 8
**Confidence:** 3

**Summary:**

The paper studies sparse linear regression in the different local differential privacy models (LDP).

For non-interactive LDP they propose an algorithm with estimation error $\tilde{O}(\frac{d\sqrt{k}}{\epsilon\sqrt{n}})$, and show a lower bound $\Omega(\frac{\sqrt{dk\log d}}{\epsilon \sqrt{n}})$ (for sub-Gaussian covariates). In addition, they show that it is possible to improve the upper bound by a factor $\sqrt{d}$ given public unlabeled covariates.

For interactive LDP they propose an algorithm with estimation error $\tilde{O}(\frac{k\sqrt{d}}{\epsilon\sqrt{n}})$, and show a lower bound $\Omega(\frac{{\sqrt{dk}}}{\epsilon \sqrt{n}})$ (for sub-Gaussian covariates)

**Strengths:**

There are a few new non-trivial results that improve over the state of the art. The paper is well-written, I really enjoyed reading it. The contribution and comparison with prior works is clear. The idea behind Algorithm 1 is very nice and seems to be new (though I'm not an expert in the field, so I'm not 100% sure).

In addition, they found and fixed a bug in one of the results of prior work [1] on the iterative LDP settings that implied an incorrect upper bound. I briefly checked it, and indeed Hoelder's inequality in the proof of Theorem 9 is used incorrectly there, so it is good that this mistake was found and fixed.

**Weaknesses:**

I didn't find major weaknesses. However there is one thing (which I formulate in the Questions below) that is confusing to me.

**Questions:**

Your proof of Theorem 7 looks very similar to the proof of Theorem 9 from [1] (and you mention that). Could you please explain what are important differences, assuming linear regression settings? (I didn't check the details, so maybe I missed something). From the first glance it looks like the proof from [1] works not only for the uniform distribution, but also for 1-sub-Guaussian distributions (modulo their wrong bound in the very beginning), and if it is the case, it should also work for you settings, or did I miss anything important?

And one minor thing: I suggest to move Table 1 to the introduction.

[1] Di Wang and Jinhui Xu. On Sparse Linear Regression in the Local Differential Privacy Model.
IEEE Transactions on Information Theory 2021

---

> ### Author Response · Authors · 2023-11-14
>
> We are grateful for the time and effort that the reviewer has dedicated to reviewing our work.
>
> 1. Answer to the question regarding Theorem 7:
>
> The main differences between Theorem 7 and Theorem 9 are two fold. First we need finer analysis on $B_1$ term due to clipping operation. Second, due to different distributional assumption we adopt a more general lemma related to RIP condition. The main contribution of Theorem 7, as we mentioned, is rectifying the flaws in the previous result. To see greater novelty in proof techniques, please refer to Theorem 34 in the Appendix for the case without the isotropic assumption.
>
> 2. Answer to the question regarding Table 1:
>
> Thank you for your suggestion. In our current version, Table 1 is placed in the Appendix due to space limit. We will definitely move to the paper if additional pages are allowed.

---

> > ### Comment · Reviewer_ZWvr · 2023-11-23
> >
> > Dear Authors,
> >
> > Thank you very much for the clarification! The score remains unchanged.

---

### Official Review · Reviewer_UbUb · 2023-10-31

**Soundness:** 2 fair
**Presentation:** 3 good
**Contribution:** 2 fair
**Rating:** 5
**Confidence:** 3

**Summary:**

This paper studies the problem of sparse linear regression under the local differential privacy setting. The authors provide new lower bound results for this problem with a $k$-sparse underlying model parameter. In addition, the authors develop efficient upper bound algorithms for the same problem.

**Strengths:**

The strengths of the current paper are summarized as follow:
1. The authors provide new lower bound results for sparse linear regression under local differential private model.
2. The authors develop new efficient algorithm for solving the same problem.

**Weaknesses:**

The weaknesses of the current paper:
1. It is unclear the dimension dependence in the lower bound is due to the hardness of the LDP setting or the norm of the data.
2. It is unclear why the authors need the $\ell_1$ norm bound in their results.
3. It is unclear why Assumptions 1 and 2 are both required in the upper bound results.
4. Why it is reasonable to consider the sparse model in the classical setting?
5. The sample complexity requirement seems to be very bad in terms of $d$.

**Questions:**

Here are some additional questions I have for the current paper:
1. For the Remark 2, why the authors claim that the sparse linear models in the non-interactive LDP setting are ill-suited? It seems to me that the dimension dependence in Theorem 1 comes from the norm of the data, what will the result look like if you assume the data vector to be $\ell_2$ norm bounded? In addition, the results in Raskutti et al. 2011 and Cai et al. 2021 seem to assume the data vector to be $\ell_2$ norm bounded.
2. For Theorem 3, why do you assume Assumptions 1 and 2 holds simultaneously? In Assumption 1, you assume $x$ with covariance $\Sigma$, and the transformed data to be Sub Gaussian. In Assumption 2, you further assume $x_i$ has variance $\sigma^2$. In addition, what is the assumption on $\zeta$?
3. If the lower bound has nothing to do with $\ell_1$ norm bound, you should give the results in terms of the $\ell_2$ norm bound.
4. Whether the upper bound results can be extended to the $\ell_2$ norm bound case?

---

> ### Author Response · Authors · 2023-11-14
>
> We are grateful for the time and effort that the reviewer has dedicated to reviewing our work.
>
> 1.Response for W1 and Q1:
>
> **We cannot agree with the reviewer's comment.** Our results confirm that the dependence of dimension is due to the hardness of the LDP setting rather than the norm of the data being upper bounded by $O(\sqrt{d})$.  In fact, in the central DP case, even under some similar assumptions on data distribution, the nearly optimal rate is $O(\sqrt{\frac{k\log d}{n}}+\frac{k\log d}{n\epsilon})$ (see assumptions (P1'), (D1') and (D2') and Theorem 4.4 in Cai et al. 2021).  Note that even in the heavy-tailed response setting with $\|x\|_2\leq O(\sqrt{d})$, we can still get an upper bound which only depends on $\text{Poly}(k, \log d)$ (see Theorem 7 and Assumption 3 in [1]). Similarly, for the non-private setting, Raskutti et al. 2011  show that the optimal rate is $O(\sqrt{\frac{k\log d}{n}})$ in the case where $x$ is $\sigma^2$-subgaussian, indicating that the $\ell_2$-norm of $x$ is upper bounded by $O(\sqrt{d})$ with high probability. We refer the reviewer to Table 1 in Appendix for a detailed comparison with some related work.
>
> [1] Hu, Lijie, et al. "High dimensional differentially private stochastic optimization with heavy-tailed data." Proceedings of the 41st ACM SIGMOD-SIGACT-SIGAI Symposium on Principles of Database Systems. 2022.
>
> 2.Response to W2.
>
> In fact, only our upper bound for the NLDP model relies on the $\ell_1$-norm bound assumption. Specifically,  we need such an assumption in the proof of Theorem 3 to bound the $T_3$ term.
> **Please refer to Page 24 in the Appendix to see how the assumption $\|\theta^{*}\|_1\leq 1$ is applied.** We will introduce an additional factor of $\sqrt{d}$ if we use $\ell_2$-norm assumption instead. As we mentioned, this assumption has been previously studied
> in the literature such as Chen et al. (2023; 2022a) and Fan et al. (2021).
>
> 3. Response to W3
>
> It is notable that Assumption 2 assumes $x$ and $y$ are sub-Gaussian, which is the most commonly used assumption in the literature of sparse linear regression even in the non-private case, such as Raskutti et al. 2011. Assumption 1 is only used for Theorem 3. The usage of $\ell_1$-norm boundedness assumption was just discussed above; $\kappa_\infty$ is vital for providing $\ell_\infty$-norm bound on the inverse of private covariance estimator (Please refer to Page 23 in the Appendix).
>
> 4. Answer to W4
>
>
> Firstly, although $k\ll d$ holds in general, **sparsity $k$ can heavily affect the estimation error.** For instance, when $k=O(d^{\frac{1}{2}})$, ignoring the sparsity structure limits the previous result to an estimation error of $O(\frac{d^\frac{3}{2}}{\sqrt{n}\epsilon})$ in LDP model. In contrast, our work attains rate of $O(\frac{d^{\frac{5}{4} }}{\sqrt{n}\epsilon})$ and $O(\frac{d} {\sqrt{n}\epsilon})$ for NLDP and interactive LDP, respectively.
>
> Secondly, from the theoretical perspective, the significant implication of sparse linear regression in ML, DP and many interconnected problems is the impetus to address  the constraints of previous techniques as they cannot be readily generalized to the general $k$-sparse case. We believe that our methods and some technical lemmas can also be used in other related problems.
>
> Thirdly, as the previous paper already showed that the estimation error will be trivial even when $k=1$ in the case of $n\ll d$. However, the problem remains inadequately understood in the low-dimensional regime, which is the motivation of our research. We aim to give an answer to the question: what is the optimal rate of sparse linear regression in the rich data regime ($n\gg d$)? Sparse linear regression in the low dimensional case $n\gg d$ has received enormous attention but remains under-studied in the DP community.  For example, [2] studies the problem of the sparse linear bandit, highlighting the critical difference between the poor data regime ($n\ll d$) and the rich data regime ($n\gg d$). Thus, we believe our problem is merited to study.
>
>
> [2] Hao, Botao, Tor Lattimore, and Mengdi Wang. "High-dimensional sparse linear bandits." Advances in Neural Information Processing Systems 33 (2020): 10753-10763.
>
> 5. Response to W5
>
> We guess the reviewer refers to the assumption of $n\geq O(d^4)$ in Theorem 3 and 5. Note that it is only used in the non-interactive case and we do not need such an assumption in the interactive setting. As we mentioned in Remark 3, this assumption is to ensure the private covariance matrix is invertible. We can relax such an assumption when there is some public unlabeled data.

---

> ### Author Response · Authors · 2023-11-14
>
> 6. Response to Q2
>
> The reason we assume the whitened covariate $x$ is subgaussian and $x$ has bounded covariance is just for convenience and to make the paper easier to read.
>
> For the assumption on $\zeta$, since we have assumed that both $y$ and $x$ are sub-Gaussian, indicating that $\zeta$ is also sub-Gaussian with variance $\sigma^2$ due to the sub-Gaussian property and our linear model.
>
> 7. Response to Q3
>
> **It is notable that even in the case $||\theta^{*}||_2 \leq 1$, all the lower bounds still hold without any changes to the proofs.** See Page 18 for the proof of Theorem 1, and Page 27 for the proof of Theorem 6. We have $\|\theta_z\|_2\leq \|\theta\|_1\leq 1$. **In the revised version, we added the above comments.**
>
> 8. Response to Q4
>
> In fact, our upper bound in the interactive setting is still valid when $\|\theta^*\|_2\leq 1$. In contrast, our upper bound for the NLDP model relies on such an assumption. Specifically,  if the assumption on $\theta^*$ is loosened to $\ell_2$-norm bounded, it would be hard to bound the $T_3$ term in the proof of Theorem 3 without introducing an additional factor of $\sqrt{d}$. **Please refer to Page 23 in the Appendix to see how the assumption $\|\theta^*\|_1\leq 1$ is applied. However, to make the paper consistent, we assume ${\|\theta^{*}\|_1\leq 1}$.**

---

> > ### Author Response · Authors · 2023-11-16
> >
> > Dear Reviewer,
> > Hope our response has addressed your concerns and positively influenced your perception of the paper, since the deadline is approaching, if you have any further questions just let us know.

---

> > > ### Author Response · Authors · 2023-11-20
> > >
> > > Dear Reviewer,
> > >
> > > Thank you so much for your time and efforts in reviewing our paper. We have addressed your comments in detail and are happy to discuss more if there are any additional concerns. We are looking forward to your feedback and would greatly appreciate you consider raising the scores.
> > >
> > > Thank you,
> > >
> > > Authors

---

### Official Review · Reviewer_qQ5H · 2023-11-01

**Soundness:** 3 good
**Presentation:** 3 good
**Contribution:** 3 good
**Rating:** 6
**Confidence:** 3

**Summary:**

This paper studies sparse linear regression under local differential privacy. Firstly, it establishes a lower bound under a non-interactive LDP protocol for sub-gaussian data. Secondly, it proposes the first upper bound that has a $\sqrt{d}$ gap compared to the aforementioned lower bound. It also demonstrates that this gap can be closed if public unlabeled data is available. Lastly, in the case of sequentially interactive protocol, this paper presents a lower bound and corrects the results of the iterative hard thresholding algorithm from prior work.

**Strengths:**

1. This paper is thorough and clearly written.
2. The problem is well-defined and important.

**Weaknesses:**

The upperbound and lowerbound do not match. It is unclear which bound is tight. Also $n$ has to be greater than $O(d^4)$ to achieve a rate of $O(d)$ in Theorem 3.

**Questions:**

1. Is the l2 norm the right metric for linear regression? For example, Cai at el (2021) consider $\|\theta^{priv}-\theta^*\|_\Sigma$, which corresponds to minimal emprical risk. Do the results also hold under this normalized metric?
2. Is k used in Algorithm 1?
3. Regarding Remark 3, is it necessary to release the covariance matrix privately in LDP model? Can you privatize the two terms in OLS solution together?

---

> ### Author Response · Authors · 2023-11-14
>
> We thank the Reviewer for the careful and detailed review as well as the constructive feedback.
>
> 1.Response to Weakness
>
> We acknowledge the gap between the upper and lower bounds established in our work. However, as sparse linear regression is a fundamental problem in both statistics and differential privacy, we do not think such gaps undermine our contributions. Here are the reasons:
>
>  Only the work (Wang \& Xu (2021))  has studied sparse linear regression in the LDP model. However, the problem is still far from well understood: (1) For lower bounds, previous results only consider the case where $k=1$ and it is technically difficult to extend to the general $k$ sparse case. **Prior to our work, there are no comparable lower bounds!** We give non-trivial proofs for our lower bounds by constructing hard instances that might be instructive for other research problems. (2) For the upper bound in the NLDP model, **we give the first algorithm with a non-trivial upper bound since there is even no previous study due to the one-round communication constraints in the model!**  Moreover, the closed-form private estimator for sparse linear regression is highly efficient, and can readily be applied to other problems. (3) Even the investigation of the upper bound in the interactive setting is still quite deficient. Previous work claims the rate is already optimal for $k=1$. **However, we found a flaw in their approach and the proof of the upper bound is partly mistaken.** We then gave a correct upper bound.
>
> In summary, sparse linear regression in the LDP model is quite difficult and challenging even in the interactive setting but our upper and lower bounds represent a significant leap forward in addressing this insufficiently studied problem.
>
> For the assumption that $n\geq O(d^4)$, note that it is only used in the non-interactive case and we do not need such an assumption in the interactive setting. As we mentioned in Remark 3, this assumption is to ensure the private covariance matrix is invertible. We can relax such an assumption when there is some public unlabeled data.
>
> 2.Response to Q1
>
> Yes, $\ell_2$-norm is a commonly used metric for our problem. Firstly,  many previous works employ the same metric [2] or adopt a metric that can be easily converted to parameter $\ell_2$-norm error bound [3]. Secondly, we can easily transform the $\ell_2$-norm bound to $||\theta^{{priv }}-\theta^*||_\Sigma$ because it is less than
>
> $ \lambda_{\max}(\Sigma)||\theta^{{priv }}-\theta^*||_2^2.$
>
> Theorem 4.2 in [4] adopts $||\theta^{{priv }}-\theta^*||_{\Sigma}$. However, in their proof, they first analyzed the convergence rate of $\ell_2$-norm error.
>
> Then armed with the boundedness and normalized rows of design matrix assumptions, they were able to develop the bound on $||\theta^{{priv }}-\theta^*||_{\Sigma}$ based on $\ell_2$-norm bound. Note that these assumptions are stronger than our Assumptions 1 and 2, therefore here we discuss the error in $\ell_2$-norm.
>
> 3.Answer to Q2 (regarding sparsity $k$ in Algorithm 1)
>
> No, $k$ is not used in our Algorithm 1. However, the sparsity in our proposed estimator $\hat{\theta}^{priv}$ is preserved by setting element-wise soft-thresolding operator's parameter $\lambda_n$ to some constant of size $O(\frac{d\log n \sqrt{\frac{1}{\delta}}}{\sqrt{n}\epsilon})$. $\lambda_n$ serves as a constraint of the $\ell_\infty$-norm error and introduced an extra error term of $O(\sqrt{k})$ when it comes to computing the $\ell_2$-norm error. Note that such phenomenon is quite common in the non-private case, such as the optimal regularization parameter in LASSO [1]
>
> 4. Answer to Q3
>
> In the non-interactive setting, we cannot say we need to privately estimate the covariance matrix. However, for our method, we have to do this due to our closed-form estimator.
>
> If we interpret your question correctly, "Can you privatize the two terms in the OLS solution together?", you are asking whether we can perturb our non-private estimator as a whole. Unfortunately, we cannot adopt the output perturbation method, i.e., adding some Gamma noise to $\hat{\theta}^{OLS}$. First, output perturbation does not preserve LDP. Secondly, according to the DP theory, the scale of the noise should be proportional to the $\ell_2$-norm sensitivity, which can be bounded through clipping operation or if the data are assumed to be bounded. However, this method fails since the magnitude of noise needs to grow polynomially with $\sqrt{\frac{d}{n}}$, causing the estimation error much larger. Moreover, the $\ell_2$-norm sensitivity is bounded only with some probability due to the existence of the inverse of the covariance matrix, indicating the algorithm does not satisfy DP.

---

> > ### Author Response · Authors · 2023-11-14
> > **literature mentioned above**
> >
> > [1] Ye, Fei, and Cun-Hui Zhang. "Rate minimaxity of the Lasso and Dantzig selector for the lq loss in lr balls." The Journal of Machine Learning Research 11 (2010): 3519-3540.
> >
> > [2] Di Wang and Jinhui Xu. "On Sparse Linear Regression in the Local Differential Privacy Model." IEEE Transactions on Information Theory, 67(2):1182–1200, February 2021. ISSN 0018-9448, 1557-9654. doi: 10.1109/TIT.2020.3040406. URL https://ieeexplore.ieee.org/ document/9269994/.
> >
> > [3] Adam Smith, Abhradeep Thakurta, and Jalaj Upadhyay. "Is Interaction Necessary for Distributed Private Learning?" In 2017 IEEE Symposium on Security and Privacy (SP), pp. 58–77, San Jose, CA, USA, May 2017. IEEE. ISBN 978-1-5090-5533-3. doi: 10.1109/SP.2017.35. URL http://ieeexplore.ieee.org/document/7958571/.
> >
> > [4] T. Tony Cai, Yichen Wang, and Linjun Zhang. "The cost of privacy: Optimal rates of convergence for parameter estimation with differential privacy." The Annals of Statistics, 49(5):2825–2850, October 2021. ISSN 0090-5364, 2168-8966. doi: 10.1214/21-AOS2058. Publisher: Institute of Mathematical Statistics.

---

> > > ### Author Response · Authors · 2023-11-21
> > >
> > > Dear Reviewer qQ5H,
> > >
> > > Thanks again for your time and valuable comments.
> > >
> > > Since the discussion stage is about to end, we are writing to kindly ask if our replies have addressed your concerns. Please kindly let us know if you have any additional concerns and we are happy to discuss more.
> > >
> > > Thank you very much!

---

### Meta-Review · Area_Chair_7D4n · 2023-12-12

**Metareview:**

This paper considers a k-sparse variant of the standard least squares problem in the interactive and non-interactive local diff. privacy model. It gives a new lower for the non-interactive case as well as an upper bound in a special centered subgaussian case. Both bounds improve on the state of art and are based on new insights about the problem, even though a significant gap of $\sqrt{d}$ exists between the bounds. In addition, this work gives a better algorithm for the interactive case and a setting where some public data is available. The problem itself is one of the most basic ones and therefore despite some limitations this is a meaningful progress.

**Justification For Why Not Higher Score:**

Bounds are far from being practically useful.

**Justification For Why Not Lower Score:**

As above.

---

### Decision · Program_Chairs · 2024-01-16

Accept (poster)